# A genome-wide association study of mass spectrometry proteomics using a nanoparticle enrichment platform

Karsten Suhre [1,2] ✉, Qingwen Chen [3], Anna Halama [1,2], Kevin Mendez[3], Amber Dahlin[3], Nisha Stephan [1], Gaurav Thareja [1], Hina Sarwath[4], Harendra Guturu[5], Varun B. Dwaraka[6], Ryan Smith[6], Serafim Batzoglou[5], Frank Schmidt[4] & Jessica A. Lasky-Su [3] ✉

Most studies to date of protein quantitative trait loci (pQTLs) have relied on affinity proteomics platforms, which provide only limited information about the targeted protein isoforms and may be affected by genetic variation in their epitope binding. Here we show that mass spectrometry (MS)-based proteomics can complement these studies and provide insights into the role of specific protein isoform and epitope-altering variants. Using the Seer Proteograph nanoparticle enrichment MS platform, we identified and replicated new pQTLs in a genome-wide association study of proteins in blood plasma samples from two cohorts and evaluated previously reported pQTLs from affinity proteomics platforms. We found that >30% of the evaluated pQTLs were confirmed by MS proteomics to be consistent with the hypothesis that genetic variants induce changes in protein abundance, whereas another 30% could not be replicated and are possibly due to epitope effects, although alternative explanations for nonreplication need to be considered on a case-by-case basis.

Protein quantitative trait loci (pQTLs) are important tools in drug target discovery and for generating new hypotheses regarding protein function[1–5]. Most pQTL studies to date have relied on affinity proteomics platforms[6–15], which provide only limited information about the targeted protein isoforms and may be affected by genetic variation in their epitope binding. Mass spectrometry (MS)-based proteomics can complement these studies and provide insights into the role of specific protein isoform and epitope-altering variants. Here we report a genome-wide association study (GWAS) to detect pQTLs using an MS-based proteomics platform with blood plasma samples from a discovery cohort of 1,260 individuals from the USA and a replication cohort of 325 individuals from Asia. We analyzed 1,980 proteins that were quantified in at least 80% of the samples and

identified 364 pQTLs, of which 102 were replicated; among these, 35 have not been reported previously. We further investigated *cis*-pQTLs identified by previous affinity proteomics GWASs for possible epitope effects. In our dataset, 30% of the evaluated pQTLs were confirmed by MS proteomics to be consistent with the hypothesis that genetic variants induce changes in protein abundance, whereas another 30% could not be replicated and are possibly due to epitope effects, although alternative explanations for nonreplication need to be considered on a case-by-case basis. Our study demonstrates the complementarity of the different proteomics approaches and reports pQTLs of biomedical relevance that are not accessible through affinity proteomics, suggesting that many more pQTLs remain to be discovered using MS-based technologies.

[1]Bioinformatics Core, Weill Cornell Medicine-Qatar, Education City, Doha, Qatar. [2]Englander Institute for Precision Medicine, Weill Cornell Medicine, New York, NY, USA. [3]Channing Division of Network Medicine, Department of Medicine, Brigham and Women's Hospital and Harvard Medical School, Boston, MA, USA. [4]Proteomics Core, Weill Cornell Medicine-Qatar, Education City, Doha, Qatar. [5]Seer, Inc., Redwood City, CA, USA. [6]TruDiagnostic Inc., Lexington, KY, USA. ✉e-mail: kas2049@qatar-med.cornell.edu; rejas@channing.harvard.edu

The development of high-throughput affinity proteomics platforms has spurred an increasing number of GWASs with protein traits (see ref. 16 for a comprehensive list). The largest published pQTL studies to date come from deCODE, utilizing the SOMAscan platform with 4,907 aptamers in 35,559 samples from Icelanders[17], and from the UK Biobank Pharma Proteomics Project (UKB-PPP) consortium using the Olink platform with dual antibodies targeting 2,923 proteins in 54,219 samples from UK Biobank participants[18]. A few MS-based GWASs have also been reported but were limited to a smaller number of proteins[19–21]. Collectively, these GWASs have reported thousands of pQTLs that are now available for further exploration such as through Mendelian randomization experiments[15] to identify new drug targets and to further the development of protein-based biomarkers[16].

However, it is important to note that pQTLs discovered using affinity proteomics methods represent genetic associations with protein-binding affinity rather than direct protein abundance, presuming a reproducible link between the number of reagents binding to their targets and the target's abundance. This link may break down when a protein-altering variant (PAV) is located at the aptamer-binding or antibody-binding site, leading to a genotype-dependent readout that does not correspond to a real change in protein abundance[9].

For example, a study on blood pressure identified a strong pQTL for circulating natriuretic peptide precursor A associated with a protein-coding variant (rs5063), but this finding failed to replicate in a sixfold larger study[22]. The authors concluded that the association was artefactual, because the discovery study used an antibody against an epitope in the midregion of the molecule, in contrast to the amino-terminal epitopes used in the replication study.

Such epitope effects can invalidate conclusions drawn from Mendelian randomization experiments, because their basic hypothesis requires that changes in the exposure, that is, the protein abundance, are causal for changes in the disease outcome. Epitope effects can also skew the prediction of protein levels using polygenic scores and confound correlations with other -omics modalities. Therefore, it is important to validate key pQTLs on an independent platform that is not susceptible to the same epitope-binding effects.

However, it should be noted that MS methods do not measure protein abundance directly either, but infer it indirectly from peptide abundances, which are proportional to the amount of digested and ionized protein fragments and their mapping to the protein isoforms that are assumed to be found in the sample.

We previously demonstrated that MS proteomics readouts can reliably distinguish between epitope QTLs and protein abundance QTLs by applying a specific data analysis protocol that excludes PAV-containing peptides from the protein quantification[23]. In the present study, we extended this approach to a larger study cohort and conducted a full GWAS using the MS-based Proteograph proteomics platform (Seer)[24,25].

The Seer technology enhances proteome coverage by using nanoparticle enrichment, followed by a data-independent acquisition protocol implemented on a Bruker timsTOF Pro 2 mass spectrometer (Bruker Daltonics). The following analysis focused on 1,980 proteins that were quantified in at least 80% of the samples (Supplementary Table 1 and Supplementary Fig. 1), out of 5,753 proteins quantified across a discovery cohort (Tarkin) of 1,260 US study participants of diverse backgrounds and a replication phase (Qatar Metabolomics study of Diabetes (QMDiab)) comprising 325 samples from participants of mainly Arab, Indian and Filipino backgrounds (Supplementary Fig. 2).

## Results

### Identification of protein QTLs

In the discovery stage (Tarkin), 364 independent protein associations reached Bonferroni's level of significance ($P < 5 \times 10^{-8}$), involving 295 genetic loci and 274 different proteins, with 177 of these associations located in *cis*. Replication was attempted using 325 samples from the

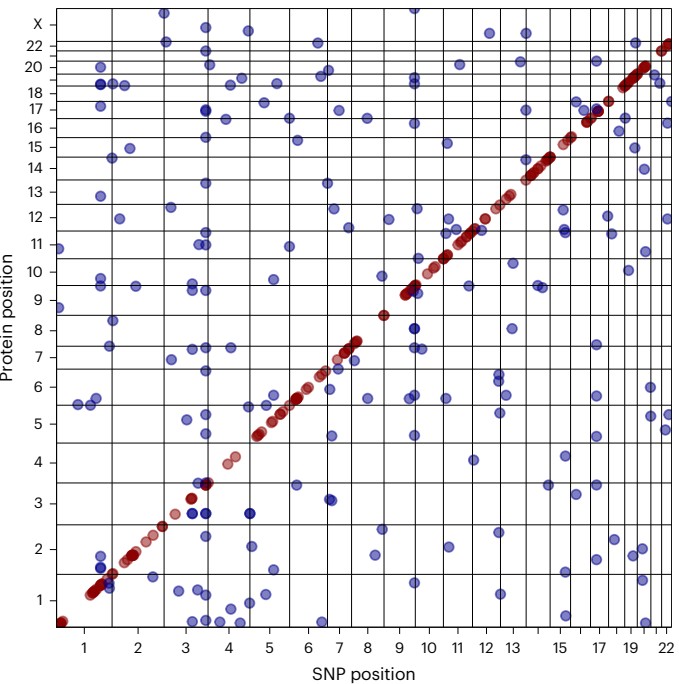

**Fig. 1 | Two-dimensional Manhattan plot.** Grid plot of the genomic position of the variant (SNP position) versus the position of the gene coding for the pQTL protein (protein position). The *cis*-pQTLs are in red and the *trans*-pQTLs in blue (plot data in Supplementary Table 2).

QMDiab study, which included matching genotype and proteomics data. To account for differences in genetic structure between the cohorts, a pQTL was considered replicated if it colocalized between the discovery and replication studies and reached a genome-wide and proteome-wide significance level of $P < 2.53 \times 10^{-11} (5 \times 10^{-8}/1,980)$ in the joint analysis. A total of 102 pQTLs (28.0%) met these criteria. All replicated pQTLs exhibited concordant effect direction (Figs. 1 and 2, Supplementary Table 2, Supplementary Figs. 3 and 4 and Supplementary Data 1 and 2).

The primary reason for the nonreplicated pQTLs is the limited size of the replication cohort: of 70 pQTLs that had 80% replication power, 58 (82.9%) replicated and most (36 out of 38) of the nominally significant ($P < 0.05$) pQTLs had concordant directionality, suggesting that most of the unreplicated pQTLs should be replicable in future larger-scale studies. In addition, 14 of the nonreplicated pQTLs exhibited a significant but different genetic signal in both cohorts, such as intelectin-1 (ITLN1). The ITLN1 pQTL overlaps with a Crohn's disease risk locus, but the role of ITLN1 in the disease mechanism is not clear[26]. We detected a strong ITLN1 signal in the QMDiab cohort that was not present in Tarkin, which may be discernible only in that cohort due to differences in lifestyle, environmental factors or population-specific genetic backgrounds (Supplementary Fig. 5). It would therefore be interesting to analyze the genetic architecture of this locus in future Crohn's disease GWASs in different populations.

Of the 102 replicated pQTLs, 53 and 52 had been identified previously by the deCODE SOMAscan study[17] and the UK Biobank Pharma Proteomics Project (UKB-PPP) Olink study[18], respectively, and 67 (65.7%) had been reported at least once by these and/or other pQTL studies curated by Open Targets[27]. A total of 35 replicated pQTLs (34.3%) were new (Table 1), although 11 of the new MS pQTLs were for proteins that had been assayed by the deCODE and/or UKB-PPP studies, but did not reach genome-wide significance at these loci, suggesting that the respective affinity assays may be targeting different isoforms, may not be reaching their detection limits or may be binding off-targets.

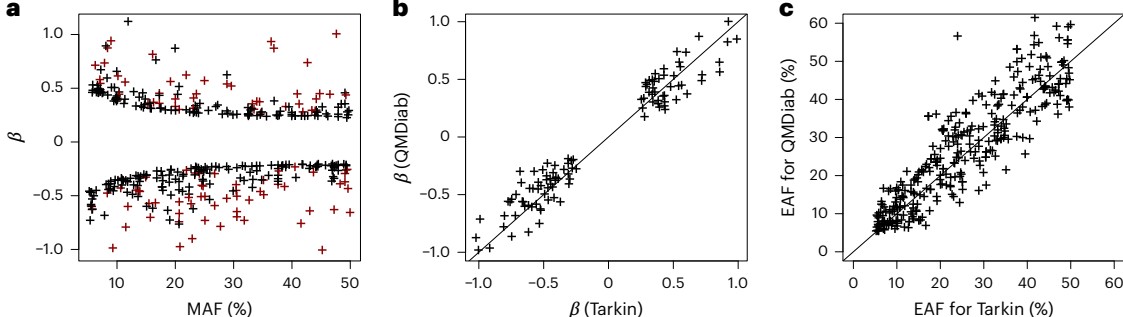

**Fig. 2 | Properties of 364 pQTLs discovered in a GWAS with MS proteomics.**
**a**–**c**, Scatterplots of effect size ($\beta$) versus minor allele frequency (MAF) (**a**), pQTL effect size ($\beta$) from the discovery (Tarkin) versus from the replication (QMDiab) study (**b**) and EAFs for Tarkin versus QMDiab (**c**). All the protein associations that reached a significance level $P < 5 \times 10^{-8}$ in the discovery study are represented, except for **b**, where only replicated pQTLs are shown (plot data in Supplementary Table 2).

## Biomedical relevance of new pQTLs

The 35 new pQTLs overlap with several loci of biomedical relevance, including a *trans*-pQTL for ANGPTL6 at the *COLEC11* locus associated with low-density lipoprotein (LDL)-cholesterol levels, a *trans*-pQTL for BRE (brain and reproductive organ expressed (TNFRSF1A modulator)) at the *CFH* locus associated with age-related macular degeneration and immunoglobulin (Ig)A nephropathy, a *cis*-pQTL for galactosylceramidase associated with inflammatory bowel disease and many others that can now be considered as candidate drug targets for these diseases.

We also identified pQTLs that complement findings from affinity-based studies, such as a *trans*-pQTL on the Olink platform for fucosidase FUCA1 (rs11155297) that we replicated. In addition, we found a *cis*-pQTL for FUCA2 at the same locus, which was not assayed by the Olink platform, but would be expected to account for the *trans*-association (Supplementary Table 2). It is interesting that the strongest association by SOMAscan at this locus was with mannosidase MAN2B2. FUCA1, FUCA2 and MAN2B2 are all enzymes involved in the lysosomal degradation of glycoproteins and glycolipids. A look-up using the Open Targets platform additionally revealed a GWAS signal for 'Total PHF-tau (SNP × SNP interaction) ($P = 2 \times 10^{-8}$)'[28]. These genetic signals obtained from three different proteomics platforms illustrate how pQTLs can be leveraged to generate hypotheses for the drug target discovery process.

## Analysis of pQTLs reported by orthogonal platforms

We then asked whether previously reported affinity pQTLs could be confirmed using an orthogonal MS technology and whether any of these pQTLs could be affected by epitope effects. We examined 319 and 392 *cis*-pQTLs from the deCODE SOMAscan study[17] and the UKB-PPP Olink study[18], respectively, for which matching genotype and protein data were available in our study (Supplementary Tables 7–10). For these pQTLs, we conducted a genetic analysis of the MS proteomics data at the peptide level and computed summary statistics for the associations of all peptides from a given pQTL protein with the respective pQTL variant. To integrate these summary statistics, we employed two approaches: first, a meta-analysis across all peptides (Supplementary Figs. 6 and 7 and Supplementary Data 3–5) and, second, the derivation of an MS-based peptide association (MSPA) score that represents the support for a true protein abundance pQTL at the peptide level.

The MSPA score is defined as the fraction of peptides that support a pQTL at an α level of 1%, weighted by the number of detections of each peptide (Methods). An MSPA score of 1 indicates that, for all detected peptides, the 99% confidence interval (CI) of the effect ($\beta$) does not contain the null, thereby supporting a protein abundance QTL. Conversely, an MSPA score of 0 indicates that none of the peptides from the protein provides statistical support for an association. We argue that the MSPA score is a more intuitive measure for the support of a genuine pQTL at the peptide level than relying on the overall statistical significance at the protein or the peptide meta-analysis level. We therefore focus on the MSPA scores in the following discussion (Fig. 3) and provide additional measures and comparisons of their properties in Supplementary Figs. 8–12. A total of 52 out of the 319 SOMAscan pQTLs (16.3%) and 62 of the 392 Olink pQTLs (15.8%) had an MSPA score ≥0.8.

Identification of an epitope effect, however, is more challenging, because lack of support for an affinity proteomics pQTL using MS methods may also result from other factors, including insufficient statistical power, limited sensitivity of the MS method, differences between targeted isoforms between platforms or the possibility that genetic variants represent different genetic signals between study populations.

To determine which affinity pQTLs could be expected to be replicated using our MS proteomics data, we conducted a power analysis (Methods). A total of 120 (for deCODE) and 167 (for UKB-PPP) pQTLs had >99% probability of reaching a multiple-testing corrected $P$ value of $P < 0.05/319$ (for deCODE) and $P < 0.05/392$ (for UKB-PPP) in Tarkin. Of these sufficiently powered pQTLs, 39 (32.5% for deCODE) and 49 (29.3% for UKB-PPP) replicated and had an MSPA score >0.8, whereas 36 (30.0% for deCODE) and 55 (32.9% for UKB-PPP) did not replicate and had an MSPA score <0.2 (Supplementary Tables 9 and 10). Our analysis also confirms a previously reported[29] epitope effect for the pQTL of GDF15, which was reported by deCODE (rank 71 in Supplementary Table 9) and UKB-PPP (rank 133 in Supplementary Table 10) and exhibited an MSPA score of 0 in our study.

## Genetic association with protein isoforms

Conflicting directionality of QTLs at the peptide level can indicate the concurrent presence of different isoforms. We identified ten such cases in deCODE and eight in UKB-PPP (Supplementary Tables 9 and 10). One example is variant rs2052534, which is a pQTL for the Serine Peptidase Inhibitor Kazal Type 5 (SPINK5) reported by both UKB-PPP and deCODE, as well as in our study on variants in linkage disequilibrium (LD) ($r^2 = 1$). The pQTL had an MSPA score of 0.5, indicating support by some, but not all, analyzed peptide associations. SPINK5, also known as Lymphoepithelial Kazal-Type-Related Inhibitor or LEKTI, plays a role in skin and hair morphogenesis and protection of the mucous epithelia. Mutations in *SPINK5* have been linked to skin disorders characterized by ichthyosis[30], such as Netherton's syndrome, as well as to hair abnormalities. GWASs have further associated *SPINK5* variants with lung phenotypes (for example, chronic obstructive pulmonary disease, forced expiratory volume) and pancreatitis. It has been previously shown that that the *SPINK5* gene generates three classes of transcripts encoding three different LEKTI isoforms, which vary in their carboxy-terminal portion[31], corresponding to three UniProt entries:

**Table 1 | List of 35 previously unreported replicated pQTLs discovered using the Seer Proteograph platform**

| Protein[a] | Description | rsID SNP (build 37) | Eff. | EAF (%) | $\beta$ | P value | Joint $\beta$ | Joint P value | PP H4 (%) | Locus | Trait association | Aff. |
|---|---|---|---|---|---|---|---|---|---|---|---|---|
| BRE | Brain and reproductive organ-expressed (TNFRSF1A modulator) | rs67908756 1:196821380:T:G | G | 20.8 | −0.991 | $1.9\times10^{-65}$ | −0.935 | $3.2\times10^{-85}$ | 98.5 | CFH | Age-related macular degeneration IgA nephropathy | – |
| LEFTY1 | Left–right determination factor 1 | rs360057 1:226074563:T:G | G | 35.2 | −0.507 | $9.9\times10^{-37}$ | −0.462 | $2.5\times10^{-41}$ | 98.2 | cis-pQTL | - | – |
| ANGPTL6 | Angiopoietin-like 6 | rs6542680 2:3640142:C:T | C | 17.3 | 0.343 | $6.1\times10^{-12}$ | 0.326 | $2.3\times10^{-15}$ | 88.3 | COLEC11 | Liver enzymes, lipid levels | – |
| SPTBN1 | Spectrin, β, nonerythrocytic 1 | rs6740893 2:54834380:G:A | A | 23.4 | −0.306 | $4.4\times10^{-11}$ | −0.325 | $7.9\times10^{-15}$ | 84.4 | cis-pQTL | Heel bone mineral density, eGFR, cystatin C | – |
| IGKV2D-29 | Immunoglobulin kappa variable 2D-29 | rs62148537 2:95365118:G:C | C | 7.1 | 0.564 | $5.1\times10^{-13}$ | 0.529 | $1.0\times10^{-15}$ | 63.9 | cis-pQTL | – | – |
| C2orf40 | ECRG4 augurin precursor | rs13014521 2:106687456:G:C | C | 11.5 | −0.809 | $5.9\times10^{-38}$ | −0.806 | $1.6\times10^{-48}$ | 99.7 | cis-pQTL | Coxarthrosis | S |
| MAN1C1 | Mannosidase, α, class 1C, member 1 | rs4305381 3:126249877:A:C | C | 24.5 | −0.420 | $7.6\times10^{-18}$ | −0.442 | $3.6\times10^{-24}$ | 93.6 | C3orf22 | – | S |
| AMBP | α₁-Microglobulin/bikunin precursor | rs1056522 3:126261345:G:A | A | 30.6 | −0.713 | $3.9\times10^{-68}$ | −0.682 | $2.6\times10^{-83}$ | 98.1 | CHST13 | – | S,o |
| CCDC132 | Coiled-coil domain containing 132 | rs1056522 3:126261345:G:A | A | 30.6 | −0.665 | $6.3\times10^{-58}$ | −0.654 | $1.1\times10^{-70}$ | 99.1 | CHST13 | – | – |
| ZNF618 | Zinc finger protein 618 | rs9835865 3:186380167:A:T | T | 49.2 | 0.421 | $2.7\times10^{-24}$ | 0.404 | $3.6\times10^{-29}$ | 93.8 | HRG | – | – |
| GNB2 | guanine nucleotide binding protein (G protein), beta polypeptide 2 | rs1042464 3:186395572:A:T | A | 49.1 | −0.310 | $5.8\times10^{-15}$ | −0.316 | $1.6\times10^{-19}$ | 97.4 | HRG | – | – |
| CTSS | Cathepsin S | rs5030062 3:186454180:A:C | C | 38.9 | 0.313 | $3.9\times10^{-15}$ | 0.325 | $9.4\times10^{-21}$ | 98.1 | KNG1 | – | s,o |
| MSN | Moesin | rs710446 3:186459927:T:C | C | 44.2 | 0.384 | $6.5\times10^{-24}$ | 0.362 | $2.3\times10^{-27}$ | 67.6 | KNG1 | – | S |
| HLA-G | MHC, class I, G | rs2517718 6:29916391:A:C | C | 43.1 | −0.658 | $3.2\times10^{-66}$ | −0.644 | $1.3\times10^{-86}$ | 100.0 | cis-pQTL | Heel bone mineral density | S |
| FUCA2 | Fucosidase, α-L-2, plasma | rs11155297 6:143825104:G:T | T | 25.1 | −0.523 | $7.3\times10^{-31}$ | −0.547 | $3.1\times10^{-43}$ | 100.0 | cis-pQTL | Total PHF–tau (SNP×SNP interaction) | - |
| PEBP4 | Phosphatidylethanolamine-binding protein 4 | rs3087803 8:22570901:C:T | T | 9.0 | 0.925 | $2.4\times10^{-47}$ | 0.935 | $1.1\times10^{-62}$ | 52.3 | cis-pQTL | – | - |
| ECM2 | Extracellular matrix protein 2, female organ and adipocyte specific | rs12338938 9:95281459:A:G | G | 37.7 | −0.287 | $3.7\times10^{-13}$ | −0.272 | $2.6\times10^{-15}$ | 53.3 | cis-pQTL | Blood pressure, lung function, BMI, hemorrhoids | - |
| FABP4 | Fatty acid-binding protein 4, adipocyte | rs597988 9:136144284:T:A | A | 34.0 | 0.361 | $1.2\times10^{-15}$ | 0.375 | $9.7\times10^{-19}$ | 57.2 | ABO | Blood clot-related disorders | S |
| FABP5 | Fatty acid-binding protein 5 (psoriasis associated) | rs529565 9:136149500:T:C | C | 33.5 | 0.288 | $7.5\times10^{-11}$ | 0.322 | $1.9\times10^{-17}$ | 97.1 | ABO | Blood clot-related disorders | s,O |
| PFKP | Phosphofructokinase, platelet | rs7920986 10:3101810:A:G | A | 21.9 | −0.279 | $2.7\times10^{-9}$ | −0.281 | $8.2\times10^{-12}$ | 72.0 | - | | – |

**Table 1 (continued) | List of 35 previously unreported replicated pQTLs discovered using the Seer Proteograph platform**

| Protein[a] | Description | rsID SNP (build 37) | Eff. | EAF (%) | β | P value | Joint β | Joint P value | PP H4 (%) | Locus | Trait association | Aff. |
|---|---|---|---|---|---|---|---|---|---|---|---|---|
| MMRN2 | Multimerin 2 | rs34587013 10:88696622:C:G | G | 8.3 | −0.493 | $2.3\times10^{-12}$ | −0.496 | $5.2\times10^{-16}$ | 74.3 | cis-pQTL | - | S |
| TSKU | Tsukushi, small leucine rich proteoglycan | rs1149596 16:76469093:C:T | T | 13.9 | −0.477 | $2.8\times10^{-17}$ | −0.444 | $1.1\times10^{-20}$ | 95.0 | cis-pQTL | Vitamin D levels | – |
| FXYD2 | FXYD domain containing ion transport regulator 2 | rs4936409 11:117694392:A:G | G | 48.3 | −0.406 | $6.7\times10^{-25}$ | −0.397 | $1.5\times10^{-31}$ | 99.9 | cis-pQTL | Urate levels | – |
| PRB1 | Proline-rich protein BstNI subfamily 1 | rs7966710 12:11522616:G:A | G | 26.7 | −0.609 | $3.8\times10^{-38}$ | −0.588 | $3.0\times10^{-44}$ | 86.9 | cis-pQTL | - | – |
| GALC | Galactosylceramidase | rs380142 14:88393918:A:C | C | 45.2 | −1.022 | $1.1\times10^{-205}$ | −0.997 | $<1\times10^{-300}$ | 89.2 | cis-pQTL | Inflammatory bowel disease | – |
| IGHV2-70[b] | Immunoglobulin heavy variable 2-70 | rs10134517 14:107173745:T:C | C | 29.6 | 0.504 | $1.4\times10^{-28}$ | 0.507 | $7.4\times10^{-38}$ | 94.7 | cis-pQTL | - | – |
| IGHV2-70[c] | Immunoglobulin heavy variable 2-70 | rs10136560 14:107195868:C:G | G | 28.8 | 0.523 | $1.6\times10^{-31}$ | 0.485 | $8.0\times10^{-35}$ | 68.5 | cis-pQTL | - | – |
| LIPC | Lipase, hepatic | rs1077835 15:58723426:A:G | G | 26.7 | −0.311 | $1.1\times10^{-12}$ | −0.290 | $7.5\times10^{-14}$ | 54.4 | cis-pQTL | Lipid levels, vitamin D | – |
| HPR | Haptoglobin-related protein | rs763665 16:72078043:C:T | T | 15.3 | 0.343 | $2.1\times10^{-10}$ | 0.357 | $7.0\times10^{-14}$ | 78.7 | cis-pQTL | Lipid levels | – |
| FN3K | Fructosamine 3 kinase | rs3848403 17:80693899:C:T | T | 50.0 | −0.673 | $6.0\times10^{-70}$ | −0.632 | $7.6\times10^{-83}$ | 85.9 | cis-pQTL | HbA1c levels | –;– |
| CDH19 | Cadherin 19, type 2 | rs985088 18:64116504:G:C | C | 13.8 | −0.471 | $1.6\times10^{-16}$ | −0.430 | $3.2\times10^{-17}$ | 81.0 | cis-pQTL | - | – |
| RCN3 | Reticulocalbin 3, EF-hand calcium-binding domain | rs73582463 19:50037446:C:G | G | 8.4 | 0.858 | $2.9\times10^{-36}$ | 0.788 | $6.9\times10^{-42}$ | 89.7 | cis-pQTL | Sex hormones, lipid levels | S |
| KLHDC1 | Kelch domain containing 1 | rs2424961 20:31694060:C:T | C | 41.9 | 0.263 | $3.8\times10^{-11}$ | 0.258 | $3.0\times10^{-13}$ | 50.1 | BPIFB4 | - | – |
| BPIFA1 | BPI fold containing family A, member 1 | rs6059187 20:31828265:A:G | G | 49.1 | −0.388 | $1.3\times10^{-24}$ | −0.385 | $5.3\times10^{-32}$ | 97.0 | cis-pQTL | Skin tanning | S |
| IGLV1-51 | Immunoglobulin λ variable 1-51 | rs6001756 22:22671670:A:G | G | 12.6 | −0.589 | $9.4\times10^{-24}$ | −0.534 | $1.2\times10^{-26}$ | 43.9 | cis-pQTL | - | – |

Details are in Supplementary Table 2. [a]Protein, HGNC name of the protein-coding gene; Eff., effect allele; EAF, effect allele frequency; β, effect size; P value, association P value; joint β and P value, joint effect size and P value for Tarkin and meta-analyzed Tarkin+QMDiab; PP H4, posterior probability of coloc hypothesis H4 being true (pQTL shares a genetic signal in Tarkin and QMDiab); Locus, for trans-pQTLs the most likely causal gene according to the Open Targets scoring scheme is listed, cis-pQTL otherwise; Trait association, summary of Open Targets disease association look-up; Aff., indicates whether the protein was assayed by affinity proteomics (O, protein assayed by UKB-PPP Olink study; S, protein assayed by deCODE SOMAscan study; upper case, marginally significant ($P > 5\times10^{-8}$ and $P < 0.05$); lower case, nonsignificant ($P > 0.05$)); MHC, major histocompatibility complex. [b]Isoform A0A0C4DH43. [c]Isoform P01814.

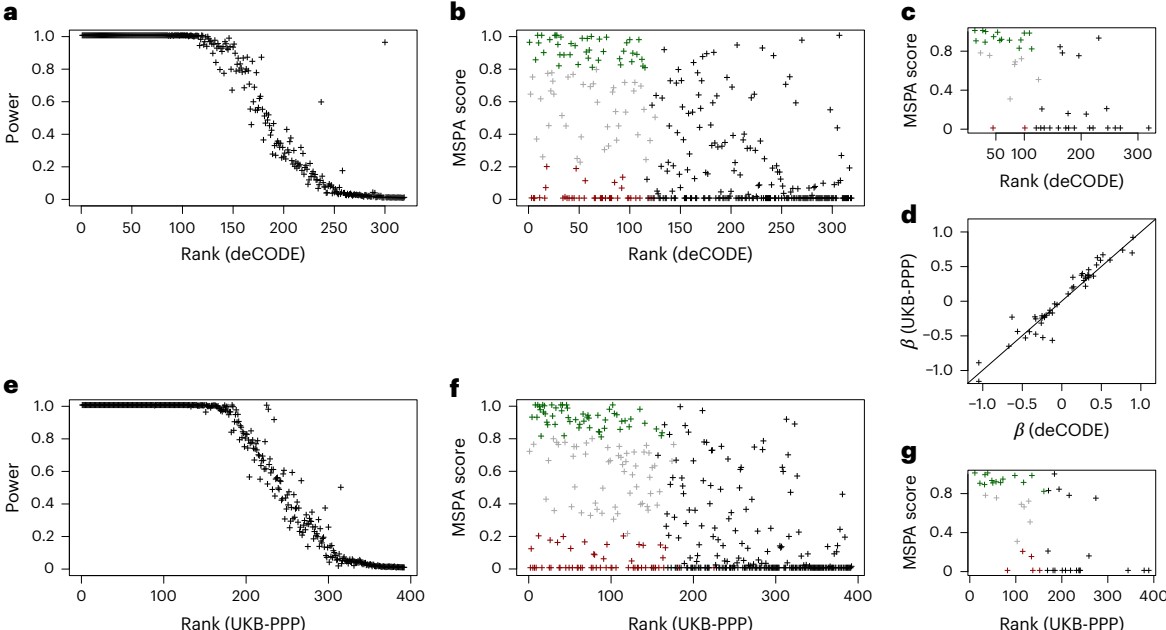

**Fig. 3 | MSPA scores and statistical power to replicate as a function of pQTL rank. a–g**, Scatterplots of the power to replicate a pQTL from the deCODE SOMAscan (**a**) and the UKB-PPP Olink (**e**) studies against the ranks of the affinity proteomics pQTLs. Scatterplots of individual MSPA scores against the ranks of the affinity proteomics pQTLs of the deCODE SOMAscan (**b**) and the UKB-PPP Olink (**f**) studies, starting with the lowest *P* value: 120 out of 319 pQTLs (37.6%) for deCODE and 167 out of 392 pQTLs (42.6%) for UKB-PPP had >99% power at

a significance level of $P < 0.05/319$ and $P < 0.05/319$ for SOMAscan and Olink, respectively, colored to indicate likely protein abundance QTLs (MSPA score >0.8; green) and likely epitope effect-driven pQTLs (MSPA score <0.2; red). MSPA scores were limited to 46 pQTLs that were reported on the same variant in deCODE (**c**) and UKB-PPP (**g**). Scatterplots of the effect size (*β*) of the 46 pQTLs reported deCODE and UKB-PPP (**d**) (plot data in Supplementary Tables 9, 10 and 12).

the 1,064 amino acid canonical form Q9NQ38, isoform Q9NQ38-3 that contains a 30-amino acid insert at position 915 and isoform Q9NQ38-2, which is truncated at position 913, followed by an additional three amino acids (Fig. 4). A fourth isoform (E5RFU9) is truncated at position 158 with an additional 43 amino acids and is likely to be proteolytically degraded. The additional amino acids of Q9NQ38-2 and E5RFU9 were not detected in the MS analysis. Consequently, four protein groups were generated (color coded black, green, red and blue in the Forest plot for the meta-analysis in Fig. 4). These different protein groups associated with the genetic variant rs2052534 at different strengths (for a multiple alignment of the detected peptides and the SPINK5 isoforms, see Supplementary Table 11). These observations can be explained parsimoniously by a genetic variant that influences splicing near residue Lys913, thereby increasing the generation of Q9NQ38-3 isoforms while decreasing that of Q9NQ38. Supporting this hypothesis, the Genotype-Tissue Expression (GTEx) project reports a strong splice QTL for this variant in the relevant tissues (skin and esophagus). This example underscores both the complexity and the level of detail that can be obtained by combining MS proteomics approaches with genetic analyses.

### Validity of the MSPA score

To further support the validity of the MSPA score as a proxy for the detection or nondetection of a genetic association and its potential to identify true positive protein abundance pQTLs, we selected all pQTLs that were reported on the same genetic variant by deCODE and UKB-PPP, and for which matching protein and genetic data were also available in Tarkin and QMDiab. This set of 46 pQTLs served as a reference set for comparing effect sizes and directionality without having to rely on proxy SNPs (Supplementary Table 12). Several key observations emerged from this set of pQTLs. First, in contrast to the overall distribution of the MSPA scores (Fig. 3b,f)—where there were many high ranking pQTLs that had low or zero MSPA scores—the top

ranking pQTLs in this reference set almost all had high MSPA scores (Fig. 3c,g). This finding suggests that these pQTLs were likely unaffected by epitope effects. This inference is further supported by the near-perfect correlation of the effect sizes between the Olink and the SOMAscan platforms (Fig. 3d). Given that these pQTLs were preselected based on their detection across both affinity platforms, it appears that a pQTL being detected by both affinity platforms is a strong indicator of a true protein abundance QTL and the absence of an epitope effect. This is a reasonable assumption because it is unlikely that two different affinity binders target the same surface area of a protein and produce a similar epitope readout.

We further investigated whether a candidate epitope-changing variant could be identified for the pQTLs colored in red in Fig. 3b,f. We queried the Ensembl database[32] for coding variants that could potentially alter epitopes (Supplementary Table 13) and found that, for 22 out of 36 SOMAscan pQTLs and 29 out of 55 Olink pQTLs, such a variant had been reported (requiring LD $r^2 > 0.8$). The lead pQTL SNP or a SNP in perfect LD ($r^2 = 1$) was apparently epitope changing in all but 5 of the 51 cases. In addition, in 15 out of these 51 cases, a PAV-containing peptide was also detected on the Proteograph platform, with heterozygotes exhibiting approximately half the protein level, further confirming the quantification of the protein variant in blood (Supplementary Fig. 13). In eight cases (SERPING1, CPN2, SERPINA1, ENO3, HDGF, APOBR, IGFBP3 and APOL1), both alleles—the reference (REF) and alternate (ALT)—were detected and showed significant associations with the coding variant, whereas all non-PAV-containing peptides did not associate with the variant. Thus, the absence of a protein abundance pQTL in these cases is not attributable to limitations in peptide quantification.

To rule out that differences in genetic architecture contributed to a low MSPA score for some of the pQTLs, we computed a coloc-MSPA score based on genetic colocalization, defined as the weighted fraction of those peptides for which coloc favors a shared (H4) or a different

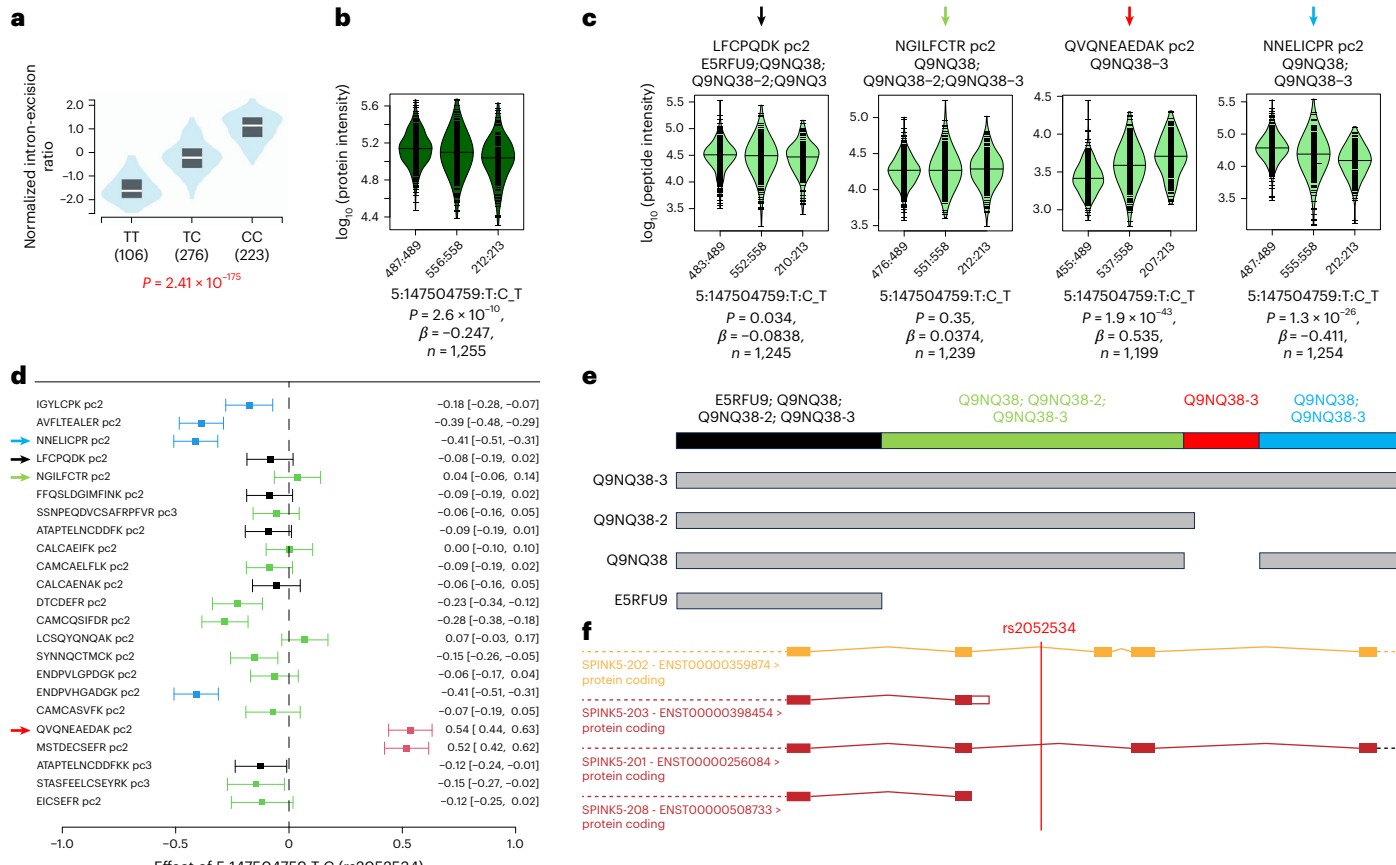

**Fig. 4 | Peptide level analysis of the rs2052534 SPINK5 pQTL. a–c**, Violin plots of a splice QTL from GTEx (**a**), SPINK5 at the protein level (**b**) and SPINK5 at the peptide level for representative peptides of the four protein groups (**c**). **d**, Forest plot of the peptide level meta-analysis. **e**, UniProt isoforms and protein groups. **f**, Gene structure indicating the alternative splice forms (black, green, red and blue) from Ensembl with the position of rs2052534 indicated. The C-allele of rs2052534 leads to a higher exon excision ratio and results in lower levels of the peptide QVQNEAEDAK, which is specific for the intron-retaining Q9NQ38-3

isoform (note the reverse order of the alleles in **a** versus **b** and **c**). This SPINK5 pQTL illustrates how peptide level information can be used to identify different isoform QTLs. However, it also reveals current limitations in the automated quantification of such isoforms, because this information did not translate to the protein level QTL, which in this case reflects the average of the isoform signals. A multiple alignment of all detected peptides is shown in Supplementary Table 11, violin plots for all peptide associations are provided as Supplementary Data 5 and regional association plots are in Supplementary Figure 16.

(H3) signal between the UKB-PPP and Tarkin studies (Methods and Supplementary Table 10). Out of 224 pQTLs with MSPA < 0.2, only one pQTL (ITGA2) had a coloc-MSPA > 0.8 for H3, none for H4 and 205 had a coloc-MSPA < 0.2 for the combination of H3 or H4.

## Discussion

In summary, our analysis suggests that >30% of the affinity proteomics pQTLs are reproducible by MS proteomics in a study of our size, whereas another 30% cannot be replicated and may thus be attributable to epitope effects. The remainder might be cases where a genetic variant interferes with the affinity binding, but at the same time might affect protein abundance via some other biological pathway. Among the 76 pQTLs with MS support (Supplementary Table 17) were 24 that had a disease-relevant association in the GWAS catalog (Table 2), whereas, among the 91 pQTLs without MS support (Supplementary Table 13), there were 17 (Table 3).

This study presents a comprehensive GWAS using the MS-based Seer Proteograph platform, accompanied by a full replication, and employs a proteomics data analysis protocol that accounts for genetic variants within the analyzed peptide[23]. Our methodology not only identified new pQTLs on proteins previously unassessed by affinity proteomics platforms but also re-examined previously reported affinity pQTLs for potential confounding due to epitope-altering variants. We estimate that 30% of the sufficiently powered pQTLs that we

evaluated may be influenced by such effects, regardless of the affinity platform. However, given the limitations in statistical power that restricted evaluation to only the strongest affinity pQTLs—which are the ones most likely to be enriched for epitope effects due to ascertainment bias—this estimate should be considered as an upper bound and affinity binders targeting different isoforms may also explain some cases.

We reported a total of 364 pQTLs, of which 102 were successfully replicated in an independent population, at a replication rate that matches expectations based on post-hoc power calculations. We also identified instances where nonreplication may be explained by differences in genetic architecture and possibly also differences in environmental and lifestyle factors between the discovery and replication cohorts, which are drawn from very different populations. Nevertheless, this diversity enhances the robustness and translatability of the replicated pQTLs across populations.

The use of a proteome FASTA library that accounts for PAVs was central to our study. Without using this approach, a very high number of false-positive pQTLs would have been detected, as we previously discussed[23]. Traditional bottom-up proteomics data analysis pipelines often rely on a limited peptide library for protein quantification, where the presence of a single peptide with a large effect can skew the overall quantification. A PAV-containing peptide would not be detected in homozygotes of the alternative allele and heterozygotes would have half the peptide level. Inclusion of such

**Table 2 | Disease-relevant pQTLs from deCODE and UKB-PPP with >99% power that were replicated and had an MSPA score >0.8**

| Assay (platform) | Protein description | GWAS trait | P value | PMID |
|---|---|---|---|---|
| ACP1 (S,O) | Acid phosphatase 1 | Estimated glomerular filtration rate | $1\times10^{-11}$ | 31152163 |
| ASPN (S) | Asporin | Systolic blood pressure | $9\times10^{-13}$ | 30224653 |
| CD109 (O) | CD109 molecule | Heel bone mineral density | $8\times10^{-10}$ | 30598549 |
| CFB (O) | Complement factor B | Age-related macular degeneration | $6\times10^{-31}$ | 21665990 |
| COLEC11 (S) | Collectin subfamily member 11 | Liver enzyme levels | $1\times10^{-34}$ | 33972514 |
| CTSB (S,O) | Cathepsin B | Parkinson's disease | $4\times10^{-16}$ | 31701892 |
| CYTL1 (S) | Cytokine like 1 | Peak expiratory flow | $4.2\times10^{-37}$ | 30804560 |
| F11 (S,O) | Coagulation factor XI | Venous thromboembolism | $8\times10^{-96}$ | 31676865 |
| FGL1 (S,O) | Fibrinogen-like 1 | Lung function (FEV$_1$/FVC) | $6\times10^{-21}$ | 30595370 |
| IL7R (O) | Interleukin 7 receptor | Hay fever and/or eczema | $1\times10^{-48}$ | 31361310 |
| LPA (O) | Lipoprotein(a) | Coronary artery disease | $3\times10^{-103}$ | 33020668 |
| LTBP3 (O) | Latent transforming growth factor β-binding protein 3 | Lung function (FEV$_1$/FVC) | $6\times10^{-13}$ | 30595370 |
| MBL2 (S,O) | Mannose-binding lectin 2 | Heel bone mineral density | $4.7\times10^{-13}$ | 30598549 |
| MFAP2 (S) | Microfibril-associated protein 2 | Lung function (FEV$_1$/FVC) | $7\times10^{-76}$ | 30595370 |
| NID2 (S,O) | Nidogen 2 | Colon polyp | $6\times10^{-11}$ | 34594039 |
| PLTP (S,O) | Phospholipid transfer protein | HDL-cholesterol levels | $6\times10^{-263}$ | 34226706 |
| SFTPD (O) | Surfactant protein D | Lung function (FEV$_1$/FVC) | $2.1\times10^{-19}$ | 34226706 |
| SHBG (O) | Sex hormone-binding globulin | Type 2 diabetes (adjusted for BMI) | $1\times10^{-10}$ | 30297969 |
| TGFBI (S,O) | Transforming growth factor β induced | Insomnia | $4\times10^{-17}$ | 30804565 |
| VMO1 (O) | Vitelline membrane outer layer 1 homolog | LDL-cholesterol levels | $1.6\times10^{-11}$ | 32493714 |

Shown here are pQTLs with >99% power to replicate in Tarkin that overlap with a disease-relevant association in the GWAS catalog (LD, $r^2 > 0.7$) and were replicated and had an MSPA score >0.8. The assay (platform) indicates the respective assay used in the deCODE or SOMAscan (S) and UKB-PPP or Olink (O) study. PubMed ID (PMID) and the strength of the GWAS trait association (P value) are indicated. Further details are in Supplementary Table 17. FEV$_1$, forced expiratory volume in 1s; FVC, forced vital capacity.

**Table 3 | Disease-relevant pQTLs from deCODE and UKB-PPP with >99% power that were not replicated and had an MSPA score <0.2**

| Assay (platform) | Protein description | GWAS trait | P value | PMID | Missense variant identified | PAV peptide detected |
|---|---|---|---|---|---|---|
| APOA5 (S) | Apolipoprotein A5 | Cardiovascular disease | $7.1\times10^{-14}$ | 33959723 | x | . |
| APOBR (O) | Apolipoprotein B receptor | Crohn's disease | $6.9\times10^{-22}$ | 26192919 | x | 2 |
| APOL1 (O) | Apolipoprotein L1 | Glomerulosclerosis | $5\times10^{-13}$ | 20668430 | x | 2 |
| F13B (O) | Coagulation factor XIII B chain | Systolic blood pressure | $3\times10^{-9}$ | 33230300 | . | . |
| FCGR2A (S) | Fcγ receptor IIa | Ulcerative colitis | $1.4\times10^{-41}$ | 26192919 | . | . |
| FCGR2A (O) | Fcγ receptor IIa | Aspartate aminotransferase levels | $4\times10^{-34}$ | 33462484 | x | 1 |
| GDF15 (S) | Growth differentiation factor 15 | Severity of nausea and vomiting of pregnancy | $2\times10^{-41}$ | 29563502 | x | 1 |
| ICAM1 (O) | Intercellular adhesion molecule 1 | Liver enzyme levels | $3\times10^{-11}$ | 33972514 | x | . |
| IGFBP3 (O) | Insulin-like growth factor-binding protein 3 | Pulse pressure | $6.7\times10^{-108}$ | 33230300 | x | 2 |
| ITGB6 (O) | Integrin subunit β$_6$ | Adolescent idiopathic scoliosis | $2\times10^{-31}$ | 30019117 | . | . |
| ITIH3 (S) | Inter-α-trypsin inhibitor heavy chain 3 | Bipolar disorder | $5\times10^{-14}$ | 33479212 | x | 1 |
| ITIH3 (O) | Inter-α-trypsin inhibitor heavy chain 3 | Osteoarthritis (with total hip replacement) | $1\times10^{-16}$ | 34450027 | . | . |
| LPL (O) | Lipoprotein lipase | Coronary artery disease | $3.9\times10^{-11}$ | 29212778 | x | . |
| MFGE8 (O) | Milk fat globule EGF and factor V or VIII domain containing | Liver enzyme levels | $7\times10^{-20}$ | 33972514 | . | . |
| SERPING1 (S) | Serpin family G member 1 | Lung function (FEV$_1$/FVC) | $1.8\times10^{-10}$ | 30804560 | x | 2 |
| TGOLN2 (O) | Trans-Golgi network protein 2 | HDL-cholesterol levels | $1\times10^{-12}$ | 32203549 | x | . |
| TNFSF13 (O) | Tumor necrosis factor superfamily member 13 | IgA nephropathy | $9\times10^{-11}$ | 22197929 | x | . |

As Table 2, but for pQTLs that did not replicate and had an MSPA score <0.2. The presence of a missense variant and the detection of one or both protein-altering alleles using the MS platform are reported. Further details are in Supplementary Table 13.

PAV-containing peptides in protein quantification would thus lead to the equivalent of an epitope effect—that is, a pQTL signal that does not fully reflect genotype-dependent protein abundance. Moreover, PAV peptides corresponding to the alternative allele are absent from an in-silico digest library of the standard UniProt database. Indeed, using data from a standard proteomics data-processing run, we observed cases where fragment spectra of these peptides mismatched to peptides from other proteins and, in extreme cases, led to false protein identifications.

Our study also has some caveats. Mapping proteins across platforms using UniProt identifiers can be challenging, because affinity proteomics platforms sometimes report multiple protein identifiers when they are targeting protein complexes or use the UniProt identifier specific to the proteoform employed to generate the affinity binder, whereas alternative proteoforms may be included in the MS library, occasionally leading to annotations by protein groups rather than specific proteins. Furthermore, mapping UniProt IDs to gene names can also be challenging, especially when the underlying database versions do not match between studies. The level of matching between the MS and affinity protein identifiers has therefore been reported in the respective tables.

It should also be acknowledged that nanoparticle-enriched MS proteomics methods are not entirely free of potential 'epitope-like' effects. Although less likely due to their less specific electrostatic and hydrophobic interactions, protein–nanoparticle binding can, in principle, be modified by genetic variation. Genotype-associated missingness could potentially indicate such effects, though an association observed with more than one nanoparticle run makes them less likely. We found that 15.3% of the proteins involved in our pQTLs had a missingness pQTL, whereas 51.4% of our pQTL associations are supported by more than one nanoparticle run (Supplementary Note 1).

Although MS proteomics is not biased toward any particular set of preselected proteins, it is biased toward protein isoforms that are present in the utilized database, proteins that are enriched using one of the five nanoparticles and proteins that can be cleaved into peptides detected by the applied MS proteomics method, such as highly abundant proteins. There are also some differences in the protein panels covered by the different technologies. Relative to their respective panel size, the Seer platform covers the largest fraction of cytoplasmic proteins, whereas Olink leads in membrane proteins. SOMAscan has the lowest fraction of extracellular proteins but most proteins originate from the nucleus. As expected, low abundance proteins, such as cytokines, are less frequently detected by the Seer platform (Supplementary Table 6). As an additional level of quality control, we also analyzed associations with age and sex and found that these were generally concordant between the affinity and MS-based proteomics platforms (Supplementary Note 2).

Taken together, we demonstrate the complementarity of MS proteomics with affinity approaches by validating associations that may be driven by epitope effects and by substantially extending the panel of proteins accessible to pQTL studies. We report new pQTLs of biomedical relevance and provide important insights at the peptide level regarding the genetic architecture of previously reported pQTLs. We also show that MS-based and affinity-based methods are complementary when it comes to interpreting pQTLs in the presence of multiple isoforms, because affinity methods discriminate proteins at the level of the folded protein, whereas MS methods work on the peptide level at a higher 'resolution' by mapping multiple peptides to different parts of the protein.

## Online content

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

## Methods

### Ethics

The original Tarkin study was reviewed and approved by the Institute of Regenerative and Cellular Medicine (IRCM) Institutional Review Board (IRB) and the Massachusetts General Brigham (MGB) IRB. Participants provided written informed consent to take part in the study. The Weill Cornell Medicine–Qatar (WCMQ) IRB determined that use of the Tarkin data for the present project did not meet the definition of human research for this study (IRB document no. HRP-532). The QMDiab study was approved by the institutional research boards of WCMQ under protocol no. 2011-0012 and Hamad Medical Corporation under protocol no. 11131/11, and complies with all relevant ethical regulations. For forthgoing work with the study nonhuman participant research, determination was obtained. The study design and conduct complied with all relevant regulations regarding the use of human study participants and was conducted in accordance with the criteria set by the Declaration of Helsinki.

### Cohorts

The samples used in the Tarkin study were obtained from the MGB Biobank. Joint phenotype and genotype data were available for 1,260 samples, comprising 662 women and 598 men with an age range of 23–99 years (median 70 years, mean 67.2 years). Of the participants 1,057 self-reported as white. This subset of MGB samples together with its deep omics characterization is referred to as the 'Tarkin study' in this paper[33]. For replication, a total of 345 previously unthawed, citrate blood plasma samples from participants of the Qatar Metabolomics study of Diabetes (QMDiab), including women and men of predominantly Arab, Indian and Filipino ancestries, with and without diabetes in an age range of 18–80 years were assayed using the Proteograph platform (Seer)[23,34,35].

### Genotyping

Imputed genotype data for 1,980 samples of the Tarkin study was received on a per-chromosome basis in vcf format (build 37, imputed using Minimac3, no insertions and/or deletions (indels)). The genotype data were filtered for biallelic variants and variant names were standardized using bcftools (v.1.16), converted to the plink format and filtered using PLINK2 (v.2.00a5LM) with the options --geno 0.1 --mac 10 --maf 0.05 --hwe 1E-15. For 1,260 samples, proteomics data were available. These samples were merged into a single genotype file and further filtered using PLINK2 with the options --maf 0.05 --hwe 1E-6 --geno 0.02, leaving data for 5,461,287 genetic variants. The first ten genetic principal components were then computed using PLINK2 with the --pca option. QMDiab samples were genotyped using the Illumina Omni 2.5 array (v.8) and imputed using the SHAPEIT software with 1000 Genomes (phase 3) haplotypes. Genotyping data were available for 325 of the 345 samples with proteomics data.

### Proteomics

The workflow used for the proteomics analyses of the Tarkin and the QMDiab samples were essentially identical and have been previously described in detail[23]. Briefly, plasma samples were prepared using the Proteograph workflow[24,25] (Seer) to generate purified peptides that were then analyzed using a dia-PASEF method[36] on a timsTOF Pro 2 mass spectrometer (Bruker Daltonics). Each study was conducted at independent times using two mass spectrometers. One mass spectrometer coincidentally was reused in both studies. DIA-NN (1.8.1)[37] was used to derive peptide and protein intensities. A library-free search based on UniProt UP000005640_9606 was used, processing the data a second time using the match between runs (MBR) option. Two additional libraries were created, one excluding common (minor allele frequency (MAF) >10%) PAV-containing peptides and one injecting the alternative alleles into the reference protein sequences. These libraries were referred to as the PAV-exclusive library and the PAV-inclusive library, respectively.

Details of this library generation process have been described elsewhere[23]. DIA-NN's normalized intensities (PG.Normalised) were used as protein readout. Some 5,753 unique protein groups were quantified, 4,109 were detected in at least 20% of the samples and 1,980 had <20% of missing values. Wherever a protein was detected at this level in more than one nanoparticle run, the one with the largest sum of protein intensities was retained. Protein levels were then log(scaled), residualized using age, sex and the ten first genotype principal components and finally inverse-normal scaled.

### Genome-wide association

Genetic associations of 1,980 residualized and inverse-normal scaled protein levels with 5,461,287 genetic variants in 1,260 samples were evaluated using linear models (PLINK v.1.90b7.1, option --linear). Missing data points were excluded. For six proteins, the residualization did not entirely remove association with these confounders. Three of these proteins presented with inflated GWAS statistics and the corresponding pQTLs were removed from the analysis (UniProt V9GYE7, B1AKG0 and O15230). For every protein, the strongest association reaching genome-wide significance ($P < 5 \times 10^{-8}$) within a ±10-Mb window was retained. Lead pQTLs were then clumped into loci using an LD cutoff of $r^2 = 0.9$ and $r^2 = 0.6$, which led to the identification of 308 and 295 loci, respectively (Supplementary Table 2). Conditionally independent variants were identified by iteratively conditioning on the previously identified genetic variants until no further variant reaching an ad-hoc significance level of $P < 10^{-6}$ was found.

### Replication

Replication was attempted using data from 325 samples of the QMDiab study that had joint genotype and proteomics information. Power to replicate 80% of the pQTLs was determined as the 80% quantile of the $P$ values obtained from 1,000 random samples from the Tarkin dataset using the number of samples available in the QMDiab for the respective genotype–protein pair. Colocalization analysis using the coloc software[38] was conducted using summary statistics for all variants shared between Tarkin and QMDiab within a ±1-Mb window around the respective lead variant. A pQTL was considered replicated if (1) coloc suggested H4 (presence of a shared genetic signal) as the most likely hypothesis and (2) the joint $P$ value computed for the strongest association on a shared variant between Tarkin and QMDiab reached a genome-wide and proteome-wide significance level of $P < 5 \times 10^{-8}/1,980$. Concordance of directionality was verified. We also implemented an alternative replication strategy as follows: for every pQTL with matching protein data in QMDiab ($n = 328$), we identified all proxies of the lead SNP ($r^2 \geq 0.1$) using LD information from Tarkin. We then identified all proxies of these variants in QMDiab ($r^2 \geq 0.1$) using LD information from QMDiab. We considered a pQTL replicated when the strongest association of these proxies had a $P < 0.05/328$ (Supplementary Table 2).

### Overlap with previous Olink and SOMAscan pQTLs

Summary statistics from the UKB-PPP Olink study[18] and deCODE SOMAscan study[17] were downloaded from the respective sites. Associations were reported for 4,660 proteins by deCODE and for 2,908 proteins by UKB-PPP. All protein associations on variants that matched one of the Proteograph pQTLs were retrieved. Matching at the protein level was conducted in three steps: (1) identical UniProt ID between the MS and affinity study; (2) match to one of the UniProt IDs in a protein group, regardless of the isoform version (ignoring the dash-number in the UniProt ID); and (3) match at the level of the protein-coding gene. An association reported by deCODE and Olink was considered as significant at levels of $P < 5 \times 10^{-8}$ and as marginally significant $P < 0.05$ and $P > 5 \times 10^{-8}$. The respective information regarding the level of the match is recorded in the respective Supplementary Tables.

## Evaluation of age and sex associations

Summary statistics for age and sex associations were retrieved from the supplementary Excel files of ref. [17] for deCODE SOMAscan (sheet ST01) and of ref. [18] for UKB-PPP Olink (sheet ST5). Associations with age and sex for the Tarkin study were computed with PLINK using identical datasets and models as for the genome-wide association (option --linear no-snp). Proteins were matched using UniProt identifiers. In rare cases when there were matches to multiple protein groups, the strongest association was retained.

## Evaluation of Olink and SOMAscan *cis*-pQTLs

The *cis*-pQTLs were obtained from the supplementary tables of the respective studies (ST02 from ref. [17] for deCODE SOMAscan and ST10 from ref. [18] for UKB-PPP Olink). The pQTLs were limited to variants that were located on autologous chromosomes, had a suitable replication SNP available in both Tarkin and QMDiab (LD $r^2 > 0.8$) and an MAF > 5% in the Tarkin study. Matching of proteins between platforms was done using UniProt IDs, allowing for matches to protein groups that contained the UniProt ID and matches at the gene level, as described above for the overlap with previous Olink and SOMAscan pQTLs. Ambiguous cases where more than one matching protein group was found were omitted. In cases where protein readouts for multiple nanoparticle runs were available, the one with the highest single number of peptide detections was used. Cases where the number of quantified proteins in Tarkin was <80% were excluded. The SOMAscan platform sometimes uses multiple aptamers. In these cases, the strongest association was retained.

## Annotation of pQTLs

The Open Targets[27] platform (v.22.10) was used via the API to annotate pQTL variants with most likely causal genes, variant effect, overlapping disease GWAS hits and gene expression, splice variant and proteomics QTLs, ordered by increasing *P* value and reported separately for *cis*-QTLs and *trans*-QTLs. The Open Targets look-up comprised both, same variant across stored datasets and reference to GWAS regional lead signals using LD, limited to LD $r^2 > 0.7$ between the pQTL and GWAS lead signal. Hyperlinks to individual Open Targets pages with more detailed information, including references for the disease associations, are provided in Supplementary Table 2. A comprehensive list of all queried studies can be found in the Open Targets release notes (https://genetics-docs.opentargets.org/release-notes). Protein epitope changing variants were identified using Ensembl[32].

## Definition of the MSPA score

We define the MSPA score of a pQTL as follows:

$$\text{MSPA score} = \sum_{i=1}^{k_{\text{pep}}} \frac{n_i}{n_{\text{tot}}} \times \delta \left( c_i^{\text{upper}} \times c_i^{\text{lower}} > 0 \right)$$

where $k_{\text{pep}}$ is the number of different peptides that have been detected for a given pQTL protein, $n_i$ is the number of samples in which a peptide *i* from the given protein has been detected, $n_{\text{tot}} = \sum_{i=1}^{k_{\text{pep}}} n_i$ is the total number of individual peptide detections, $c_i^{\text{upper}}$ and $c_i^{\text{lower}}$ are the upper and lower 99% bound of the CI for the effect size of the genetic association of the pQTL variant with peptide *i*, and $\delta$ (condition) is a function that takes a value of 1 if the condition in its argument is true and 0 otherwise. We further computed MSPA scores based on genetic colocalization (coloc-MSPA) between Tarkin and UKB-PPP, where the score is defined as the weighted fraction of a given hypothesis being considered most likely by coloc[38]. The coloc-MSPA score for a shared genetic signal (hypothesis H4) is thus computed using the above formula with $\delta$ (H4 most likely) and similar for a different genetic signal (hypothesis H3).

## Power analysis

We determined power to replicate pQTLs from the deCODE and UKB-PPP studies using an F test. For the computation of the noncentrality parameter (NCP), we used the effect size ($\beta_{\text{aff}}$) from the UKB-PPP or deCODE pQTLs, whereas the s.e. (s.e._MS) was taken from Tarkin protein associations at the respective loci to account for the variability of the MS measurements, that is, NCP = $(\beta_{\text{aff}}/\text{s.e.}_{\text{MS}})^2$. Power was then computed as $1 - P(F < F_{\text{crit}}, \text{d.f.}_1, \text{d.f.}_2, \text{NCP})$ where *P* represents the cumulative probability of the *F* distribution with the NCP. The degree of freedom d.f._1 was set to the number of samples in Tarkin with valid data for the respective pQTL − 2 and d.f._2 was set to 2 (slope and offset). $F_{\text{crit}}$ was determined as the *F* value corresponding to a significance level of $P < 0.05/319$ and $P < 0.05/392$ to account for the number of tested pQTLs for deCODE and UKB-PPP, respectively (Fig. 3 and Supplementary Tables 9 and 10).

## Peptide mapping

Protein quantification was performed using algorithms implemented in DIA-NN[37]. Although DIA-NN applies calibrated methods and in particular accounts for false discovery rates (FDRs) in the mapping of peptides to proteins, the number of peptides mapped to a given protein and their coverage of the protein can vary and are a measure for the robustness of the protein identification and its quantification. We therefore aligned all peptides to their respective protein amino acid sequences (R Biostrings, v.2.60.2) and plotted their coverage against their sample-wise detection rates (Supplementary Fig. 14 and Supplementary Data 6). The average (median) number of peptides detected per sample was 9.6 (5.6) for Tarkin and 6.4 (2.9) for QMDiab (Supplementary Fig. 15g). To examine the overall between-study variation of the protein and peptide quantifications, we compared the mean intensities and their s.d. values between Tarkin and QMDiab. The average intensities of the 1,980 analyzed proteins correlated with $r^2 = 0.78$ (slope = 0.84) and their s.d. values correlated with $r^2 = 0.69$ (slope = 0.71). The average intensities of the individual peptides that map to these proteins correlate with $r^2 = 0.61$ (slope = 0.76) (Supplementary Fig. 15a–f and Supplementary Table 14).

## Protein- and peptide-level associations for all nanoparticle runs

To reduce the multiple-testing burden, we analyzed only a single nanoparticle run for every protein in the main GWAS, selecting the one with the highest intensity at <20% missingness. However, in many cases, a same protein or peptide can be detected during multiple nanoparticle runs. Their associations can provide additional insights and support for the validity of a given pQTL. We therefore computed the summary statistics of all pQTL variants with all proteins and peptides in all nanoparticle runs using identical methods as for the main GWAS (Supplementary Table 15).

## Missingness analysis

We conducted a GWAS on missingness using PLINK (v.1.90b7.1, option --model fisher) including all proteins in all nanoparticle runs that had a missingness <95%. For each variant association, we retained the lower of the two *P* values obtained from Fisher's exact test with a dominant and a recessive genetic model. To identify the lead variants across multiple nanoparticle runs, we clumped the associations into loci, that is, for every protein, the strongest association reaching an ad-hoc significance level of $P < 5 \times 10^{-12}$ within a ±10-Mb window was retained. Lead pQTLs were then clumped into loci using an LD cutoff of $r^2 = 0.9$.

## Reporting summary

Further information on research design is available in the Nature Portfolio Reporting Summary linked to this article.

## Data availability

The MS-proteomics data for the Tarkin study have been deposited with ProteomeXchange under project accession no. PXD048709. The MS-proteomics data of QMDiab are available on ProteomeXchange with accession no. PXD042852. Summary statistics of this GWAS were deposited in the GWAS catalog with identifiers GCST90570713 to GCST90572692. Consent obtained from the study participants does not allow deposition of genetic information in public databases. Researchers affiliated with a research institution may request access to genetic data on an individual basis from the corresponding authors (K.S., WCMQ, Doha, Qatar for QMDiab and J.A.L.-S. for Tarkin). Access is subject to approval by the respective institutional research boards of WCMQ and the MGB Biobank.

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

## Acknowledgements

We are grateful to all study participants for providing their time and blood. K.S. and F.S. are supported by the Biomedical Research Program at WCMQ, a program funded by the Qatar Foundation. K.S. is also supported by Qatar National Research Fund (grant nos. NPRP11C-0115-180010 and ARG01-0420-230007). J.A.L.-S. is supported by the National Institutes of Health (grant nos. R01HL123915, R01HL155742 and U19AI168643). The statements made herein are solely the responsibility of the authors. We thank G. Venkataraman, A. Alavi and J. Wang for developing software workflows that were used in the proteogenomic analyses in this work.

## Author contributions

Financial support: K.S. and J.A.L.-S. Study design: K.S. and J.A.L.-S. Data analysis: K.S., H.G. and S.B. Provided materials and conducted experiments: Q.C., A.H., K.M., A.D., N.S., G.T., H.S., V.B.D., R.S. and F.S. Manuscript writing: K.S., S.B. and J.A.L.-S. All authors contributed to the interpretation of the results and critically reviewed the manuscript.

## Competing interests

H.G. and S.B. are employees and/or stockholders of Seer R.S. and J.A.L.-S. are scientific advisors to Precion and TruDiagnostic. J.A.L.-S. has a sponsored research agreement with TruDiagnostic and has previously consulted for Cambrian and Ahara. The remaining authors declare no competing interests.

## Additional information

**Correspondence and requests for materials** should be addressed to Karsten Suhre or Jessica A. Lasky-Su.

# Reporting Summary

## Statistics

For all statistical analyses, confirm that the following items are present in the figure legend, table legend, main text, or Methods section.

| n/a | Confirmed | |
|---|---|---|
| ☐ | ☒ | The exact sample size (*n*) for each experimental group/condition, given as a discrete number and unit of measurement |
| ☒ | ☐ | A statement on whether measurements were taken from distinct samples or whether the same sample was measured repeatedly |
| ☐ | ☒ | The statistical test(s) used AND whether they are one- or two-sided<br>*Only common tests should be described solely by name; describe more complex techniques in the Methods section.* |
| ☐ | ☒ | A description of all covariates tested |
| ☐ | ☒ | A description of any assumptions or corrections, such as tests of normality and adjustment for multiple comparisons |
| ☐ | ☒ | A full description of the statistical parameters including central tendency (e.g. means) or other basic estimates (e.g. regression coefficient) AND variation (e.g. standard deviation) or associated estimates of uncertainty (e.g. confidence intervals) |
| ☐ | ☒ | For null hypothesis testing, the test statistic (e.g. *F*, *t*, *r*) with confidence intervals, effect sizes, degrees of freedom and *P* value noted<br>*Give P values as exact values whenever suitable.* |
| ☒ | ☐ | For Bayesian analysis, information on the choice of priors and Markov chain Monte Carlo settings |
| ☒ | ☐ | For hierarchical and complex designs, identification of the appropriate level for tests and full reporting of outcomes |
| ☐ | ☒ | Estimates of effect sizes (e.g. Cohen's *d*, Pearson's *r*), indicating how they were calculated |
| | | *Our web collection on statistics for biologists contains articles on many of the points above.* |

## Software and code

Policy information about availability of computer code

| Data collection | no specific code was used for data collection |
|---|---|
| Data analysis | publicly available software was used for data analysis using simple algorithms for the computation of linear models, pre- and post-processing of data, in particular: bcftools (version 1.16), plink2 (version v2.00a5LM), DIA-NN (1.8.1), Open Targets (version 22.10), R Biostrings (version 2.60.2) |

For manuscripts utilizing custom algorithms or software that are central to the research but not yet described in published literature, software must be made available to editors and reviewers. We strongly encourage code deposition in a community repository (e.g. GitHub). See the Nature Portfolio guidelines for submitting code & software for further information.

## Data

Policy information about availability of data

All manuscripts must include a data availability statement. This statement should provide the following information, where applicable:
- Accession codes, unique identifiers, or web links for publicly available datasets
- A description of any restrictions on data availability
- For clinical datasets or third party data, please ensure that the statement adheres to our policy

The MS-proteomics data for the Tarkin study has been deposited with ProteomeXchange under project accession code PXD048709 and will be made public at time

of publication. The MS-proteomics data of QMDiab is already available on ProteomeXchange with identifier PXD042852. Summary statistics of the GWAS were deposited in the GWAS catalog with identifier GCP001281. Consent obtained from the study participants does not allow deposition of genetic information in public databases. Researchers affiliated with a research institution may request access to genetic data on an individual basis from the corresponding authors (Karsten Suhre, Weill Cornell Medicine - Qatar, Doha, Qatar for QMDiab and Jessica A. Lasky-Su for Tarkin). Access is subject to approval by the respective institutional research boards of Weill Cornell Medicine – Qatar and the Massachusetts General Brigham (MGB) Biobank.

# Research involving human participants, their data, or biological material

Policy information about studies with human participants or human data. See also policy information about sex, gender (identity/presentation), and sexual orientation and race, ethnicity and racism.

| Reporting on sex and gender | we only use the term "sex" in reference to the biological attribute |
|---|---|
| Reporting on race, ethnicity, or other socially relevant groupings | we only use the term "ethnicity" in reference to the underlying genetic diversity of the study participants |
| Population characteristics | no specific population characteristics are mentioned |
| Recruitment | participants were recruited on a random basis |
| Ethics oversight | The original Tarkin study was reviewed and approved by the IRCM IRB and the Mass General Brigham IRB. Participants provided written informed consent to take part in the study. The WCMQ IRB determined that use of the Tarkin data for the present project does not meet the definition of human research for this study (IRB document HRP-532). The QMDiab study was approved by the institutional research boards of Weill Cornell Medicine – Qatar under protocol #2011-0012 and of Hamad Medical Corporation under protocol #11131/11 and complies with all relevant ethical regulations. For forthgoing work with the study a non-human subject research determination was obtained. The study design and conduct complied with all relevant regulations regarding the use of human study participants and was conducted in accordance with the criteria set by the Declaration of Helsinki. |

Note that full information on the approval of the study protocol must also be provided in the manuscript.

# Field-specific reporting

Please select the one below that is the best fit for your research. If you are not sure, read the appropriate sections before making your selection.

☒ Life sciences ☐ Behavioural & social sciences ☐ Ecological, evolutionary & environmental sciences

For a reference copy of the document with all sections, see nature.com/documents/nr-reporting-summary-flat.pdf

# Life sciences study design

All studies must disclose on these points even when the disclosure is negative.

| Sample size | In GWAS, sample size determines the power to detect genetic associations. Previous GWAS with MS proteomics had significantly smaller sample size and/or smaller protein panels. |
|---|---|
| Data exclusions | Exclusion criteria were the missingness of the protein (>20%) and low minor allele frequency for the genomics data (<5%). |
| Replication | Full replication was provided using the QMDiab study, requiring Bonferroni significance. |
| Randomization | Does not apply |
| Blinding | Does not apply |

# Reporting for specific materials, systems and methods

We require information from authors about some types of materials, experimental systems and methods used in many studies. Here, indicate whether each material, system or method listed is relevant to your study. If you are not sure if a list item applies to your research, read the appropriate section before selecting a response.

## Materials & experimental systems

| n/a | Involved in the study |
|-----|----------------------|
| ☒ | Antibodies |
| ☒ | Eukaryotic cell lines |
| ☒ | Palaeontology and archaeology |
| ☒ | Animals and other organisms |
| ☒ | Clinical data |
| ☒ | Dual use research of concern |
| ☒ | Plants |

## Methods

| n/a | Involved in the study |
|-----|----------------------|
| ☒ | ChIP-seq |
| ☒ | Flow cytometry |
| ☒ | MRI-based neuroimaging |

## Plants

| Seed stocks | N/A |
|-------------|-----|
| Novel plant genotypes | N/A |
| Authentication | N/A |

