## [Peer Review File · Nature Genetics]

A genome-wide association study of mass spectrometry proteomics using a nanoparticle enrichment platform

Corresponding Author: Professor Karsten Suhre

A version of this paper was originally rejected for publication by Nature Genetics, however that decision was reconsidered after appeal by the authors.

Version 0:

Decision Letter:

8th July 2024

Dear Karsten,

Your Letter "A genome-wide association study of mass spectrometry proteomics using the Seer Proteograph platform" has been seen by three referees. You will see from their comments below that, while they find your work of potential interest, they have raised substantial concerns that must be addressed. In light of these comments, we cannot accept the manuscript for publication at this time, but we would be interested in considering a suitably revised version that addresses the referees' concerns.

We hope you will find the referees' comments useful as you decide how to proceed. If you wish to submit a substantially revised manuscript, please bear in mind that we will be reluctant to approach the referees again in the absence of major revisions.

To guide the scope of the revisions, the editors discuss the referee reports in detail within the team, including with the chief editor, with a view to identifying key priorities that should be addressed in revision, and sometimes overruling referee requests that are deemed beyond the scope of the current study. In this case, we ask that you address all technical queries related to the pQTL analyses and their interpretation, further integrate the pQTL results with disease association data to illustrate the value of the Seer platform, and ideally increase the replication sample sizes. We hope you will find this prioritized set of referee points to be useful when revising your study. Please do not hesitate to get in touch if you would like to discuss these issues further.

If you choose to revise your manuscript taking into account all reviewer and editor comments, please highlight all changes in the manuscript text file. At this stage, we will need you to upload a copy of the manuscript in MS Word .docx or similar editable format.

*2) If you have not done so already, please begin to revise your manuscript so that it conforms to our Letter format instructions, available [here](http://www.nature.com/ng/authors/article_types/index.html). Refer also to any guidelines provided in this letter.

*3) Include a revised version of any required Reporting Summary: <https://www.nature.com/documents/nr-reporting-summary.pdf>

Please be aware of our [guidelines](https://www.nature.com/nature-research/editorial-policies/image-integrity) on digital image standards.

Link Redacted

If you wish to submit a suitably revised manuscript, we hope to receive it within 3-6 months. If you cannot send it within this time, please let us know. We will be happy to consider your revision so long as nothing similar has been accepted for publication at Nature Genetics or published elsewhere. Should your manuscript be substantially delayed without notifying us in advance and your article is eventually published, the received date would be that of the revised, not the original, version.

Nature Genetics is committed to improving transparency in authorship. As part of our efforts in this direction, we are now requesting that all authors identified as 'corresponding author' on published papers create and link their Open Researcher and Contributor Identifier (ORCID) with their account on the Manuscript Tracking System (MTS), prior to acceptance. ORCID helps the scientific community achieve unambiguous attribution of all scholarly contributions. You can create and link your ORCID from the home page of the MTS by clicking on 'Modify my Springer Nature account'. For more information, please visit www.springernature.com/orcid.

Thank you for the opportunity to review your work.

Sincerely,
Kyle

Kyle Vogan, PhD
Senior Editor
Nature Genetics
<https://orcid.org/0000-0001-9565-9665>

Referee expertise:

Referee #1: Genetics, proteomics, molecular QTLs

Referee #2: Genetics, proteomics, molecular QTLs

Referee #3: Genetics, proteomics, molecular QTLs

Reviewers' Comments:

Reviewer #1:
Remarks to the Author:

Review Nature Genetics, A genome-wide association study of mass spectrometry proteomics using the Seer Proteograph platform

Suhre et al. performed an interesting investigation of protein quantitative trait loci based on the Seer multiplex proteomics platform. pQTL studies have shown to be of importance in drug discovery as well as understanding of molecular mechanisms associated with disease, as shown by the SCALLOP consortium and by UK biobank PPP. Most pQTL studies have been performed based on the Olink or Somascan platforms, which are affinity methods that come with the well-known challenge of identifying pseudo-pQTLs, i.e. pQTLs that reflect epitope differences rather than true expression differences.

Comments

The study populations in the present work are ethnically diverse with supposedly quite different lifestyles and genetic backgrounds. Yet, no protein or genetic data across ethnicities are presented. Please provide a plot of population stratification, i.e. the PCs, so that the reader can get a sense for the population structure. Were some of the peptides only

observed in people with certain genetic backgrounds? If so, please provide that information, even though the PC adjustment may take care of most systematic population differences.

Figure 2b shows the beta levels of identified pQTL across Tarkin and QMdiab. The within platform replication seems lower than previously published Olink studies, which generally show very consistent betas. I lack a section in which the authors explain or speculate on these differences, for example in the discussion section stating other limitations.

Line 80, please assign identified pQTL as cis- and trans- according to common practice and report the number of cis- and trans-pQTL separately.

Line 85, why only attempt to replicate pQTL based on a power calculation? Statistical power should be done a priori rather than looking at post-hoc power. A straight measure of replication across the datasets would be more informative, despite the sample size differences. If the sample size for replication is small, then the alpha value should be set according to the sample size.

Line 95, Some peptides would capture a PAV, and these were removed from the MSPA score (line 128). Does this mean that the authors went back to the genotyping data to check if the individual carried the PAV or not, and then removed the peptide measurement? Or does it mean that if there was a potential PAV for a certain peptide, it was removed from all individuals in the study? If the latter, was a frequency cutoff used?

Line 128-139, what is the advantage of using a combined peptide score to capture total protein expression vs. conducting a genetic association test for each peptide and then meta-analyses or else combine the results?

Minor comments:

Line 45, "The largest studies to date", suggest change to "The largest studies published to date"

Line 49, suggest to add reference on MR in drug discovery, Folkersen et al. Nat. Metab. 2020

Line 50, there are already some pQTLs known to be caused by epitope effects, e.g. GDF-15, adiponectin, etc. I think the intro could be improved if some specific proteins were exemplified.

Line 53, suggest to add additional references regarding epitope effects as these have been well known for many years and described in literature

Line 59, suggest to remove the word "smaller-scale" from "A few smaller-scale MS-based GWAS..." since the expression degrades previous work in this space, and the fact that at least one of the studies used a larger sample than the authors describe, albeit fewer proteins.

Line 65, suggest to remove the unnecessary word "unbiased", with the motivation that no proteomics method is "unbiased", for example MS is typically biased towards medium and high abundance proteins.

Line 294, LD clumping based on $R^2 < 0.1$ may be too liberal for pQTL studies with often strong effect sizes. A cutoff at 0.01 or 0.001 to identify truly independent sentinel variants have been used in other studies.

Reviewer #2:

Remarks to the Author:

Review Comments

The study by Suhre et al. presents a GWAS on plasma protein levels measured with a recently developed mass spectrometry platform in a small-to-moderately sized cohort, demonstrating a number of interesting technical insights in the mapping of protein quantitative trait loci. Most importantly the authors raise (again) and also in line with other work (e.g., Niu et al. MedRxiv 2022) concerns about a substantial number of highly utilized cis-pQTLs from much larger plasma proteomics studies using affinity reagents. For example, they provide evidence that about a third of the most strongly associated cis-pQTLs are potential artefacts with widespread implications for the (over)use as instruments in downstream applications such as Mendelian randomization.

While the study provides important insights for the discovery of pQTLs, that might still benefit from addressing the concerns listed below, it rather reads like a technical report with no insights about the biological (e.g., no presentation on trans-pQTLs or mechanisms underlying cis-pQTL associations) and possibly clinical relevance of 'newly' identified pQTLs or how these insights might be leveraged by the community to improve downstream tasks. It had already been demonstrated that molecular QTLs are somewhat depleted for associations with clinically relevant phenotypes and hence it needs to be demonstrated why these more sophisticated but yet not scalable method is worth investing further into.

1) Please omit mentioning of vendors in the title of scientific manuscripts.

2) Picky, but the use of 'drug discovery' is highly inflated and since the authors do not present evidence how their data can inform drug target discovery, they should better tone such reference down.

3) Why only 2k proteins present in >80% of samples? How does the number of identified pQTLs compare to previous studies? I might be wrong, but studies of similar size have identified much more pQTLs previously for a comparable or even lower number of proteins.

4) The sentence "it should be noted that SOMAscan and Olink pQTLs represent genetic associations with protein binding affinity, rather than protein expression" is not quite true. It should be acknowledged that in most cases there is a reproducible link between the number of reagents that bind to their targets and the abundance of the target. What the authors refer to is the very specific case, in which we have genetic association that may lead to a dissociation of this dependency.

5) Please avoid terms like 'unbiased' it is virtually impossible to do anything 'unbiased', as this would assume that we would have complete knowledge of the world. For example, in the current study, it is unclear whether there is any differential binding of protein targets in plasma that are not picked up by nanoparticles or peptides that can be mapped in DIA-NN (which relies again on our knowledge of what protein isoforms exist) So, I encourage the authors to not only be more conscious about the language used, but also to discuss these limitations.

6) Why age and sex associations?

7) Can the authors please elaborate more on potential sources of non-replication? To what extent did different LD-structures contribute to the rather moderate level of replication, e.g., selecting a tag SNP in Tarkin which is not in LD with the underlying causal variant in the Arab cohort?

8) The stratification of cis-pQTLs on the peptide levels is important, but currently lacks a systematic assessment. How often are pQTLs supported by only a subset of matching peptides and to what extent can this be attributed to peptides matching to multiple proteins? I appreciate that the MSPA score tries to get at this, but it does not account for the fact the peptides may map to multiple different protein targets and might further be biased if soluble and complete forms of the same protein are clumped together (affinity reagents might better distinguish here).

9) How were novel/reported pQTLs classified? Given large variations in ancestral background a regional definition seems to be most appropriate, by which a 1-Mb window surrounding the sentinel variant has not been reported with any protein measurement previously. To what extent are novel pQTLs explained by proteins not targeted before?

10) Given the very limited sample size, the authors should rather stay away from quantitative assessments like 'an expected novel discovery rate when using the Proteograph platform of about one in three compared to existing affinity pQTLs'. The same applies to the inference of how many epitope effects there possibly are with Olink/SomaLogic, given that the moderate sample size of this study will almost certainly bias the results. It would be much more insightful to expand these analyses given that there are already substantial concerns about the most strongly associated cis-pQTLs. Further, the MSPA generated does not distinguish either for the differential abundance of different protein isoforms or soluble fragments.

11) Presenting technical data on reproducibility of the nanoparticle enrichment and MS measurements would be important.

12) The age and sex association are somewhat odd, since it is not really known what the ground truth would be. If anything, such analysis should be used to establish any potential systematic differences across platforms, e.g., is there anything specific to proteins seen to be associated with age with one but not the other platforms. It is also odd to place this analysis within the reporting of genetic work.

13) The section "One third of affinity proteomics pQTLs are potentially affected by epitope effects." is written in a very odd style, and too often the authors refer to massive sets of simple plots instead of providing some tangible insights and subsequent justification. For the overlap between pQTLs discovered in SomaLogic or Olink with Seer, it is also unclear to what extent the true underlying causal variant had been selected. While the trans-ancestral design is desirable, it also complicates these matters since we would need to be even more sure about the true underlying causal variant.

14) The reasoning not to demonstrate the biological relevance of 'newly' identified pQTLs is pointless and would have been important to distinguish this paper from a merely technical report.

15) I don't understand this sentence 'The WCMQ IRB determined that use of the Tarkin data for the present project does not meet the definition of human research for this study (IRB document HRP-532', why is this study not human research?

16) The ancestral assignment in the Tarkin study is odd. What is meant by 'white'? Why have participants not been assigned to ancestral groups using external PC loadings? This might introduce residual population stratification and hence also explain inflated GWAS stats for some proteins, which is really unusual for such a small data set.

17) Using an r^2 of 0.9 to clump sentinels across proteins is too stringent and other thresholds, possibly down to 0.6 should be explored to define loci.

18) Please clarify whether ANML or non-AMNL results have been used to compare to deCODE results.

19) Many of the inferences drawn across platforms implicitly assume that each of them measures the very same 'thing', which is unlikely to be true, and correlation analysis across different proteomic technologies would be needed to establish this. For example, it had already been suggested that different technologies measure different isoforms of the very same protein and hence comparing those must lead to different results.

20) The Manhattan-like plot looks odd and should be replaced with a beta vs. MAF plot or similar, given that most associations are cis-pQTLs (so the location in the genome doesn't really matter).

21) To what extent are pQTLs discovered with MS also prone to technological artefacts? For example, the authors previously highlighted the role of variants mapping to sites changing tryptic digestion.

22) Figure 5 is the very central message of this work, but looks really odd. The massive overlay of the sigmoid curve and some other highlighted section appears superficial, simply categorizing the MSPA would be more informative. Also, how would this figure look like using z-scores instead of ranks, and to what extent has allele frequencies been taken into account? For example, a very common missense variant introducing an epitope effect will have a much stronger p-value compared to less frequent variants.

23) Figure 7 is somewhat hard to interpret, since the comparator is missing, since PAV-peptides were excluded from the study.

24) What distinguishes proteins detected with Seer but not with SomaLogic and Olink? Do the authors observe any differences between proteins commonly detected by each of the methods compared to those measured with only one of the affinity reagents?

25) The study centres a lot around cis-pQTLs, but also identifies rather pleiotropic trans-pQTLs frequently linked to blood cell counts. How did the authors account for such residual findings? What quality were samples of?

26) Finally, to what extent can the authors provide guidance on how to use their results to refine pQTLs studies in much larger affinity-based studies? For example, to what extent are secondary signals at cis-loci valid proxies, even if the main signal might be an artefact.

27) Minor, but phenoscanner is outdated and no longer maintained. SNPs should rather be queried using the OpenGWAS or OpenTargets portals, the latter would also allow to establish overlap of pQTLs with GWAS credible sets.

Reviewer #3:
Remarks to the Author:

See attached pdf.

Version 1:

Decision Letter:

24th September 2024

Dear Karsten,

Your revised Letter entitled "A genome-wide association study of mass spectrometry proteomics using a nanoparticle enrichment platform" has been seen by the original referees, whose comments are provided below. In the light of their advice, we have decided that we cannot offer to publish your manuscript in Nature Genetics.

In particular, while the referees find that the manuscript has improved in revision, Reviewers #2 and #3 have ongoing concerns about the level of advance your findings represent over earlier work and the strength of the novel conclusions that can be drawn from the analyses. We are persuaded that these reservations are sufficiently important as to preclude publication of this study in Nature Genetics.

Although we regret that we cannot offer to publish your paper given these reviews, we have discussed your manuscript and the reviewers' comments with our colleagues at Nature Communications, and they have agreed that they would send an appropriately revised version back to the original reviewers if you were to transfer the manuscript. Should you wish to have your revised paper considered by Nature Communications, please use the link to the SpringerNature manuscript transfer service in the footnote, once the revision is ready, and include a point-by-point response to the reviewers' concerns.

Your handling editor at Nature Communications would be Dr. Margot Brandt (margot.brandt@us.nature.com). If there is anything you would like to discuss before transferring the paper and its reviews, please don't hesitate to contact her by e-mail.

Please note that Nature Communications is a fully open access journal. For information about article processing charges, open access funding, and advice and support from Springer Nature, please consult the Nature Communications Open Access page (<https://www.nature.com/ncomms/about/open-access>).

I am sorry that we cannot be more positive on this occasion, but we hope that you will find the referees' comments helpful when preparing your paper for submission elsewhere.

Sincerely,
Kyle

Kyle Vogan, PhD
Senior Editor
Nature Genetics
<https://orcid.org/0000-0001-9565-9665>

Referee expertise:

Referee #1: Genetics, proteomics, molecular QTLs

Referee #2: Genetics, proteomics, molecular QTLs

Referee #3: Genetics, proteomics, molecular QTLs

Reviewers' Comments:

Reviewer #1 (Remarks to the Author):

The authors have made several significant improvements to the manuscript, including updating the analyses flow and adding a meta-analysis based pQTL discovery approach. The authors have furthermore elegantly addressed the major and minor points I raised in the previous round of reviews. I have no additional questions or concerns.

Reviewer #2 (Remarks to the Author):

Reviewer comments

The paper by Suhre et al. clearly improved during review and raises an important point about the potential artificial nature of pQTLs based on affinity reagents. However, the paper still falls short demonstrating the overall relevance, including biological implications (see comments below). If this paper is kept as a letter, and emphasis is on a technical aspect, I am not sure what it delivers over and above preceding publications by the authors and others. There is still also a tendency of a somewhat biased comparison with affinity reagents. That is, little is done to understand to what extent the more than 50% of discovered but not investigated proteins suffer from similar 'epitope-like' effects due to a change in affinity to the used nanoparticles (see comments).

Rebuttal (reviewer 2)

General remark. The colocalization strategy to establish trans-ancestral sharedness of genetic signals is somewhat odd, in a sense that colocalization is highly reliant on a conserved LD-backbone across cohorts. While it resolves, to some extent, the issue of missing SNP overlap, it will fail if haplotypes are fragmented across ancestries. An alternative, less sophisticated but hopefully more stable strategy, would be to compute all proxies (say $r^2 \geq 0.1$) for each lead pQTL from two different studies, and quantify the overlap of SNPs using in-study LD for each. One may call 'replicated' pQTLs those that share one or more SNP.

Point 11. While I agree on the ability to trace MS performance via protein content from different nanoparticles, the effect of efficacy of the nanoparticles on the type and abundance of assayed proteins is not well captured, but important for our understanding, including false-negative findings. For example, while the authors rightly flag non replicating results for strong cis-pQTLs based on affinity reagents, I find it equally likely that some associations are not replicated as expected due to poor or very variable efficacy of the nanoparticles to extract a given protein. One might even speculate that PAVs also affect binding to nanoparticles and that the issues claimed to be resolved here still persist.

Point 14. I like the additional information on the biological relevance of identified pQTLs a lot but wonder whether the current presentation is either too simplistic or needs toning down in wording. For example, several trans-pQTL findings are highlighted, but we don't know whether these findings are the result of horizontal pleiotropy by which the genetic variant affects protein and disease independently of each other, or whether, as suggested by the authors, proteins indeed mediate disease associations. In particular the reference for drug target discovery is exaggerated.

Also, how is an association with FUCA2 a cis-pQTL for FUCA1, isn't this more of an example of poor peptide mapping in the DIA-NN workflow? The link to AD is vastly exaggerated since there is no association with AD (despite well-powered studies) at this locus. The examples would benefit from expert review.

Point 24 The separation of proteins across platforms is interesting, but can the authors please clarify, why cytokines should be underrepresented by the Seer technology?

Paper

1) Picky, but 'drug discovery' should be replaced by 'drug target discovery' and even from this point it is a long way to go, since in GWAS we cannot easily (or at all) establish the exact disease mechanisms or judge whether intervening on the target will reverse the diseases. More humble language should be used throughout, in particular since the authors present at best very weak evidence for the biological relevance of pQTLs.

2) Second sentence in the abstract: I am a bit tired of people claiming MS-based proteomics being 'unbiased', there are at least two steps that will make this platform selective: 1) the affinity of proteins to certain nanoparticles (some reliably measured proteins with affinity proteomics are not at all present in the data presented), and 2) the behavior and number of peptides measured in the MS to later on 'infer' protein abundances from the original plasma sample. Those limitations should be clearly highlighted. There is a reason why affinity-based assays are still the technique of choice for most clinical chemistry assays used in clinical routine.

3) Introduction: The efforts by deCODE and UKB-PPP are certainly stunning, but this neglects a large body of very influential studies published beforehand. Instead of the sample size or protein coverage, the authors would be better off clearly demonstrating the usefulness of such studies. Why should we invest in even more and more expensive techniques? In particular, since there is a growing body of evidence that molecular QTLs are actually depleted for phenotypic consequences.

4) Also why do the authors claim that MS is 'immune' against epitope effects. It is great to have multiple peptides, in case they uniquely map to a protein, but if the PAV changes affinity to the nanoparticles a similar thing will happen. This effect is currently somewhat ignored in the presented data, given that only proteins with >80% valid values are evaluated.

5) Given the inherent variation in day-to-day and even more so lab-to-lab performance of MS instruments, I wonder to what extent the low replication rate is also explained by missing some key peptides for proteins of interest. Can the authors at least confirm that a similar type and number of peptides was found between both studies? This would be in particular important to investigate for findings such as ITLN1.

6) Does the newly conducted MA that now complements the MSPA score take differing peptide intensities of the same protein target into account?

7) The construction of the MSPA still appears a bit odd, maybe flawed statistically. Controlling each effect estimate for individual peptides at a one percent level will not ensure a 1% level for the overall protein quantification. However, a general pragmatic approach is acceptable, but the authors may want to avoid any reference to statistical significance. They should also provide a criterion at which point the MSPA starts giving reliable results, i.e., I assume that a minimum number of peptides should be requested.

8) The new SPINK5 example is possibly the most appealing one added during review, but I am wondering whether this also points to a potential flaw in the analysis of pQTLs using mass spec data, if certain peptides are omitted from quantification, as they may introduce peptide isoforms that cannot be mapped to protein groups later on. It would also be important to provide a locuszoom plot incorporating the sQTL information from GTEx in skin.

9) Figure 3: Please remove the suspected curve or provide a sound statistical fitting of a model. It is also completely unclear to me, how the authors infer 'power to replicate' from this figure. Power to replicate will depend on effect size and frequency of the allele (plus measurement certainty) and can be computed for each of the pQTLs tested.

10) There are strong inferences drawn based on a set of as few as 46 pQTLs without examining any bias in terms of what proteins those are (likely high abundant and actively secreted) and hence any reference to 'gold-standard' should be avoided. Some of the inference is also circular, if the very same variant had been selected for SomaScan and Olink it is obvious that effect estimates are highly concordant, otherwise those variants were not selected in the first place.

11) It is unclear to what extent the OpenTargets look-up of pQTLs does represent a tangible phenotypic follow-up. For examples, it is unclear from the methods whether the look-up comprised the same variant across stored data sets, or does indeed refer to the table listing GWAS regional lead signals, including an estimation of the LD between the pQTL and GWAS lead signal. If the latter, what cut-off was used to declare some level of colocalization?

Reviewer #3 (Remarks to the Author):

The manuscript is significantly improved following the revisions. However, I have some remaining concerns.

1) The mapping of pQTLs onto disease and complex trait GWAS partially addresses a point I made in my previous review. Table 1 provides details of how the newly described pQTLs link to disease/complex trait GWAS associations. In their response, the authors report that they did not find any enrichment of either true pQTLs or likely epitope effect pQTL signals with respect to GWAS hits.

Despite the revisions, I did not feel much clearer as a reader how these new data should re-shape our thinking of the functional genetics underlying complex diseases and traits. The authors estimate that approximately one third of testable previously reported pQTLs may arise from epitope effects. However, it was unclear how this translates into revising understanding of the mechanisms underpinning disease GWAS associations. For example, how many protein-disease associations identified by previous pQTL work are now in doubt? Are there important Mendelian randomisation analyses that gave spurious positive results based on using pQTLs resulting from epitope effects? Perhaps there were instances where previous pQTL-based MR gave unexpected results discordant from other observational associations (e.g. associations of a protein with incident disease risk) that can now be explained by epitope effects? I appreciate there may be some technical difficulties in quantifying the extent to which SNP-protein-disease links need to be revised because the GWAS Catalog has lumped diseases together with other traits. Nevertheless, this could be done, and I would suggest that a disease-focused analysis would be most valuable given the authors pitch around drug discovery (i.e. ignore anthropomorphic and more esoteric traits).

A very tractable starting point would be the list of pQTLs in Fig S13 (described by the authors as "high ranking" pQTLs based on protein measurements using Somalogic/Olink but with low MPSA scores). How many of these are also disease GWAS hits and how does the new data revise our previous understanding of the molecular mechanisms underlying disease risk at these loci?

2) "a trans-pQTL at the CFH locus that implicates the TNFRSF1A modulator BRE in age-related macular degeneration and IgA nephropathy"

CFH (complement factor H) is a highly pleiotropic locus. The complement system is strongly implicated in age-related macular degeneration and IgA nephropathy and complement targeting therapeutics have been successfully developed/are in late-stage development for both these conditions (e.g. iptacopan just approved by the FDA in IgA nephropathy). Why do the authors think TNFRSF1A is the key mediator of these disease associations rather than CFH itself or the many other trans-associated proteins at this pleiotropic locus? This seems very unlikely to me, particularly given the recent success of complement blocking drugs.

3) I felt the writing was too casual/colloquial, and at times imprecise for scientific writing. While each instance might seem like a minor point, this issue was sufficiently frequent that there was a cumulative effect. Nature Genetics is the premier genetics journal and sets the precedent for the field.

A non-exhaustive list of examples (PS it would have made reviewing much easier if line numbers were provided!):

"Interestingly, our analysis further revealed that the strongest association with SOMAscan at this locus is with mannosidase MAN2B2."

-> SOMAscan is a platform. There cannot be an association "with SOMAscan". (The authors mean association with proteins assayed on SOMAscan)

"Our Open Targets lookup ..." (imprecise/colloquial)

"although eleven of the novel MS pQTLs had been assayed by the deCODE and/or UKBPPP studies "

-> a pQTL is a genetic association with a protein. It cannot be "assayed" - the protein is assayed, not the pQTL. See below re "MS pQTLs".

"suggesting that the respective affinity assays may be targeting different isoforms or having other issues."

-> 'or having other issues' is very vague - what is meant by this?

"...and many others that can now be taken into consideration in the development of drug targets for these diseases. "

-> How many others? It would be better to enumerate.

"The pQTL has an MSPA score of 0.5, placing it into the "grey zone". "

-> I understand what is meant, but this is too colloquial.

" Roughly, the first 100 pQTLs of each study appear to have sufficient power to replicate in Tarkin. "

-> feels imprecise

Minor

Table 1. a) Presumably the first column should say "Protein" and not "Gene". b) The authors should indicate cis vs trans. c) Need to define columns more clearly in the legend.

I do not think the language of "affinity pQTLs" or "MS pQTLs" is appropriate.

Fundamentally, a pQTL is a biological entity. Underpinning the association between locus and protein level is a causal genetic variant. That association has a biochemical mechanism (e.g. an allele in a SNP in a promoter that alters transcription factor binding which in turn impacts mRNA and thus protein levels, or a protein-altering variant that affects secretion from the cell). Thus, something either is or isn't a pQTL. The authors can refer to pQTLs reported based on using affinity-based proteomic measurements and they can flag a subset of those as likely artefacts. However, the definition of a pQTL cannot be changed to make it measurement-system dependent: a variant either is or isn't a pQTL.

ditto for "Olink pQTLs" and "SOMAscan pQTLs" - Olink is a commercial company.

"We found a strong ITLN1 signal in the QMDiab cohort that is not present in Tarkin, which may be discernible only in that cohort due to differences in lifestyle (Figure S5)."

I could not find any data to support the claim that this was due to lifestyle. Was there any?

NB in Figure S5 legend this claim is repeated: discussion is not appropriate for a Fig legend.

" and 67 (65.7%) had been reported at least once by these and/or other pQTL studies covered by Open Targets " -> change "covered" to "curated by" or "included in the OpenTargets database"

"We believe that the MSPA score is a more intuitive measure..."

-> Avoid "We believe..." (this is science, not religion!). Perhaps "We argue..." or "We propose..."

"GWAS further suggest a role for SPINK5 in lung function phenotypes (COPD, FEV) and pancreatitis."

-> More precise to say GWAS have identified associations between SPINK5 and the traits listed.

"that is, cases where a genetic variant interferes with the affinity binding, but at the same time affects protein expression via some biological feedback mechanism."

-> What was the evidence for the statement regarding this putative feedback mechanism"?

"While MS proteomics is not biased versus any particular set of pre-selected proteins"

-> Yes it is not biased to human curation of a set of proteins, but it is biased to higher abundance proteins.

The authors refer to protein expression. Actually, since they are measuring plasma proteins, expression is not necessarily accurate. Protein expression is intracellular and proteins then reach the circulation in different ways (e.g. secretion, trafficking to cell membrane and cleavage, and others). So protein levels or protein abundance would be more correct.

**Although we cannot offer to publish your manuscript, we have consulted with our colleagues at Nature Communications, and they have agreed to continue the review of your manuscript. To transfer your manuscript to Nature Communications after completing the requested revisions, please use our manuscript transfer portal. You will not have to re-supply manuscript metadata and files, unless you wish to make modifications. For more information, please see our http://www.nature.com/authors/author_resources/transfer_manuscripts.html?WT.mc_id=EMI_NPG_1511_AUTHORTRANSF&WT.ec_id=AUTHOR manuscript transfer FAQ page.

Version 2:

Decision Letter:

IMPORTANT: Please note the reference number: NG-LE65652R1-Z Suhre. This number must be quoted whenever you communicate with us regarding this paper.

11th December 2024

Dear Karsten,

Thank you for asking us to reconsider our decision on your manuscript "A genome-wide association study of mass spectrometry proteomics using a nanoparticle enrichment platform". I have discussed your point-by-point response and proposed resubmission with my editorial colleagues. We agree that adding a comparative analysis of pQTLs across proteins measured on multiple nanoparticles could provide useful insights into the extent to which the nanoparticle platform is sensitive to epitope-like effects. We also think that providing further analysis of the extent to which the interpretation of pQTLs at disease-associated loci have been impacted by epitope effects from affinity-based platforms would be valuable. We therefore invite you to revise your manuscript along the lines that you propose for further consideration and peer review. Please note that we would require clear support from the reviewers at the next round of review as a condition for publication in Nature Genetics rather than Nature Communications.

When preparing a revision, please ensure that it fully complies with our editorial requirements for format and style; details can be found in the Guide to Authors on our website (<http://www.nature.com/ng/>).

Please be sure that your manuscript is accompanied by a separate point-by-point response to the points raised at the previous round of review. At this stage, we will need you to upload:

1) A copy of the manuscript in MS Word .docx format.

2) The Editorial Policy Checklist:

<https://www.nature.com/documents/nr-editorial-policy-checklist.pdf>

3) The Reporting Summary:

(Here you can read about the role of the Reporting Summary in reproducible science:

<https://www.nature.com/news/announcement-towards-greater-reproducibility-for-life-sciences-research-in-nature-1.22062>)

Please use the link below to be taken directly to the site and view and revise your manuscript:

Link Redacted

With best wishes,
Kyle

Kyle Vogan, PhD
Senior Editor
Nature Genetics
<https://orcid.org/0000-0001-9565-9665>

Version 3:

Decision Letter:

9th May 2025

Dear Karsten,

Your revised Letter "A genome-wide association study of mass spectrometry proteomics using a nanoparticle enrichment platform" has been seen by Reviewer #2. (Reviewer #3 was also asked to comment but was unable to provide a review at this round.) You will see from the comments below that Reviewer #2 is satisfied with the technical aspects of the work but has ongoing concerns about aspects of the study's framing and interpretations. We remain interested in the possibility of publishing your study in Nature Genetics, but we would like to consider your response to these ongoing concerns in the form of a further revision before we make a final decision on publication.

We therefore invite you to revise your manuscript again taking into account these comments. Please highlight all changes in the manuscript text file. At this stage, we will need you to upload a copy of the manuscript in MS Word .docx or similar editable format.

*2) If you have not done so already, please begin to revise your manuscript so that it conforms to our Letter format instructions, available

http://www.nature.com/ng/authors/article_types/index.html here

*3) Include a revised version of any required Reporting Summary: <https://www.nature.com/documents/nr-reporting-summary.pdf>

It will be available to referees (and, potentially, statisticians) to aid in their evaluation if the manuscript goes back for peer

review.

Please be aware of our [guidelines](https://www.nature.com/nature-research/editorial-policies/image-integrity) on digital image standards.

EXTENDED DATA FIGURES

Link Redacted

We hope to receive your revised manuscript within 4-8 weeks. If you cannot send it within this time, please let us know.

Nature Genetics is committed to improving transparency in authorship. As part of our efforts in this direction, we are now requesting that all authors identified as 'corresponding author' on published papers create and link their Open Researcher and Contributor Identifier (ORCID) with their account on the Manuscript Tracking System (MTS), prior to acceptance. ORCID helps the scientific community achieve unambiguous attribution of all scholarly contributions. You can create and link your ORCID from the home page of the MTS by clicking on 'Modify my Springer Nature account'. For more information, please visit www.springernature.com/orcid.

Sincerely,
Kyle

Kyle Vogan, PhD
Senior Editor
Nature Genetics
<https://orcid.org/0000-0001-9565-9665>

Reviewers' Comments:

Reviewer #2 (Remarks to the Author):

I deeply appreciate the additional effort done by the authors to address my concerns, which helped to clarify technical details. The major finding of the study, the non-replication of previously published 'pQTLs', is robust, important and timely, as the authors also outline in their rebuttal. As a technical report, this paper is great and performed at highest standards.

However, I still take issue in the argument and provided evidence by the authors related to the biological, let alone druggable, relevance of the additional/novel findings presented (see detailed comments below). This is still superficial, which is in part due to the limited sample size, and therefore rather limited 'revenue' of pQTLs with impactful consequences. I agree with the authors that pQTLs should ideally tag variants that modulate the abundance of protein in tissues or blood but would propose a more nuanced view that might be taken into account. There are clear cut examples, by which common coding variants associated with disease risk introduce alternate amino acid sequences of relevant proteins that are more or less well targeted with affinity reagents. While this clearly breaks the underlying assumption of affinity binding reagent take up being proportional to protein abundance, it provides an important indication that the alternate protein product is indeed circulating in blood, and may further, possibly similarly importantly, indicate that protein function rather than protein levels are important for the disease process. This important aspect is currently lacking in the study.

Detailed comments:

1. I am sorry for being nit-picky, but MS does not measure protein abundances either. They measure the abundance of peptides that may (hopefully) be unique to proteins, but have similar limitations in terms of alternative peptides not in databases or similar that make the in silico 'reproduction' of proteins tricky. I agree that true pQTLs should affect abundance

of proteins in plasma, but structural changes may well change behaviour of peptides in the MS or similar. In reality, neither affinity nor MS does really measure protein abundance but infer it indirectly with different sets and assumptions. Let alone the fact that protein function rather than abundance might be the truly interesting insight to get at. Further, one might even question the narrow definition of protein isoform, with PAVs likely introducing proteins with different functional properties that are, quite rightly, no longer detected by affinity reagents. This would explain the successful colocalization of very many proteins with disease endpoints that have likely a PAV as underlying causal variant, e.g., GDF-15 and hyperemesis gravidarum (that the authors, rightly?, flag as epitope effect).

2. The justification for colocalization to attempt replication 'To account for differences in genetic structure between the cohorts' does not make sense. As outlined in my previous response, colocalization is highly sensitive to misspecification in the LD-backbone, most importantly, it assumes that the LD-backbone for the two traits is the very same. If the truly underlying causal variant is tagged by only partially overlapping haplotypes in the two cohorts due to ancestral differences, coloc will vote in favour of H3 (which may account for the ITLN1 finding presented). Trans-ancestral fine-mapping would be needed but would also require whole-genome sequencing data.

3. How is the CFH locus novel? The AMD missense variant and proxies in LD are one of the most pleiotropic loci for the SomaLogic platform.

4. It is essential to demonstrate relevance of pQTLs beyond plasma protein levels, but the conclusion 'and fourteen others that can now be considered in the development of drug targets for these diseases.' is simply too early. It is by no means clear that the cis-pQTLs act only on the protein target or other genes nearby as demonstrated for eQTLs. Let alone that most GWAS traits linked do not represent disease that would be considered relevant for drug development efforts.

5. I am not happy with the response on the FUCA1/2 example. The authors still suggest relevance to AD, which is not supported by large-scale AD GWAS (the authors didn't even attempt to provide a look-up). FUCA1/2 encode for fucosidases with a potentially broad spectrum of protein targets, and the association with Total PHF-tau might well be due to horizontal rather than vertical pleiotropy. Let alone that the trait analyses mentions SNP x SNP interaction that is not otherwise specified. I like to note here that post-translational modifications and genes acting on those are known pleiotropic pQTLs and may, similar to epitope effects, affect protein quantification by affinity reagents, but also MS.

6. The SPINK5 examples lacks the link to the phenotype. While findings are reported, OpenTargets or other data bases do not link the pQTL to associated skin diseases, or any other disease.

7. I struggle understanding the methodological framework of the missingness GWAS, which is an important addition for the overall technical validity of the paper as it tries to address potential epitope artefacts. It is very reassuring to see that half of the proteins had evidence from multiple nanoparticles, but that also meant that the other half is similar doubtful as epitope effects by affinity methods, I guess. Are those pQTLs without support from multiple nanoparticles more likely to tag PAVs? Also, why was not a logistic regression model used instead of a Fisher test that could cope with genetic confounding/population stratification, which likely accounted for the somewhat odd enrichment of loci across ethnicities?

Version 4:

Decision Letter:

Our ref: NG-A65652R3

15th September 2025

Dear Karsten,

Thank you for submitting your revised manuscript "A genome-wide association study of mass spectrometry proteomics using a nanoparticle enrichment platform" (NG-A65652R3). In light of your responses to Reviewer #2, we will be happy in principle to publish your study in Nature Genetics as an Article pending final revisions to comply with our editorial and formatting guidelines.

We are now performing detailed checks on your paper, and we will send you a checklist detailing our editorial and formatting requirements soon. Please do not upload the final materials or make any revisions until you receive this additional information from us.

Thank you again for your interest in Nature Genetics. Please do not hesitate to contact me if you have any questions.

Sincerely,
Kyle

Kyle Vogan, PhD
Senior Editor
Nature Genetics
<https://orcid.org/0000-0001-9565-9665>

Point-by-point response to the reviewers' comments

We appreciate the thorough and positive comments made by the Reviewers. Here is a summary of the major revisions that we made in response to their concerns:

- *We adapted the pQTL identification and replication strategy and now account for possible differences in genetic architecture by using a colocalization test for pQTL signals between discovery and replication cohort.*
- *We addressed the possibility that MS proteomics targets different isoforms and added a meta-analysis on the peptide level that should also respond to several other questions raised by the reviewers.*
- *We added additional support for the use of the MSPA score to characterize the replication of affinity pQTLs using an MS-based platform and also provide alternative measures (see new **Figures S8-S12**).*
- *We revised the identification of overlapping signals from affinity proteomics and now provide information on whether a new pQTL was declared novel because the protein was not measured before, because the signal on the affinity platform was not significant, or because the platforms might be targeting different isoforms.*
- *To highlight the biomedical interest of the newly identified pQTLs we annotated all variants using the latest version of Open Targets and included discussions of the following use cases to the manuscript:*
 - *We showcase a concrete example of how MS proteomics can provide new insights about pQTLs that involve multiple isoforms (SPLINK5, see new **Figure 4**). This pQTL provides an MS proteomics perspective of a splice-pQTL reported by GTEx and highlights how information from these different technologies may be integrated.*
 - *We report the intelectin-1 (ITLN1) pQTL as an example of a case where the genetic signal differs between the two cohorts (coloc hypothesis H3). This pQTL may be of relevance to the fine mapping of this Crohn's disease risk locus (see new **Figure S5**).*
 - *At the example of the FUCA1/2 pQTL we outline how pQTLs from multiple platforms can be used to generate new hypotheses for the drug discovery process.*

While we would also have loved to increase the replication sample size, this was unfortunately not possible due to cost and time restrictions. Note, however, that the evaluation of the deCODE and UKBPPP *cis*-pQTLs is based on the larger Tarkin study with 1260 samples, and also, while a replication rate of 28% may appear low, we actually replicated 80% of the pQTLs with 80% replication power, matching expectations and suggesting that most of the yet unreplicated pQTLs will replicate in future larger-scale studies, i.e. that these pQTLs are not to be considered false positives.

Please find below our point-by-point response to the referees' comments. We also provide a change-tracked version of the manuscript, highlighting all changes made to the text, including revisions to conform to your Letter format. Finally, we also updated the Reporting Summary and Checklist.

Reviewer #1:

Suhre et al. performed an interesting investigation of protein quantitative trait loci based on the Seer multiplex proteomics platform. pQTL studies have shown to be of importance in drug discovery as well as understanding of molecular mechanisms associated with disease, as shown by the SCALLOP consortium and by UK biobank PPP. Most pQTL studies have been performed based on the Olink or Somascan platforms, which are affinity methods that come with the well-known challenge of identifying pseudo-pQTLs, i.e. pQTLs that reflect epitope differences rather than true expression differences.

Response: We thank the reviewers for their thoughtful comments and hope to have addressed all issues they raised.

Comments

The study populations in the present work are ethnically diverse with supposedly quite different lifestyles and genetic backgrounds. Yet, no protein or genetic data across ethnicities are presented. Please provide a plot of population stratification, i.e. the PCs, so that the reader can get a sense for the population structure. Were some of the peptides only observed in people with certain genetic backgrounds? If so, please provide that information, even though the PC adjustment may take care of most systematic population differences.

*Response: Following the reviewer's suggestion, we added the genetic PCA plots for Tarkin and QMDiab to the Supplement. We share the reviewer's view that the PC adjustment takes care of most systematic population differences. While peptides from some proteins were enriched in some populations, we did not identify any peptides that were only observed in people with certain genetic backgrounds. Associations at the protein level across ethnicities are provided in **Table S4** where we report all associations with the co-variables, including the genotype PCs. For instance, as can be seen in the PCA plots, genoPC1 differentiates individuals of European and African genetic ancestry. The strongest association in our study with genoPC1 is apolipoprotein F (APOF). Investigating population differences in protein expression may be an avenue of further research using our data, but we feel that discussing these results is out of scope for our present work. Note that in response to other points, we added a colocalization analysis between the Tarkin and QMDiab cohort. This analysis identified a number of pQTLs with differences in the underlying genetic signals. In the revised version we highlight the intelectin-1 (ITLN1) pQTL as an example of a case where the genetic signal differs between the two cohorts (coloc hypothesis H3). This is a Crohn's disease locus, and we believe that the signal observed in QMDiab, which is stronger than in the original association identified in the larger Tarkin cohort, may be detectable solely in this cohort because of differences in lifestyle and/or exposure (see new **Figure S5**).*

Figure 2b shows the beta levels of identified pQTL across Tarkin and QMDiab. The within platform replication seems lower than previously published Olink studies, which generally show very consistent betas. I lack a section in which the authors explain or speculate on these differences, for example in the discussion section stating other limitations.

*Response: The original Figure 2b included non-replicated loci, which may have given the impression of poor within-platform replication (see revised **Figure 2b**, where we now only include replicated loci). Upon request of reviewer #2 we computed an estimate of the correlation between repeated measurements, using proteins quantified consistently in >80% of samples in all five nanoparticle runs (see new **Figure S1**). The median Spearman correlation for these quasi-technical replicates is $RHO = 0.68$, which we believe is consistent with the spread seen in Figure 2b.*

Line 80, please assign identified pQTL as cis- and trans- according to common practice and report the number of cis- and trans-pQTL separately.

*Response: We assigned pQTLs as cis- and trans-acting in **Table S2** and now report the number of cis-pQTLs in the paper as follows:*

“A total of 364 independent protein associations reached a Bonferroni level of significance ($p < 5 \times 10^{-8}$), involving 295 genetic loci and 274 different proteins, with 177 of the associations located in-cis.”

Line 85, why only attempt to replicate pQTL based on a power calculation? Statistical power should be done a priori rather than looking at post-hoc power. A straight measure of replication across the datasets would be more informative, despite the sample size differences. If the sample size for replication is small, then the alpha value should be set according to the sample size.

Response: We were not clear in the description of our initial approach: We did not attempt to replicate pQTLs based on a power calculation but merely conducted a post hoc power analysis to evaluate the quality of the replication, showing that the replication rates are consistent with the size of the replication cohort.

In the revision we applied a modified approach that we believe responds to these and also some of the other reviewers' concerns:

We extended the number of pQTLs taken forward to replication, but now retain only the strongest association in a region (+/-1MB) that reaches a genome-wide Bonferroni level of significance ($p < 5E-8$).

*In contrast to our previous approach, we did not include secondary signals at a same locus and a same protein at this step but conducted a separate genetic refinement using conditional analyses later. We report the conditionally significant pQTLs in the new **Table S3**.*

*We then tested all primary pQTLs for colocalization between the discovery and the replication cohort. The corresponding regional association plots are provided as **Data S2**. We introduce these plots in the manuscript now at the example of the MST1 pQTL (**Figure S3**).*

In the revised approach we consider a pQTL replicated (1) if the genetic signal is shared between both studies (H4 as the most likely coloc hypothesis), and (2) if the joint association is genome- and proteome-

wide significant, that is, requiring a p-value for the meta-analyzed discovery and replication cohort on the lead SNP of $p < 5 \times 10^{-8} / 1,980$. We found that all pQTLs that satisfied these criteria also had concordant directionality.

This approach allows to additionally identify signals where differences in the genetic architecture play a role, that is, associations that support coloc hypothesis H3. We flag these cases in **Table S2** and provide an example in the new **Figure S5** with the *ITLN1* pQTL.

Line 95, Some peptides would capture a PAV, and these were removed from the MSPA score (line 128). Does this mean that the authors went back to the genotyping data to check if the individual carried the PAV or not, and then removed the peptide measurement? Or does it mean that if there was a potential PAV for a certain peptide, it was removed from all individuals in the study? If the latter, was a frequency cutoff used?

Response: The latter is the case. We previously showed that PAV containing peptides can generate spurious pQTLs in MS proteomics similar to epitope effects in affinity proteomics (Suhre et al., Nat. Comms., 2024). In that paper we developed the approach that we used here, which is to remove all PAV containing peptides with MAF > 10% from the MS library. This cut-off is a trade-off between keeping the MS library size under control while covering all relevant potential PAVs. An ideal approach would be, as the reviewer suggests, to go back to the individual genotype data and use sample-specific libraries. However, doing so requires the development of these features for the current MS data analysis software packages, which is technically out of our reach and beyond the scope of our present work.

Line 128-139, what is the advantage of using a combined peptide score to capture total protein expression vs. conducting a genetic association test for each peptide and then meta-analyses or else combine the results?

Response: The MSPA score represents an ad hoc approach to combine the peptide-level associations by binarizing the peptide level associations into different and not different from the null, at a confidence level of 99%, rather than computing a weighted average, as done in a meta-analysis. The MSPA score provides in our view a more intuitive measure of the joint support from the different peptides, and one that varies less with the effect size of the individual genetic variants. Also in response to some of the

other reviewers' comments, we added a characterization of the MSPA score as a function of MAF, effect size, z-score, protein abundance, etc. (see **Figure S8-S12**).

Nevertheless, we also agree with the reviewer that conducting a genetic association test for each peptide and then meta-analyzing the results is a possible alternative. We therefore now include such a meta-analysis and report the results in parallel to the MSPA scores in **Table S7-S8**. We found that the meta-analyzed effect sizes correlate strongly with the effect sizes obtained from the original analyses using protein levels (see **Figure S10c-d**).

Interestingly, the forest plots generated for the meta-analysis were also instrumental for the identification of cases where a genetic variant associates differently with different protein groups, that we added to the revised manuscript. We are grateful to the reviewer for this suggestion, as it led us to identify the very interesting *SPINK5* case (see new **Figure 4**).

Minor comments:

Line 45, "The largest studies to date", suggest change to "The largest studies published to date"

Response: As suggested, we changed the text to "The largest studies published to date".

Line 49, suggest to add reference on MR in drug discovery, Folkersen et al. Nat. Metab. 2020

Response: We added the reference as suggested.

Line 50, there are already some pQTLs known to be caused by epitope effects, e.g. GDF-15, adiponectin, etc. I think the intro could be improved if some specific proteins were exemplified.

Response: We followed the reviewer's advice and added the following example of an already known pQTL to be caused by an epitope effect to the introduction:

"For instance, a study on blood pressure identified a strong pQTL for circulating natriuretic peptide precursor A (NPPA) with a protein coding variant (rs5063) but failed to replicate it in a six-fold larger study [PMID 19219041]. The authors concluded that the association was artefactual, as the discovery

study used an antibody against an epitope in the mid-region of the molecule, as opposed to the N-terminal epitopes used in the replication study.”

We thank the reviewer for pointing out the GDF-15 example and added it as a confirmatory case as follows:

“Our analysis also confirms a previously reported epitope effect for pQTL of GDF-15 [ref Pietzner et al., Nat. Comms., 2021], reported by deCODE (rank 71 in **Table S9**) and UKBPPP (rank 133 in **Table S10**), which has an MSPA score of zero in our study.”

Line 53, suggest to add additional references regarding epitope effects as these have been well known for many years and described in literature

Response: We agree. To our knowledge, the first paper discussing the effect of epitope changing variants in affinity proteomics GWAS was Enroth et al. (Nature Communications, 2014). We added this reference.

Line 59, suggest to remove the word "smaller-scale" from "A few smaller-scale MS-based GWAS..." since the expression degrades previous work in this space, and the fact that at least one of the studies used a larger sample than the authors describe, albeit fewer proteins.

Response: As suggested, we removed "smaller-scale" from the sentence.

Line 65, suggest to remove the unnecessary word "unbiased", with the motivation that no proteomics method is "unbiased", for example MS is typically biased towards medium and high abundance proteins.

Response: As suggested, we removed the qualifier "unbiased".

Line 294, LD clumping based on $R^2 < 0.1$ may be too liberal for pQTL studies with often strong effect sizes. A cutoff at 0.01 or 0.001 to identify truly independent sentinel variants have been used in other studies.

Response: In the revised version we use conditional analysis to identify independent signals. This R^2 cutoff therefore no longer applies.

Reviewer #2:

The study by Suhre et al. presents a GWAS on plasma protein levels measured with a recently developed mass spectrometry platform in a small-to-moderately sized cohort, demonstrating a number of interesting technical insights in the mapping of protein quantitative trait loci. Most importantly the authors raise (again) and also in line with other work (e.g., Niu et al. MedRxiv 2022) concerns about a substantial number of highly utilized cis-pQTLs from much larger plasma proteomics studies using affinity reagents. For example, they provide evidence that about a third of the most strongly associated cis-pQTLs are potential artefacts with widespread implications for the (over)use as instruments in downstream applications such as Mendelian randomization.

While the study provides important insights for the discovery of pQTLs, that might still benefit from addressing the concerns listed below, it rather reads like a technical report with no insights about the biological (e.g., no presentation on trans-pQTLs or mechanisms underlying cis-pQTL associations) and possibly clinical relevance of ‘newly’ identified pQTLs or how these insights might be leveraged by the community to improve downstream tasks. It had already been demonstrated that molecular QTLs are somewhat depleted for associations with clinically relevant phenotypes and hence it needs to be demonstrated why these more sophisticated but yet not scalable method is worth investing further into.

Response: We thank the reviewers for their thoughtful comments and hope to have addressed all issues they raised.

1) Please omit mentioning of vendors in the title of scientific manuscripts.

Response: As suggested, we removed the vendor’s name from the title.

2) Picky, but the use of ‘drug discovery’ is highly inflated and since the authors do not present evidence how there data can inform drug target discovery, they should better tone such reference down.

Response: We apologize if we appeared to inflate the use of pQTLs for drug discovery. This was not our intention. However, having participated in the recent UKB Olink GWAS (Sun et al., Nature 2023), K.S. can

testify to the large interest and amount of funding that pharma companies are currently pouring into pQTL studies (i.e. via the UKB-PPP consortium). In any case, we toned down reference to drug discovery and also added the following references to illustrate how pQTLs can inform drug target discovery:

- Mountjoy et al., An open approach to systematically prioritize causal variants and genes at all published human GWAS trait-associated loci, *Nat. Gen.* 2023
- Minikel et al., Refining the impact of genetic evidence on clinical success, *Nature* 2024
- Carss et al., Using human genetics to improve safety assessment of therapeutics, *Nat Rev Drug Discov.* 2023

3) Why only 2k proteins present in >80% of samples?

Response: Indeed, the number of proteins that are detectable on the Seer platform is much larger (> 5,000). The number of proteins that can be detected is an important benchmark in the MS proteomics community. However, replication of QTLs for proteins at higher missingness becomes increasingly unlikely. Also, in proteomics GWAS with common variants we are primarily interested in ubiquitous proteins that are detected in almost every sample, rather than rare biomarker proteins that may only be detectable in blood under specific disease conditions. To detect rare variant associations much larger sample sizes would be needed. We therefore decided to use this conservative threshold.

How does the number of identified pQTLs compares to previous studies? I might be wrong, but studies of similar size have identified much more pQTLs previously for a comparable or even lower number of proteins.

Response: The reviewer is correct, studies of similar size using affinity proteomics platforms identified more pQTLs for a comparable number of proteins. For instance, in our first pQTL study using the Somalogic 1.1k platform (Suhre et al., *Nature Comms*, 2017) with a comparable sample size we identified 539 pQTLs, compared to 364 primary pQTLs and 74 conditional pQTLs in the present study, while including fewer proteins (1,124 vs. 1,980). We attribute this difference to higher technical variability inherent to MS-based methods. However, while affinity methods have better CVs within the same platform, they often do not agree with each other, suggesting that some of the affinity pQTLs may be specific to the affinity binder.

4) The sentence “it should be noted that SOMAscan and Olink pQTLs represent genetic associations with protein binding affinity, rather than protein expression” is not quite true. It should be acknowledged that in most cases that there is a reproducible link between the number of reagents that bind to their targets and the abundance of the target. What the authors refer to is the very specific case, in which we have genetic association that may lead to a dissociation of this dependency.

Response: We agree and extended this statement following the wording of the reviewer’s comment:

“However, affinity proteomics QTLs represent genetic associations with protein binding affinity, rather than protein expression, which assumes that there is a reproducible link between the number of reagents that bind to their targets and the abundance of the target. This link may break when a protein altering variant (PAV) is located in the site where the aptamer or antibody binds, thereby leading to a genotype-dependent read-out that does not correspond to a real change in protein level [ref Enroth et al., 2014].”

5) Please avoid terms like ‘unbiased’ it is virtually impossible to do anything ‘unbiased’, as this would assume that we would have complete knowledge of the world. For example, in the current study, it is unclear whether there is any differential binding of protein targets in plasma that are not picked up by nanoparticles or peptides that can be mapped in DIA-NN (which relies again on our knowledge of what protein isoforms exist) So, I encourage the authors to not only be more conscious about the language used, but also to discuss these limitations.

Response: We removed the term “unbiased” from the text and added the following text to the discussion of the limitations:

“While MS proteomics is not biased versus any particular set of pre-selected proteins, it is biased towards protein isoforms that are present in the utilized database, proteins that are enriched using one of the five nano-particles, and proteins that can be cleaved into peptides and are detectable by the applied MS proteomics method.”

6) Why age and sex associations?

*Response: We initially included this analysis as an indirect means of qc'ing the data, mainly because reviewers of our previous pQTL studies requested such analyses. Personally, we have no strong feelings about including or discarding this part. It adds interesting insights but is out of the scope of our GWAS. Therefore, and also to comply with the word and figure limits of the Nature Genetics Letter format, we moved the age and sex associations to the **Supplementary Text** where an interested reader can still peruse the data.*

7) Can the authors please elaborate more on potential sources of non-replication? To what extent did different LD-structures contribute to the rather moderate level of replication, e.g., selecting a tag SNP in Tarkin which is not in LD with the underlying causal variant in the Arab cohort?

Response: This is an important point that we elaborate further in the revised manuscript as follows:

“The main reason for the non-replicated pQTLs is the limited size of the replication cohort: Of 70 pQTLs that had 80% replication power, 58 (82.9%) replicated and most (36 out of 38) of the nominally significant ($p < 0.05$) pQTLs had concordant directionality, suggesting that many of the unreplicated pQTLs are replicable in future larger-scale studies. Also, 14 of the non-replicated pQTLs had significant but different genetic signal in both cohorts.”

*With the revised replication strategy based on colocalization, selecting a tag SNP in Tarkin which is not in LD with the underlying causal variant in the Arab cohort should not be an issue anymore: In addition, in the revised Table S2 we now also report the strongest association in QMDiab, regardless of LD, so that more complex genetic architectures can be revealed. Using this information, we identified the ITLN1 case (see new **Figure S5**).*

8) The stratification of cis-pQTLs on the peptide levels is important, but currently lacks a systematic assessment. How often are pQTLs supported by only a subset of matching peptides and to what extent can this be attributed to peptides matching to multiple proteins? I appreciate that the MSPA score tries to get at this, but it does not account for the fact the peptides may map to multiple different protein

targets and might further be biased if soluble and complete forms of the same protein are clumped together (affinity reagents might better distinguish here).

Response: We agree that peptides matching to multiple isoforms can be an issue and that stratification of cis-pQTLs on the peptide levels is important. We now address this concern by conducting a meta-analysis at the peptide level (see also our response to reviewer #1 above) where we identify those cases where peptides matching multiple isoforms show conflicting directionality. Conflicting directionality of the QTLs at the peptide level can be indicators for the concurrent presence of different isoforms. We identified ten such cases in the investigated deCODE cis-pQTLs and eight in UKBPPP and added a discussion of one such case (SPINK5, new **Figure 4**).

9) How were novel/reported pQTLs classified? Given large variations in ancestral background a regional definition seems to be most appropriate, by which a 1-Mb window surrounding the sentinel variant has not been reported with any protein measurement previously. To what extent are novel pQTLs explained by proteins not targeted before?

Response: In the revised version, novel/reported pQTLs were identified in three ways:

1. Direct lookup of the pQTL variants in the summary statistics from the deCODE and UKBPPP studies.
2. Proximity (+/- 1MB window) of our pQTL lead SNPs to a matching pQTL reported in the Supplementary Tables of the deCODE and UKBPPP papers (this captures pQTLs that are not in LD).
3. Identification of a pQTL using the Open Targets API (this captures pQTLs from most previous pQTL studies).

If a p-QTL was not found by any of these three methods, it was declared novel. 217 of the 364 pQTLs were novel according to this criterion, including 35 of the 102 replicated pQTLs. We report the details of this analysis in **Table S2**.

To what extent are novel pQTLs explained by proteins not targeted before? Twenty-four of the 35 novel replicated pQTLs were explained by proteins not targeted before. Eleven proteins were assayed in

*deCODE and/or UKBPPP but did not reach nominal ($p < 0.05$) or genome-wide ($p < 5 \times 10^{-8}$) significance at these loci (independent of LD), suggesting that the respective assays target different isoforms or have sensitivity issues. We provide this information in the updated text and revised **Table 1**.*

10) Given the very limited sample size, the authors should rather stay away from quantitative assessments like ‘an expected novel discovery rate when using the Proteograph platform of about one in three compared to existing affinity pQTLs’. The same applies to the inference of how many epitope effects there possibly are with Olink/SomaLogic, given that the moderate sample size of this study will almost certainly bias the results. It would be much more insightful to expand these analyses given that there are already substantial concerns about the most strongly associated cis-pQTLs. Further, the MSPA generated does not distinguish either for the differential abundance of different protein isoforms or soluble fragments.

*Response: We followed the reviewer’s advice to qualify quantitative statements and expanded the analysis for strong affinity pQTLs that are lacking MS support; see new **Figure S13** where we show cases where a strong signal on PAV containing peptides was detected while no MS support for a protein expression QTL signal was found.*

11) Presenting technical data on reproducibility of the nanoparticle enrichment and MS measurements would be important.

*Response: Given the high costs of the experiments, we did not run duplicate samples to directly evaluate reproducibility on our data set. However, the Seer platform implicitly provides such information as many proteins were quantified in more than one nanoparticle run. We argue that proteins detected and quantified in all five nano particle extracts at low missingness can be viewed as quasi-technical replicates. For the present study we therefore computed the correlations (Spearman) between the nanoparticle replicates of 615 proteins that were quantified in over 80% of the samples in all five nanoparticle runs. For each set of measurements, we retained the strongest correlation of the ten pairwise correlations, assuming that the most favorable correlation represents the most similar conditions and that this correlation is a lower bound for the correlation of two true technical replicates. The median Spearman rho for these proteins was 0.68. We report these values in **Table S1**, added a*

histogram as new **Figure S1**, and verified that the MSPA scores were not biased towards proteins with a high technical reproducibility (**Figure S12**, panels **m-p**).

12) The age and sex association are somewhat odd, since it is not really known what the ground truth would be. If anything, such analysis should be used to establish any potential systematic differences across platforms, e.g., is there anything specific to proteins seen to be associated with age with one but not the other platforms. It is also odd to place this analysis within the reporting of genetic work.

Response: We agree. Please refer to our response to a similar point above. This analysis has been moved to the **Supplementary Text**.

13) The section “One third of affinity proteomics pQTLs are potentially affected by epitope effects.” Is written in a very odd style, and too often the authors refer to massive sets of simple plots instead of providing some tangible insights and subsequent justification. For the overlap between pQTLs discovered in SomaLogic or Olink with Seer, it is also unclear to what extent the true underlying causal variant had been selected. While the trans-ancestral design is desirable, it also complicates these matters since we would need to be even more sure about the true underlying causal variant.

Response: We entirely rewrote the section in question (see revised manuscript). We moved the large number of plots to a Supplementary Data status as a reference for readers who wish to verify details for specific pQTLs.

*We agree that the trans-ancestral design while desirable for many other reasons complicates matters when it comes to comparing genetic signals between populations. We therefore added the colocalization analyses to the replication. To address the question to what extent the true underlying causal variant had been selected we conducted a colocalization analysis between Tarkin and UKBPPP on the peptide level. We then computed coloc-MSPA scores for coloc hypothesis H3 (different genetic signal between Tarkin and UKBPPP), hypothesis H4 (shared signal), and both (signal in both, regardless of the genetic architecture, see **Methods**). This approach eliminates the need to evaluate the signal on a single SNP and accounts for the genetic architecture of the entire region. We found that only one pQTL switched from MSPA <0.2 to coloc-MSPA > 0.8 status. We added the following text to represent this observation:*

“To rule out that differences in genetic architecture led to a low MSPA score for some of the pQTLs we computed a similar score based on genetic colocalization as the weighted fraction of peptides for which coloc favours a shared (H4) or a different (H3) signal between the UKBPPP and Tarkin studies (see **Methods** and **Table S10**). Out of 224 pQTLs with MSPA < 0.2 only one pQTL (ITGA2) had a coloc-MSPA > 0.8 for H3, none for H4, and 205 had a coloc-MSPA < 0.2 for the combination of H3 or H4.”

14) The reasoning not to demonstrate the biological relevance of ‘newly’ identified pQTLs is pointless and would have been important to distinguish this paper from a merely technical report.

Response: We agree. In the revised paper we therefore annotated all pQTLs using Open Targets to link the pQTLs to most available GWAS studies (Table S2). We report the relevant biomedical associations for the 35 novel pQTLs in Table 1 and highlight examples further in the revised text as follows (reviewer #3 raised a similar point and we reproduce this response there):

“The 35 novel pQTLs overlap with several loci of biomedical relevance, including a trans-pQTL at the COLEC11 loci with ANGPTL6 that associates with LDL-cholesterol levels, a trans-pQTL at the CFH locus that implicates the TNFRSF1A modulator BRE in age-related macular degeneration and IgA nephropathy, a cis-pQTL for galactosylceramidase GALC with inflammatory bowel disease, and many others that can now be taken into consideration in the development of drug targets for these diseases.

We also found pQTLs that complement findings from affinity pQTLs, like an Olink trans-pQTL for fucosidase FUCA2 (rs11155297) that we replicate and extend by the “missing” cis-pQTL for FUCA1 (Table S2). Interestingly, our analysis further revealed that the strongest association with SOMAscan at this locus is with the mannosidase MAN2B2. FUCA1, FUCA2, and MAN2B are all enzymes involved in the lysosomal degradation of glycoproteins and glycolipids. Our Open Targets lookup additionally revealed a GWAS signal for “Total PHF-tau (SNP x SNP interaction)” (p-value = 2×10^{-8}) [PMID:32450446]. While speculative, these genetic signals obtained on three different proteomics platforms provide a starting point for further investigations into the role of glycoproteins in Alzheimer’s pathology, and more generally outline how pQTLs can be used to generate hypotheses for the drug discovery process.”

We also added the discussion of a pQTL that reflects an over-lapping splice QTL (SPINK5, see new Figure 4) and a pQTL with differing genetic signals between the cohorts (ITLN1, new Figure S5). These examples should demonstrate the biological relevance of the underlying pQTLs.

Finally, we extended the annotation of all pQTLs using the Open Targets API, including the pQTLs that we evaluated for epitope effects from the deCODE and UKBPPP study (**Tables S9-S10**).

15) I don't understand this sentence 'The WCMQ IRB determined that use of the Tarkin data for the present project does not meet the definition of human research for this study (IRB document HRP-532', why is this study not human research?

Response: This sentence is from our IRB document and declares that the Weill Cornell IRB determined that their researchers analyzing Harvard data is not considered human research in the IRB sense, that is, they do not interact with the Harvard study participants, nor do they have access to any of their private data. This determination implies that the present study does not require full board IRB review from Cornell.

16) The ancestral assignment in the Tarkin study is odd. What is meant by 'white'? Why have participants not be assigned to ancestral groups using external PC loadings? This might introduce residual population stratification and hence also explains inflated GWAS stats for some proteins, which is really unusual for such a small data set.

Response: The ancestral assignment is based on self-reporting ("white" is from a questionnaire item). Missing responses explain the unassigned samples. Please note that ancestral assignment did not enter the data analysis, but that the first ten genomic PC components were used as covariates to account for ancestry. In the revised version we added PCA plots and removed counts of participants of any specific self-reported ancestry altogether. Only three proteins showed inflated GWAS statistics that might be due to residual confounding with ancestry. We write:

"Three of these proteins presented with inflated GWAS statistics and the corresponding pQTLs were removed from the analysis (UniProt V9GYE7, B1AKG0, and O15230)."

*All other proteins had genomic inflation (λ) values that were close to one (individual values are in **Tables S1 and S2**). We therefore believe that there is little to no residual population stratification.*

17) Using an r^2 of 0.9 to clump sentinels across proteins is too stringent and other thresholds, possibly down to 0.6 should be explored to define loci.

Response: We explored the clumping using the suggested threshold of 0.6. Fourteen pQTLs were affected and assigned to other loci rather than being considered as independent signals. We report the results of both sentinel clumping approaches in Table S2 and used the suggested $r^2 = 0.6$ to report the loci count in our paper.

18) Please clarify whether ANML or non-ANML results have been used to compare to deCODE results.

Response: We used the summary statistics and supplementary data that were provided with the Ferkingstad et al. study (Nat. Gen. 2021). To our knowledge, the study used only one single data set and there is no reference in that paper to whether this was ANML or non-ANML data.

19) Many of the inferences drawn across platforms implicitly assume that each of them measures the very same 'thing', which is unlikely to be true, and correlation analysis across different proteomic technologies would be needed to establish this. For example, it had already been suggested that different technologies measure different isoforms of the very same protein and hence comparing those must lead to different results.

Response: We agree. We modified the manuscript in several places to reflect this caveat, i.e. state that non-replication of an affinity pQTL using MS can be due to platforms measuring different "things", i.e. we now report the level of matching of affinity proteins to MS protein groups and added an example where multiple isoforms are detected at the MS level (SPINK5, new **Figure 4**).

20) The Manhattan-like plot looks odd and should be replaced with a beta vs. MAF plot or similar, given that most associations are cis-pQTLs (so the location in the genome doesn't really matter).

Response: We added the suggested a beta vs. MAF plot (**Figure 2a**) and, as requested by reviewer #1, replaced the Manhattan plot by a 2D grid plot (**Figure 1**).

21) To what extent are pQTLs discovered with MS also prone to technological artefacts? For example, the authors previously highlighted the role of variants mapping to sites changing tryptic digestion.

Response: This is a very important point. In principle, when peptides with coding variants are included into the protein quantification, spurious pQTLs that are similar to the affinity proteomics epitope pQTLs can be observed. We addressed this point in our previous work (Suhre et al., Nature Comm., 2024), where we defined the approach that we implement here in our GWAS. In that approach we account for variants that change tryptic digestion sites and remove these from the library, together with the “simple” amino acid substitutions. In the discussion we now write the following:

“The use of a proteome FASTA library that accounts for PAVs was central to our study. Without using this approach, a very high number of false positive pQTLs would have been detected, as we previously discussed [Suhre et al., Nat Comms 2024]. Traditional bottom-up proteomics data analysis pipelines often rely on a limited peptide library for protein quantification, and the presence of a single peptide with a large effect can skew the quantification. A PAV-containing peptide would not be detected in homozygotes of the alternate allele and heterozygotes would have half the peptide level. The inclusion of PAV-containing peptides into the protein quantification would hence lead to the equivalent of an epitope effect, that is, a pQTL signal where in reality, there is no genotype dependence of the protein expression level. Also, PAV peptides corresponding to the alternate allele are not present in an in-silico digest library of the standard UniProt database. Indeed, using data from a standard proteomics data processing run, we observed cases where fragment spectra of these peptides miss-matched to peptides from other proteins and in extreme cases led to false protein identifications.”

22) Figure 5 is the very central message of this work, but looks really odd. The massive overlay of the sigmoid curve and some other highlighted section appears superficial, simply categorizing the MSPA would be more informative. Also, how would this figure look like using z-scores instead of ranks, and to what extent has allele frequencies been taken into account? For example, a very common missense variant introducing an epitope effect will have a much stronger p-value compared to less frequent variants.

Response: Following the reviewer’s suggestion we added a number of analyses to investigate the dependence of the MSPA score as a function of z-score, effect size, log(p-value) and MAF (Figure S8-S12).

Please note that deCODE reported explained variance rather than S.E. in their Table ST02. The dependence of the MSPA score on these measures is similar for the strong pQTLs (imagine drawing a vertical line between the area containing the red/green dots and the rest but compresses the region of the less significant pQTLs. The BETA / S.E. plot for UKBPPP is quite similar (mirrored) to our plot against the ranks.

To further support our reasoning that our study is sufficiently powered to detect the strongest 100 pQTLs from deCODE and UKBPPP, and that a third of these pQTLs had no MS support, we computed the rolling average of the MSPA score as a function of pQTL rank (new **Figure S12**), showing that the fraction of pQTLs with low MSPA scores (<0.2) levels off at ~30% for the strongest pQTLs and that the number of pQTLs with high MSPA scores (>0.8) drops to near zero for weaker pQTLs, above rank>150 for deCODE and >200 for UKBPPP. These observations support our interpretation of the original Figure 5, indicating that the red region represents pQTLs with sufficient power to detect these pQTLs using MS, which yet do not show a signal.

23) Figure 7 is somewhat hard to interpret, since the comparator is missing, since PAV-peptides were excluded from the study.

Response: PAV-peptides were not entirely excluded from the study, only from the quantification of the proteins used in the GWAS. The first two rows of the old Figure 7 (now **Figure S3**) show peptides that do not include PAVs (using the PAV exclusive library). However, whenever a PAV containing peptide was identified we included it in the third row of these plots (using the PAV inclusive library). In any case, the comparator is not the number of PAV containing peptides, but the number of non-PAV containing peptides that support a non-null association with the genotype, as the MSPA score is a weighted fraction of these peptide level QTLs.

We also added a new **Figure S13** that shows the deCODE and SOMAScan cis-pQTLs with low MSPA scores for which also PAV-peptides were detected. These cases clearly show that the PAV-containing peptides capture the SNP while the other peptides all are devoid of a pQTL signal on the peptide level, both in Tarkin and QMDiab.

24) What distinguishes proteins detected with Seer but not with SomaLogic and Olink? Do the authors observe any differences between proteins commonly detected by each of the methods compared to those measured with only one of the affinity reagents?

*Response: To answer the reviewer's question, we annotated the proteins reported by the three studies using the Ingenuity Pathway Analysis (IPA) tool (reported in the new **Table S6**). We added this information to the manuscript as follows:*

*"There are also some differences in the protein panels covered by the different technologies (**Table S6**). Relative to their respective panel size, the Seer platform covers the largest fraction of cytoplasmic proteins while Olink leads in membrane proteins. Somalogic has the lowest fraction of extracellular proteins but most proteins originating from the nucleus. As expected, low abundance proteins, such as cytokines, are less frequently detected by the Seer platform."*

25) The study centres a lot around cis-pQTLs, but also identifies rather pleiotropic trans-pQTLs frequently linked to blood cell counts. How did the authors account for such residual findings? What quality were samples of?

Response: Blood cell counts are frequently found to overlap with GWAS hits, not specifically with pQTLs. This is probably because there are so many blood cell type QTLs in the GWAS catalog. We did not specifically account for blood cell types, as we did not have access to this information, nor did most of the affinity proteomics studies. However, we agree that this may be a potential confounder for trans-pQTLs and should be included in the future.

The samples were of biobank quality, that is, unfrozen samples processed and aliquoted immediately after blood draw using standard protocols and then kept at -80C until measurement.

26) Finally, to what extent can the authors provide guidance on how to use their results to refine pQTLs studies in much larger affinity-based studies? For example, to what extent are secondary signals at cis-loci valid proxies, even if the main signal might be an artefact.

Response: Thank you for this question. Indeed, refinement of pQTLs is central when collecting genetic evidence to support specific drug targets. This task cannot be automated and run in high throughput but requires a manual case-by-case approach to view all available evidence, both from large scale affinity-based studies that have the power to identify weaker signals, and from MS-based studies like ours that provide a more detailed view down to the peptide and isoform level. We have collected as much of that information as possible in the Supplementary Tables and Data.

The reviewer is correct, the presence of an epitope effect for one genetic variant does not invalidate the assay as a whole, and secondary signals at cis-loci can be valid proxies. In particular, rare variants may disrupt affinity binding, and thus suggest strong epitope effects, but they would only affect a few samples.

It should also be noted that epitope effect-causing variants may be clinically relevant even if they do not change the protein expression (like GDF-15). Eventually, the impact of the variant on the protein structure needs to be evaluated, part of which is accessible through the association with peptides that match to different isoforms of the protein under investigation, part of it can also be achieved with affinity binders that target specific isoforms.

*In the revised version we provide an example of how our data can be used on a one-by-one basis to interpret complex pQTLs, like the SPINK5 case (see new **Figure 4**).*

27) Minor, but phenoscanner is outdated and no longer maintained. SNPs should rather be queried using the OpenGWAS or OpenTargets portals, the latter would also allow to establish overlap of pQTLs with GWAS credible sets.

*Response: Thank you for the suggestion. We now use the Open Targets API to annotate the SNPs (results are aggregated in **Table S2** for the 364 discovered pQTLs, and in **Tables S9-S10** for the deCODE and UKBPPP pQTLs that we evaluated in this work).*

Reviewer #3:

This paper maps the genetic determinants (pQTLs) of 1,980 plasma proteins in a relatively modest size cohort (n=1,260 in the discovery cohort). Over the past few years there has been a rapid increase in the number and sample size of pQTL studies, mostly using aptamer-based (SomaScan) or antibody-based proteomic measurements (e.g. Olink Proteomics). The largest of these studies performed pQTL mapping in a subset of UK Biobank participants (~50K individuals) using the Olink proteomics platform (Sun et al, Nature 2023). Although the current study is considerably smaller than these previous studies, it adds value to the field through the use of a mass spec (MS)-based platform. Antibody or aptamer-based technologies are vulnerable to so-called 'epitope effects' whereby a coding genetic variant changes protein shape and affects binding of the antibody and aptamer thus creating a spurious pQTL association. The MS-based measurements in the present study potentially circumvent this issue. Using MS-based pQTL mapping, the authors suggest a substantial proportion of previously reported putative pQTLs to be the result of technical effects rather than quantitative differences in protein levels, although I have some reservations about the robustness and generalisability of this conclusion given that MS may lack sensitivity and this study is underpowered versus the deCODE and UK Biobank pQTL comparator studies. If the authors' conclusion is correct (and they do provide some specific convincing examples), it would be an important finding with implications for revisiting the interpretation of downstream analyses such as Mendelian randomisation (MR). In addition, the authors were able to perform pQTL mapping for proteins not included on the Somalogic or Olink platforms.

This study may represent an important proof of principle for the field and with implications for interpretation (and potential revision) of previous MR studies. However, sample size and particularly the small replication cohort lead to concerns about the robustness of the findings. Generally, the paper was clearly written but parts of the Results felt more like Methods, and some re-writing to improve accessibility to a wider audience would help. Figures need improvement in terms of clarity and better display of useful information (e.g. Figs 1 and 3). The authors should better map their data (new pQTLs and 'pQTLs' that debunk as technical artefacts of binding assays) onto disease GWAS association signals for biological insights.

Response: We thank the reviewers for their thoughtful comments and hope to have addressed all issues they raised. Please refer to our responses to the specific points below.

Point 1. (major)

The replication cohort was a very weak aspect of the study given that a) the sample size was very low and b) the ancestry of the replication cohort was different from the discovery cohort. Accordingly, the authors are forced to use a liberal statistical threshold to define 'replication'. The reader is left unconvinced by replication (of particular relevance to the previously unreported pQTLs in Table 1), and conversely lack of replication for a given protein is hard to interpret given the lack of power. Ideally, the replication cohort needs to be expanded, but I appreciate there may be practical and funding constraints and despite the weakness of the replication cohort I appreciate the potential value of the discovery data.

Response: We agree that the replication cohort is small compared to the discovery and we appreciate the reviewer's understanding that extending the study would be unpractical due to funding constraints.

However, please note that we did not use a liberal statistical threshold to define 'replication' but adhered to standards generally accepted in the GWAS field (i.e. using Bonferroni levels of significance for multiple testing). In the revised version (and in response to concerns that difference in genetic architecture could be an issue) we further added a colocalization analysis and now call a pQTL replicated if (1) coloc finds that both studies most likely share a same genetic signal and if (2) the joint p-value from both studies exceeds a genome- and proteome-wide level of significance ($p < 5 \times 10^{-8} / 1,980$). All pQTLs that met these criteria also had concordant directionality.

In addition, we provide a post-hoc power analysis and show that we replicate over 80% of the pQTLs that have 80% replication power. We further found that most (36 out of 38) of the nominally significant ($p < 0.05$) pQTLs had concordant directionality, suggesting that most discovered pQTLs are in principle replicable, given sufficient power.

Please also note that the evaluation of the deCODE and UKBPPP cis-pQTLs is based on the much larger Tarkin study.

Point 2.

Abstract:

“Genome-wide association studies (GWAS) with proteomics are *essential* tools for drug discovery”

This is an untrue statement. Most transformative drugs were developed without knowledge from proteogenomics (e.g. antibiotics for infection; anti-TNF blockade and other biological therapies for inflammatory disease; aspirin, antihypertensives and statins for cardiovascular disease; chemotherapy and checkpoint blockade for cancer etc etc). It is true that (in retrospective analyses) drugs which target a pathway implicated by human genetics are more likely to be successful in late phase clinical trials but this does not equate to being ‘essential’, and the number of drugs developed de novo primarily off the basis of genetics remains very small (e.g. PCSK9 inhibitors are a rare example). Please avoid hyperbole and maintain scientific accuracy by changing to ‘essential’ to ‘useful’.

Response: We apologize if we were inaccurate in our wording – we meant “essential” in the sense of “central to the work of the data analyst”. We avoid such hyperbole in the revised version.

Point 3.

Abstract

Please change ‘blood samples’ -> ‘plasma’ samples for maximum clarity (there are differences in pQTL between serum vs plasma for some proteins).

Response: We agree – we intended to emphasize the use of “blood samples” in contrast to other bodily fluids or tissues. In the revised version we now refer to “blood plasma samples”.

Point 4.

Replication: “A total of 90 pQTLs satisfied these criteria.” Please clarify here exactly what is meant by a “pQTL” in this context. Specifically, is it a locus-protein association or just the number of loci (irrespective of pleiotropic pQTLs associated with multiple proteins)? And are conditionally independent variants counted in this total?

*Response: Our definition of a pQTL is that of a lead variant – protein pair in a region of +/- 10MB, where we consider conditionally independent variants in a same region as secondary pQTLs. In the revised version we identify and report secondary pQTLs in **Table S3**, but we do not count them with the 364*

pQTLs in order not to inflate the results. We then clumped the lead pQTLs at pleiotropic loci, initially at an LD level of $r^2=0.9$, and upon request of reviewer #2 now also at $r^2=0.6$. We found 308 and 295 independent primary signals, respectively (Table S2). We explain these points in the revised version.

Point 5.

Page 4 “MS-based proteomics platforms generate a *rich* set of readouts”.

‘Rich’ is imprecise and sounds more like a sales pitch for the technology– please rephrase using appropriate precise scientific wording.

Response: We apologize if this sounded like a sales pitch. To be clear, it was the first author’s wording, who has no stakes in the technology nor the Seer company and also works equally well with several of the affinity proteomics providers. What I meant to convey is that in contrast to affinity proteomics, where the user obtains just one readout per protein and sample, in MS proteomics the user has to make sense of data for many peptides and in addition has a choice of multiple spectral data analysis approaches and their parameters, including the selection of the most suitable protein library. I used this term as a euphemism for “MS data analysis is complicated, but it can be rewarding”. We avoid this term in the revised version.

Point 6. (major)

Fig 1 (Manhattan plot) doesn’t convey any useful biological information as there is no annotation of any peaks/loci or which proteins these pertain to. Please consider a more informative plot – there are many other ways of conveying the data (see previous pQTL papers for examples e.g. Circos plot or grid plot).

Response: As suggested, we replaced the Manhattan plot by a grid plot. We feel that Manhattan plots mainly serve the purpose of visualizing the overall genomic distribution of the signals, i.e. to show that there are no “weird” patterns in the association signals, like hyper-pleiotropic loci (e.g. the VTN locus on the SOMAscan platform).

Point 7. (major)

Fig 3. I found the annotation on the plots confusing and it was not readily clear whether all peptides relate to the same protein. Does the annotation in the left lower corner apply just to that row or all subplots? It appears that the peptides shown vary between the discovery cohort (green) and

replication (grey). There was too much visual clutter and inconsistent labelling (eg the suffix “pc3” lower case in some title and “NP5” in another). Some variants are indicated by chr:position (discovery cohort) and others rsid (replication). Is there a reason for the inconsistency? The legend was very long (I suspect beyond journal word limits) and the need for such length reflects poor graphical display. Please re-make this figure to improve clarity.

In what sense is the pQTL shown “prototypical”?

Response: We agree that these plots did not serve the purpose of a figure in a paper. They are actually Supplementary Data and we now label and treat them as such.

The purpose of reproducing one page of these plots as “prototypical” pQTL in the manuscript was to direct the reader to this data (“prototypical” was meant in the sense of “a representative example”; we dropped that word). In the revised version we moved this figure to the supplement. We also modified the legend to clarify the specific questions that the reviewer asked: the annotation in the left lower corner applies to the entire page, the peptides shown vary between the discovery cohort (green) and replication (grey) because we only show the four strongest peptide associations per cohort, and their order can vary, the suffix “pc3” refers to the precursor charge of the peptide and “NP5” refers to the nano-particle extract from which the protein and peptides were issued. Variants were indicated by chr:position in the discovery cohort and by the corresponding rs identifiers in the replication cohort in order to include both identifiers and save space.

*We also understand that some of these details were not sufficiently explained and modified the plots for clarity. In particular, as we now provide these plots as Supplementary Data, we felt that we could increase the number of plots in order to visualize a maximum of information, now showing associations for up to 19 peptides per study. It is our understanding that a reader may want to take a closer look at specific pQTLs, without having to download and process the raw MS data, which is available on the PRIDE database. As explained elsewhere, we also added forest plots of a peptide-level meta-analysis that will allow the reader to verify on a case-by-case basis whether there are multiple isoforms involved in the associations, and if their directionality varies. We provide an example for the interpretation of this data with the SPLINK5 case (see new **Figure 4**).*

Point 8.

Page 5: overlapping pQTLs from UKBB and DeCode studies. Was some check that these were likely the same signal performed? E.g. colocalization analysis or at least a check that the sentinel variant in the authors study was in high LD ($r^2 > 0.8$) with the sentinel in the external study.

Response: In the revised version we conducted the pQTL lookup in three steps (see also our response to a related comment by reviewer #2):

- 1. Direct lookup of the pQTL variants in the summary statistics from the deCODE and UKBPPP studies*
- 2. Proximity (+/- 1MB window) of our pQTL lead SNPs to a matching pQTL reported in the Supplementary Tables of the deCODE and UKBPPP papers (this captures pQTLs that)*
- 3. Report of a pQTL in any of the proteomics GWAS covered by openTargets*

We believe to have captured and characterized most overlapping pQTLs by this approach.

*In our definition of novel pQTLs we are conservative and only count pQTLs that were not detected by any platform in any of the three steps. We therefore also count pQTLs as known if they are only in proximity to a previously reported pQTL. In **Table S2** we further report the LD r^2 between our pQTL variants and the pQTLs reported by the UKBB and deCode studies.*

Point 9a. (major)

Page 5 subtitle “One third of affinity proteomics pQTLs are potentially affected by epitope effects.”

This claim cannot be fully substantiated given only a small fraction of cis pQTLs reported in previous studies were actually testable here, and therefore is potentially misleading. Please re-word to reflect the actual findings.

Response: Also in response to comments from reviewer #2, we added:

“Identification of an epitope effect, however, is more challenging, as lack of support of an affinity proteomics pQTL using MS methods can also be due to other reasons, including insufficient statistical power, lack of sensitivity of the MS method, platforms targeting different isoforms, or genetic variants representing different genetic signals between study populations.”

*In the revised manuscript we then discuss these possibilities at the hand of new analyses that we added: lack of sensitivity by looking at the correlation between quasi-replicates (**Figure S1 and S12m-p**) and the*

dependence of the MSPA score on absolute protein concentrations obtained from the Human Protein Atlas (**Figure S11**); lack of power by further investigating the dependence of the MSPA score on pQTL rank (**Figure S12**), targeting different isoforms by including the meta-analysis on the peptide level (**Figure S6**), and differences in genetic signal by colocalization at the peptide level (**Table S10** and main text, where we write: “Out of 224 pQTLs with MSPA < 0.2 only one pQTL (ITGA2) had a coloc-MSPA > 0.8 for H3, none for H4, and 205 had a coloc-MSPA < 0.2 for the combination of H3 or H4.”).

We reworded the summary of our findings as follows:

“One third of the evaluated pQTLs were confirmed by MS proteomics to be consistent with the hypothesis that genetic variants induce changes in protein expression while another third could not be replicated and are possibly due to epitope effects, but alternative explanations for non-replication need to be considered on a case-by-case basis.”

Point 9b. (major)

Note that MS typically lacks sensitivity compared to antibody or aptamer based protein measurements. An alternative explanation for a lack of genetic association signal in the authors’ data (aside from lack of power) is insufficiently low lower limit of detection of the MS measurements. This is important issue that would substantially change the interpretation of the authors data.

To address this, I would suggest the authors test whether lack of replication of a UKBB or deCODE pQTL signal in their MS data correlates with abundance of the protein in question. They could use absolute quantification data where available (e.g. for many proteins mean plasma concentrations are well known). Alternatively, both Olink and Somalogic also provide a lower limit of detection for their assays (relative values). The authors could gauge whether a protein is low abundance by looking at the distributions of values in UKBB samples in relation to the Olink LLOD.

Response: We address this issue as suggested by the reviewer. We used absolute quantification data that was available in The Human Protein Atlas (HPA) for 1629 out of the 1980 proteins we investigate (<https://www.proteinatlas.org/humanproteome/blood+protein>). HPA provides estimated protein concentrations of proteins detected in human plasma based on immunoassays and mass spectrometry-based proteomics in The Human Plasma Proteome chapter (added to ST1 for reference). Using this data,

we checked whether the MSPA score of the UKBB and deCODE cis-pQTLs correlate with the abundance of the protein in question and found no correlation (See **Figure S11**; Spearman $p > 0.05$ in both cases).

Point 10.

The MSPA explanation could be worded more clearly, particularly given its central importance as a Method. I think the authors are using this as measure of confidence in whether there is a genetic (pQTL) association but the text says “This score is designed as a proxy for the likelihood of a protein expression signal being observed in the MS proteomics data”. ‘Expression signal’ is too vague and could imply whether the protein is detected. I would also suggest moving the technical detail from Results to Methods and using formula/equations to improve clarity.

Response: We agree and reworded the definition of the MSPA score in the text as follows:

“The MSPA score is defined as the fraction of peptides that support a pQTL at an alpha level of 1%, weighted by the number of detections of each peptide (see Methods). An MSPA score of one implies that for all detected peptides the 99% confidence interval of the effect (beta) does not contain the null, and thus support a protein expression QTL, while an MSPA score of zero indicates that there is no statistical support for an association by any of the peptides derived from that protein at all.”

We also added a formula-based definition to the online methods:

Definition of the MSPA score. We define the MSPA score of a pQTL as follows:

$$\text{MSPA score} = \sum_{i=1}^{k_{\text{pep}}} \frac{n_i}{n_{\text{tot}}} \cdot \delta(c_i^{\text{upper}} \cdot c_i^{\text{lower}} > 0)$$

where k_{pep} is the number of different peptides that have been detected for a given pQTL protein, n_i is the number of samples in which a peptide i from the given protein has been detected, $n_{\text{tot}} = \sum_{i=1}^{k_{\text{pep}}} n_i$ is the total number of individual peptide detections, c_i^{upper} and c_i^{lower} are the upper and lower 99% bound of the confidence interval for the effect size of the genetic association of the pQTL variant with peptide i , and

$\delta(\text{condition})$ is a function that takes a value of one if the condition in its argument is true and zero otherwise.”

Point 11. (major)

Table 1: previously unreported pQTLs. It would be helpful to add a column to indicate whether previous pQTL studies have measured this protein. I.e. is the novel pQTL because the protein was not tested in pQTL studies previously, or it was tested on a different proteomics platform and there was no significant association?

Response: In the revised version we report the reasons why a pQTL was not found in deCODE and UKBPPP (Table S2). As requested, we also added the relevant information as a column to the revised Table 1. We summarize the results as follows:

“A total of 35 replicated pQTLs (34.3%) were novel (Table 1), although eleven of the novel MS pQTLs had been assayed by the deCODE and/or UKBPPP studies but did not reach genome-wide significance at these loci, suggesting that the respective affinity assays may be targeting different isoforms or having other issues.”

Note that we ruled out differences in genetic architecture as we also consider MS pQTLs that are merely in the vicinity of previously reported affinity pQTLs as not novel.

Point 12a. (major)

“While we reported many novel pQTLs, about one third of the identified pQTLs, we opted not to highlight any new biologically relevant findings.”

This seems like a strange comment. In particular, a key analysis is to overlay disease GWAS associations onto the pQTLs detected here and present some summary of these findings, even if the authors choose not to discuss particular examples in details.

Response: We apologize for this shortcoming. We did overlay disease GWAS associations onto the pQTLs detected using PhenoScanner but initially chose not to highlight these in the paper. We understand that this omission gave our paper somewhat a feel of a technical report. We hope to have remedied this

*shortcoming by now reporting relevant disease overlaps in **Table 1**. We now highlight new findings of biological relevance in the main manuscript as follows (reviewer #2 raised a similar point and we reproduce our response there):*

“The 35 novel pQTLs overlap with several loci of biomedical relevance, including a trans-pQTL at the COLEC11 loci with ANGPTL6 that associates with LDL-cholesterol levels, a trans-pQTL at the CFH locus that implicates the TNFRSF1A modulator BRE in age-related macular degeneration and IgA nephropathy, a cis-pQTL for galactosylceramidase GALC with inflammatory bowel disease, and many others that can now be taken into consideration in the development of drug targets for these diseases.

We also found pQTLs that complement findings from affinity pQTLs, like an Olink *trans*-pQTL for fucosidase FUCA2 (rs11155297) that we replicate and extend by the “missing” *cis*-pQTL for FUCA1 (**Table S2**). Interestingly, our analysis further revealed that the strongest association with SOMAscan at this locus is with the mannosidase MAN2B2. FUCA1, FUCA2, and MAN2B are all enzymes involved in the lysosomal degradation of glycoproteins and glycolipids. Our Open Targets lookup additionally revealed a GWAS signal for “Total PHF-tau (SNP x SNP interaction)” (p-value = 2×10^{-8} [PMID:32450446]. While speculative, these genetic signals obtained on three different proteomics platforms provide a starting point for further investigations into the role of glycoproteins in Alzheimer’s pathology, and more generally outline how pQTLs can be used to generate hypotheses for the drug discovery process.”

*We further added new cases of biological interest, i.e. a GTEx splicing QTL seen at the peptide level (SPINK5, see new **Figure 4**) and a pQTL with a population specific pQTL signal and relevance for Crohn’s disease (ITLN1, see new **Figure S5**)*

*We also added the discussion of a pQTL that reflects an over-lapping splice QTL (SPINK5, see new **Figure 4**) and a pQTL with differing genetic signals between the cohorts (ITLN1, new **Figure S5**). These examples should demonstrate the biological relevance of the underlying pQTLs.*

*Finally, we extended the annotation of all pQTLs using the Open Targets API, including the pQTLs that we evaluated for epitope effects from the deCODE and UKBPPP study (**Tables S9-S10**).*

Point 12b. (major)

Moreover, an analysis examining the disease-relevance of the likely ‘epitope effect’ artefactual pQTLs vs the ‘real’ pQTLs would be highly informative. Which are more enriched for disease associations? If it turns out that the ‘epitope effect’ pQTLs are, this would suggest further interesting biology mediating disease through alternative protein forms rather than through quantitative genetic regulation of protein abundance.

Response: We added Open Targets disease associations to the respective tables that allows examining the disease-relevance of the pQTLs on a case-by-case basis as a function of descriptors like the MSPA score and peptide level derived parameters, like the number of conflicting directionalities. The addition of the SPINK5 example is a result of this effort. However, we did not find any enrichments either way.

Point 12c.

Supp Table 10. How many of the putative ‘epitope effect pQTLs’ are predicting to be damaging to protein function (e.g. the authors did a look up using Ensembl already so this data should be easily obtainable)

*Response: We added variant effect annotations and the caddRaw and caddPhred scores from Open Targets for all deCODE and UKBPPP pQTLs to **Tables S9-S10** for reference. We did not observe any significant difference in these scores between the top ranking pQTLs with a low (<0.2) and a high (>0.8) MSPA score.*

A genome-wide association study of mass spectrometry proteomics using a nanoparticle enrichment platform

Point-by-point response to the reviewers' concerns

(reviewer comments are boxed, our responses are in italic)

Reviewer #1 (Remarks to the Author):

The authors have made several significant improvements to the manuscript, including updating the analyses flow and adding a meta-analysis based pQTL discovery approach. The authors have furthermore elegantly addressed the major and minor points I raised in the previous round of reviews. I have no additional questions or concerns.

Response: Thank you for this positive evaluation.

Reviewer #2 (Remarks to the Author):

The paper by Suhre et al. clearly improved during review and raises an important point about the potential artificial nature of pQTLs based on affinity reagents. However, the paper still falls short demonstrating the overall relevance, including biological implications (see comments below). If this paper is kept as a letter, and emphasis is on a technical aspect, I am not sure what it delivers over and above preceding publications by the authors and others.

Response: We understand that we may have put too much emphasis on the technical aspects of potential artifacts of affinity pQTLs, thereby falling short of demonstrating the overall relevance and biological implications of our findings. We hope to have remedied this shortcoming in the revised version.

Here is what we believe our paper delivers over and above preceding publications:

Affinity proteomics GWAS have identified many pQTLs of biomedical relevance that can facilitate drug target development. However, these technologies alone cannot distinguish between genetic variants that change antibody and aptamer binding, generate different protein isoforms or induce genuine changes in protein abundance. Affinity proteomics GWAS are also limited to proteins for which suitable aptamer or antibody binders exist. While affinity platforms are hugely successful in large-scale pQTL studies, mass spectrometry proteomics is still considered by many as the method of reference.

This is the first population-based pQTL study using mass spectrometry at scale with one of the most comprehensive MS platforms currently available, employing five different nanoparticle enrichments to reach – for MS platforms – unprecedented proteome coverage.

We demonstrate its complementarity to affinity-based technologies, revealing their strengths and also their limitations. We show that around one third of the proteins covered by MS are currently not

accessible to affinity proteomics, but we are also clear about the fact that another third of the proteins are accessible to affinity proteomics only.

We further show that up to 30% of the affinity pQTLs can be replicated by MS proteomics, increasing their value in the drug target evaluation process, while we also show that another 30% do not show any sign of replication in our study, despite being sufficiently powered. These pQTLs may be affected by epitope effects and should be further investigated before using these pQTLs in Mendelian randomization studies, for instance. Upon the request of reviewer #3 we added a new Table 2 that highlights these pQTLs.

The UKBPPP consortium just announced that it will analyze up to 600,000 more samples of UKB, while deCODE will run 50,000 UKB samples using the SOMAScan platform. Our paper demonstrates where and how MS proteomics can complement these large-scale studies and addresses important questions asked by the community.

Having participated in both large affinity pQTL studies with Somalogic and Olink (Sun et al., Nature 2018; Sun et al., Nature 2023), and also as a member of the UKB ISAB in the discussions around further cohort phenotyping, I (K.S.) know that the larger community of researchers in the pharmaceutical and biomedical research field, both commercial and academic, is highly interested in learning more about the strengths and limitations of nanoparticle-enrichment MS-based proteomics GWAS, and we are providing just that in our paper.

There is still also a tendency of a somewhat biased comparison with affinity reagents.

Response: We regret if we generated the impression of being somewhat biased against affinity reagents. Our intention was to avoid any kind of platform comparison, as we strongly believe that MS and affinity approaches are complementary rather than exclusive.

Please note at this point that our paper is the result of an academic Cornell-Harvard collaboration, and that the academic authors have no commercial interests or specific stakes in MS-proteomics. Seer provided the measurements as a service. We included their scientists as co-authors for their scientific contribution in the interpretation of MS-related observations.

To respond to the reviewer's concern, we revised the paper in multiple places to avoid all comparisons that may appear biased against one platform or the other.

That is, little is done to understand to what extent the more than 50% of discovered but not investigated proteins suffer from similar 'epitope-like' effects due to a change in affinity to the used nanoparticles (see comments).

Response: The reviewer raises a new point here, which is the effect of genetic variation on protein extraction during sample preparation, i.e. the nanoparticle enrichment step.

We agree that in principle similar 'epitope-like' effects due to a change in affinity to the used nanoparticles can occur.

However, binding of proteins to nanoparticles is not as specific to epitopes as it is in the case of affinity reagents but rather depends on the larger electrostatic, hydrophobic and hydrophilic properties of the

exposed protein surfaces. A single amino acid exchange is unlikely to have a major impact on this scale in most cases, although such an effect is conceivable.

Indeed, all MS-protocols that use a protein-enrichment or -depletion step are in principle prone to effects of genetic variants that change protein properties relevant to the protein extraction efficacy. Therefore, some associations based on technical rather than biological effects are to be expected.

To address this point, we conducted two new analyses: First, we computed the pQTL summary statistics for all 364 discovered pQTLs in all nanoparticle runs (new Table S15) and second, we conducted a GWAS with protein missingness on all proteins in all nanoparticle runs that were detected in more than 5% of the samples (new Table S16).

Replication of a pQTL in two or more nano particle runs would support (but not prove) the hypotheses that the pQTL is not impacted by a genetic effect affecting nanoparticle specificity. A total of 187 pQTLs (51.4%) had a significant association on a second nano particle ($p < 0.05/364/4$).

We understand that with the “50% of discovered but not investigated proteins” the reviewer refers to those proteins that were excluded from our GWAS due to their high missingness.

We believe that the reviewer is suggesting that we conduct an analysis on protein missingness, under the hypothesis that a protein would be missing if it was affected by an ‘epitope-like’ effect?

A challenge of this argumentation is that protein missingness can also be related to lower protein levels impairing protein identification. However, in order to be complete, we conducted such a GWAS on protein missingness (see new paragraph in the methods section).

We identified associations with missingness at a significance level of $p < 5 \times 10^{-12}$ at 520 loci, 329 of which were with only one of the following five proteins: MS4A14 (253 loci), HLA-C (25 loci), CD79B (20 loci), SDF2 (17 loci), and IGKV1D-13 (14). Missingness of most of these proteins was also associated strongly with self-reported race (black vs. white), suggesting that their association is confounded with race. Of the 191 remaining missingness-pQTLs, 124 were in-cis, including 27 of our 177 cis-pQTLs (15.3%). For these pQTLs a potential ‘epitope-like’ effect can therefore not be entirely ruled out, although we identified several pQTLs that are clearly driven by biology (Table S16).

We added the results of the missingness GWAS and the associations with other nanoparticles to the Supplementary Tables, allowing for further scrutiny of individual pQTLs on a case-by-case basis.

We included the following text to acknowledge this limitation to the interpretation of MS proteomics pQTLs:

“It should be acknowledged that nanoparticle-enriched MS-proteomics methods are not free of potential ‘epitope-like’ effects. Although less likely due to their less specific electrostatic and hydrophobic interactions, protein – nanoparticle binding can in principle be modified by genetic variation. Genotype-associated missingness could potentially indicate such effects, while associations supported by more than one nanoparticle run makes them less likely. We found that 15.3% of the proteins involved our pQTLs had a missingness-pQTL while 51.4% of our pQTL associations are supported by more than one nanoparticle run (see Supplementary Text).”

Rebuttal (reviewer 2)

General remark. The colocalization strategy to establish trans-ancestral sharedness of genetic signals is somewhat odd, in a sense that colocalization is highly reliant on a conserved LD-backbone across cohorts. While it resolves, to some extent, the issue of missing SNP overlap, it will fail if haplotypes are fragmented across ancestries. An alternative, less sophisticated but hopefully more stable strategy, would be to compute all proxies (say $r^2 \geq 0.1$) for each lead pQTL from two different studies, and quantify the overlap of SNPs using in-study LD for each. One may call ‘replicated’ pQTLs those that share one or more SNP.

Response: We agree that colocalization is highly reliant on a conserved LD-backbone across cohorts and that our approach might not consider a pQTL as replicated if haplotypes are fragmented across ancestries. We thereby took a conservative approach, not calling a pQTL replicated if it did not share a conserved LD-backbone.

However, we did not lose these pQTLs, but flagged them as being supported by different genetic signals, writing:

“Also, 14 of the non-replicated pQTLs had a significant but different genetic signal in both cohorts, such as intelectin-1 (ITLN1).”

To fully address the reviewer’s concern, we now also implemented their suggested alternative replication strategy (see extended methods and results in columns appended to Table S2). We found that, using the alternative method, 115 pQTLs could be considered as replicated, while our approach only considered 102 pQTLs as replicated. For 14 out of 29 pQTLs that were replicated by the alternative method but not by our approach coloc suggested the presence of a different (H3) signal between the studies, which is consistent with what reported in the paper.

As both approaches yield comparable results, we prefer flagging pQTLs that are supported by different genetic architectures rather than considering them as replicated and therefore kept our method as the reference, providing the results of the alternative approach in the supplement.

Point 11. While I agree on the ability to trace MS performance via protein content from different nanoparticles, the effect of efficacy of the nanoparticles on the type and abundance of assayed proteins is not well captured, but important for our understanding, including false-negative findings. For example, while the authors rightly flag non replicating results for strong cis-pQTLs based on affinity reagents, I find it equally likely that some associations are not replicated as expected due to poor or very variable efficacy of the nanoparticles to extract a given protein. One might even speculate that PAVs also affect binding to nanoparticles and that the issues claimed to be resolved here still persist.

Response: We address the effect of PAVs on nanoparticle binding above, please see there.

Point 14. I like the additional information on the biological relevance of identified pQTLs a lot but wonder whether the current presentation is either too simplistic or needs toning down in wording. For example, several trans-pQTL findings are highlighted, but we don’t know whether these findings are the result of horizontal pleiotropy by which the genetic variant affects protein and disease independently of each other, or whether, as suggested by the authors, proteins indeed mediate disease associations.

Response: We agree that trans-associations with disease can be the result of horizontal pleiotropy. We did not suggest that our data shows that these proteins mediate the disease associations, but merely raise the possibility that they do, pending further investigations. This caveat applies to all conclusions drawn from pQTL studies. Even in Mendelian randomization studies the absence of horizontal pleiotropy is an assumption that needs to be established with other means than the genetic association data.

In particular the reference for drug target discovery is exaggerated.

Response: Respectfully, we do not share the reviewer's view that our reference to drug target discovery is exaggerated. The UKB PPP consortium invested twenty million dollars in the measurement of the first 50,000 UK Biobank samples to generate the Olink pQTLs for the purpose of drug target discovery. They recently decided to further extend the Olink study to all available 600,000 samples (baseline and imaging repeats). At the same time, deCODE announced that they will run the first 50,000 UKB samples on the SOMAscan platform.

These investments reflect the value that pharma companies attribute to pQTLs. From confidential discussions with UKBPPP members, I (K.S.) understand that there are already new targets in which some of these pharma companies currently invest heavily and that would not have been considered without pQTL support from the UKBPPP and deCODE studies. In the long term, these efforts shall lead to new treatment options.

Our study provides an orthogonal method to evaluate pQTLs reported by affinity proteomics and also to discover new pQTLs on proteins that are not accessible to Olink and SOMAscan technologies and thereby responds to a central question in the field.

Nevertheless, as not everyone may agree with our point of view, we toned-down reference to drug target discovery at several places, in particular by rewriting the abstract (see track-changed manuscript).

Also, how is an association with FUCA2 a cis-pQTL for FUCA1, isn't this more of an example of poor peptide mapping in the DIA-NN workflow?

Response: Our explanation of the FUCA2 cis-pQTL was not clear. We now write:

"We also found pQTLs that complement findings from affinity pQTLs, like an Olink trans-pQTL for fucosidase FUCA1 (rs11155297) that we replicate. In addition, we found a cis-pQTL for FUCA2 at the same locus, which was not assayed by the Olink platform, but would be expected to explain the trans-association".

The reviewer brings up a more general concern in this context, suggesting that this might be "more of an example of poor peptide mapping in the DIA-NN workflow?".

In order to address this point, we added a peptide mapping for all 1,980 analyzed proteins as a resource to Table S1. While DIA-NN is a widely used workflow in the proteomics community, a reader questioning the mapping of a specific protein of interest can now use this table as a reference.

For instance, for FUCA1 and FUCA2 there are 8 and 16 protein specific mapped peptides, respectively. Note that no peptides map to both proteins at the same time (no homology). The average number of peptides identified per sample for FUCA1 and FUCA2 is 3.7 and 10.2 in Tarkin and 4.4 and 11.6 in QMDiab (see excerpt of Table S1 and Supplementary Data S6 below).

We are therefore confident that this case is not an example of poor peptide mapping in the DIA-NN workflow. This new table and data set will also allow readers to check all other pQTLs for their mapping quality.

	B	O	AN	AO	AP	AQ
3	protein_ids	gene_name	SORT Mapping	N peptides	peptides / sample Tarkin	peptides / sample QMDiab
23	P04066	FUCA1	20	8	3.7	4.4
48	Q9BTY2	FUCA2	45	16	10.2	11.6

Legend: Excerpt of Table S1, showing the total number of identified peptides and the average number of mapped peptides per sample for FUCA1 and FUCA2 (only selected columns are shown here).

Legend: Excerpt of the new Supplementary Data S6, visualizing the mapping of peptides to the FUCA1 and FUCA2 protein sequences; Grey: detected in QMDiab; Green: detected in Tarek.

The number of samples in which FUCA1 and FUCA2 specific peptides were detected in Tarkin (green) are plotted against the amino acid position on their protein sequences. The number of samples in which these peptides were detected in QMDiab are plotted along the lower part of the y-axis (grey).

Overlapping boxes indicate peptides that were detected in multiple precursor charge states or that correspond to peptides with missed cleavages. Similar plots and the corresponding plot data (peptide level detections) are provided for all 1,980 analyzed proteins as new Supplementary Data S6 and Table S14.

The link to AD is vastly exaggerated since there is no association with AD (despite well-powered studies) at this locus. The examples would benefit from expert review.

Response: We did not claim an association with AD. This locus is an association reported in the GWAS catalog with PHF-tau measurement. We chose to highlight this association as a use-case where all three platforms contributed information, in this case linked to protein glycosylation, and to outline how this information can be used to create new hypotheses on a GWAS catalog hit, in this case an association with “Total PHF-tau (SNP x SNP interaction)” on the same (coding) variant. We wrote:

“While speculative, these genetic signals obtained on three different proteomics platforms provide a starting point for further investigations into the role of these glycoproteins in Alzheimer’s pathology, and more generally outline how pQTLs can be used to generate hypotheses for the drug target discovery’ process.”

We do not feel that this paragraph constitutes an exaggerated claim, given the suspected role of tau in AD pathology.

Point 24 The separation of proteins across platforms is interesting, but can the authors please clarify, why cytokines should be underrepresented by the Seer technology?

Response: Cytokines are low abundance proteins and are therefore easier to measure using affinity proteomics approaches. There may also be an over-representation of cytokines on the affinity platforms, as these proteins are of high biomedical interest and might therefore have been given priority by platform developers. The competitor of Olink, Alamar, actually makes a point of focusing on cytokines as challenging targets for other platforms due to their low abundance.

Paper

1) Picky, but ‘drug discovery’ should be replaced by ‘drug target discovery’ and even from this point it is a long way to go, since in GWAS we cannot easily (or at all) establish the exact disease mechanisms or judge whether intervening on the target will reverse the diseases. More humble language should be used throughout, in particular since the authors present at best very weak evidence for the biological relevance of pQTLs.

Response: We replaced ‘drug discovery’ with ‘drug target discovery’ and used more humble language throughout (see change-tracked version).

2) Second sentence in the abstract: I am a bit tired of people claiming MS-based proteomics being ‘unbiased’, there are at least two steps that will make this platform selective: 1) the affinity of proteins to certain nanoparticles (some reliably measured proteins with affinity proteomics are not at all present in the data presented), and 2) the behavior and number of peptides measured in the MS to later on ‘infer’ protein abundances from the original plasma sample. Those limitations should be clearly highlighted. There is a reason why affinity-based assays are still the technique of choice for most clinical chemistry assays used in clinical routine.

Response: We fully agree that MS-based proteomics is not unbiased, and stated this point in the discussion section already:

“While MS proteomics is not biased versus any particular set of pre-selected proteins, it is biased towards protein isoforms that are present in the utilized database, proteins that are enriched using one of the five nanoparticles, and proteins that can be cleaved into peptides and are detectable by the applied MS proteomics method.”

To further clarify this point, we rewrote the abstract, removed all wording that could suggest that MS proteomics is unbiased (see track-changed manuscript), and addressed peptide mapping and ‘epitope-like’ effects as described in our responses above.

3) Introduction: The efforts by deCODE and UKB-PPP are certainly stunning, but this neglects a large body of very influential studies published beforehand. Instead of the sample size or protein coverage, the authors would be better off clearly demonstrating the usefulness of such studies.

Response: Respectfully, we believe to have paid credit to all key pQTL studies published beforehand (refs 6-15) and provide references that cover the usefulness of pQTL studies (refs 1-5) in the first paragraph. We focus in the introduction on the efforts by deCODE and UKB-PPP as we particularly evaluate their pQTLs later in the paper. We therefore do not feel that we are neglecting a large body of very influential

studies published beforehand. However, to be complete we added a reference to a recent review that lists all published pQTL studies, by adding:

“The development of high-throughput capable affinity proteomics platforms has spurred the conduct of an ever-increasing number of GWAS with protein traits (see Sun et al., Mol Cell Proteomics, 2024 for a comprehensive list).”

Why should we invest in even more and more expensive techniques? In particular, since there is a growing body of evidence that molecular QTLs are actually depleted for phenotypic consequences.

Response: One reason why molecular QTLs might be depleted for phenotypic consequences could be that they are enriched in genetic variants that affect the measurement assay but have little or no biological consequences. This issue can be addressed through confirmation of pQTLs on orthogonal platforms, which is a central objective of our work.

4) Also why do the authors claim that MS is ‘immune’ against epitope effects. It is great to have multiple peptides, in case they uniquely map to a protein, but if the PAV changes affinity to the nanoparticles a similar thing will happen. This effect is currently somewhat ignored in the presented data, given that only proteins with >80% valid values are evaluated.

Response: We agree that MS is not immune against epitope effects but argue that such effects are less likely to have a major impact, given that nanoparticles interact with proteins on a larger physical scale, where single PAVs play a smaller role. Nevertheless, we performed further analyses to address this point (see above).

5) Given the inherent variation in day-to-day and even more so lab-to-lab performance of MS instruments, I wonder to what extent the low replication rate is also explained by missing some key peptides for proteins of interest. Can the authors at least confirm that a similar type and number of peptides was found between both studies? This would be in particular important to investigate for findings such as ITLN1.

Response: Please note that our replication rate agrees with the power analysis. We write:

“The primary reason for the non-replicated pQTLs is the limited size of the replication cohort: Of 70 pQTLs that had 80% replication power, 58 (82.9%) replicated and most (36 out of 38) of the nominally significant ($p < 0.05$) pQTLs had concordant directionality, suggesting that most of the unreplicated pQTLs should be replicable in future larger-scale studies.”

However, the reviewer still raises several important questions here that we now address as follows:

The variability in LC-mass spectrometry (LC-MS) performance, both from day to day and across different laboratories, is a well-known issue due to variations in sample preparation protocols, liquid chromatography (LC) instruments, gradient lengths, and MS instruments/protocols. However, it is important to note that in both of our studies, all samples were analyzed at the same Seer laboratory using a standardized, fully automated workflow and were completely randomized within each study to ensure consistency.

To confirm that a similar type and number of peptides was found between both studies, we added an analysis of the correlation between the peptide and protein intensities, their standard deviation, and the average peptide coverage of the analyzed proteins between Tarkin and QMDiab (see new Figure S15).

The average intensities of the 1,980 analyzed proteins correlated with $r^2 = 0.78$ (slope = 0.84), their standard deviations correlated with $r^2 = 0.69$ (slope = 0.71). The average concentrations of the individual peptides that map to these proteins correlate with $r^2 = 0.61$ (slope = 0.76).

Furthermore, we provide the mapping of all peptides to the respective proteins as new Supplementary Dataset S6 and Table S14 (see also our responses to FUCA1/2).

For ITLN1 this analysis confirms that a similar type and number of peptides was found between both studies, albeit with less coverage in QMDiab, which is to be expected, given the lower sample size of that study.

Legend: This plot shows the number of samples in which ITLN1 specific peptides were detected in Tarkin (green), plotted against the peptide position on the ITLN1 protein sequence. Detections in QMDiab are plotted in the opposite direction (grey). Overlapping boxes correspond to peptides that were detected in multiple precursor charge states or that correspond to peptides with missed cleavages. Similar plots and the corresponding plot data (peptide level detections) are provided for all 1,980 analysed proteins as new Supplementary Data S6 and Table ST14.

6) Does the newly conducted MA that now complements the MSPA score take differing peptide intensities of the same protein target into account?

Response: Yes – the meta-analysis integrates the summary statistics from all individual peptides. Peptides with lower intensities generally have weaker statistical support and thus contribute less to the meta-analyzed effect sizes and p-values. Please note that observing differing peptide intensities for a same protein is normal, due to different physico-chemical properties (e.g. hydrophobicity) and ionization propensities of the peptides, that depend on the number of basic AA per peptide.

7) The construction of the MSPA still appears a bit odd, maybe flawed statistically. Controlling each effect estimate for individual peptides at a one percent level will not ensure a 1% level for the overall protein quantification. However, a general pragmatic approach is acceptable, but the authors may want to avoid any reference to statistical significance. They should also provide a criterion at which point the MSPA starts giving reliable results, i.e., I assume that a minimum number of peptides should be requested.

Response: This point is covered in our response to point 9). Please see there.

8) The new SPINK5 example is possibly the most appealing one added during review, but I am wondering whether this also points to a potential flaw in the analysis of pQTLs using mass spec data, if certain peptides are omitted from quantification, as they may introduce peptide isoforms that cannot be mapped to protein groups later on.

Response: We are pleased to learn that the addition of the meta-analysis on the peptide level added novel insights to the paper and the SPINK5 example reveals a key strength of MS, that is, access to isoform information.

We are not clear why the reviewer assumes that “certain peptides are omitted from quantification”. This is not the case. The data analysis of the raw MS-data was performed using the standard DIA-NN pipeline, which includes peptide identification, quantification, mapping to protein groups and protein quantification. All our analyses are downstream of that pipeline, no peptides were omitted.

The new Table S15 now explicitly lists all analyzed peptides together with the protein groups to which they were mapped by DIA-NN for reference.

We also agree that isoform identification depends on the number and quality of the detected isoform specific peptides, and its success will depend on the specific case, and also on the current limits of the existing tools, like DIA-NN.

However, this should not be regarded as a potential flaw of our paper, but represents the current state of the art, calling for future development of specific analysis software – similar to the challenge of detecting splice-forms from RNA seq data.

It would also be important to provide a locuszoom plot incorporating the sQTL information from GTEx in skin.

Response: We added a locus zoom plot for the GTEx sQTL together with one for the PLINK5 pQTL as new Figure S16, showing that both associations are driven by the same genetic signal.

9) Figure 3: Please remove the suspected curve or provide a sound statistical fitting of a model. It is also completely unclear to me, how the authors infer ‘power to replicate’ from this figure. Power to replicate will depend on effect size and frequency of the allele (plus measurement certainty) and can be computed for each of the pQTLs tested.

Response: Following the reviewer’s suggestion, we removed the suspected curve and conducted a power analysis, described in a new methods section. A total of 120/167 pQTLs had >99% probability of reaching a multiple-testing corrected p -value of $p < 0.05/319$ (deCODE) and $p < 0.05/392$ (UKBPPP) in Tarkin.

Consequently, we extended our analysis to 120/167 pQTLs in deCODE/UKBPPP, respectively, which had sufficient power to be replicated in Tarkin (previously we analyzed the 100 strongest pQTLs per study).

Of these sufficiently powered pQTLs, 39/49 (32.5%/29.3%) replicated and had an MSPA score greater than 0.8 in deCODE/UKBPPP, respectively, while 36/55 (30.0%/32.9%) did not replicate and had an MSP score below 0.2 (Table S9/S10).

These findings agree with our previous analysis, suggesting that 30% of sufficiently powered pQTLs reported by either affinity platform can be replicated using the MS platform, 30% cannot, and the remainder fall into the grey-zone.

Legend: Updated Figure 3 with the suspected curve removed and new panels (a) and (e) showing calculated power to replicate instead.

10) There are strong inferences drawn based on a set of as few as 46 pQTLs without examining any bias in terms of what proteins those are (likely high abundant and actively secreted) and hence any reference to 'gold-standard' should be avoided. Some of the inference is also circular, if the very same variant had been selected for SomaScan and Olink it is obvious that effect estimates are highly concordant, otherwise those variants were not selected in the first place.

Response: We were not clear about the argumentation why we focused on the 46 pQTLs: the effect estimates are only expected to be concordant if both platforms measure a protein's abundance, but not if one of the platforms is impacted by an epitope effect, as there is no reason to assume that aptamers and antibodies bind to the same epitope, and if they do, to respond in the same way with the same effect. Only very few of the sufficiently powered proteins in this set had low MSPA scores, in contrast to the entire set of sufficiently powered pQTLs. We argued that this supported the interpretation of the "suspected" MSPA curve. We now understand that this argumentation was not very clear.

In the revised manuscript we followed the reviewer's suggestion and now use a power analysis instead of the assumed MSPA curve, making this argumentation redundant. We now merely report the 46 pQTLs as a reference set where a strong pQTL is observed on the same SNP on both affinity platforms and refrain from calling it a gold-standard.

Note that in our past experience with metabolomics QTLs we found a reference set of strong validated genetic associations with metabolites to be useful as a benchmark in the evaluation of the performances of new platforms. We believe the same holds for proteomics.

11) It is unclear to what extent the OpenTargets look-up of pQTLs does represent a tangible phenotypic follow-up. For examples, it is unclear from the methods whether the look-up comprised the same variant across stored data sets, or does indeed refer to the table listing GWAS regional lead signals, including an estimation of the LD between the pQTL and GWAS lead signal. If the latter, what cut-off was used to declare some level of colocalization?

Response: Open Targets represents in our view one of the most tangible phenotypic follow-up tools that are available at present. It constitutes a shared effort of several pharma companies to comprehensively (and freely) collect and annotate all available genetic findings from GWAS, including different omics-QTLs. We performed this lookup in the previous round of reviews upon suggestions by one of the reviewers.

The Open Targets lookup we report in our Supplementary Tables comprises both, same variant across stored data sets, and reference to GWAS regional lead signals using LD. We limited the latter to $LD\ r^2 > 0.7$ between the pQTL and GWAS lead signal (more can be obtained through the Open Targets web-site).

The corresponding columns in Table S2 are labeled “Associations ($LD\ r^2 > 0.7$)” and “Associations (same SNP)”. We ordered the associations by decreasing strengths of association (p-value) to make the strongest associations easily visible.

We expect someone with an interest in a particular pQTL to access the Open Targets website after having identified an association of interest in our study.

In the revised version we therefore added a column with hyperlinks to Open Targets to all pQTLs in Table S2. We also extended the methods section as suggested (see revised track-changed manuscript).

Reviewer #3 (Remarks to the Author):

The manuscript is significantly improved following the revisions. However, I have some remaining concerns.

Response: Thank you for your suggestions. We hope to have addressed all remaining concerns. Please find below our responses to the specific points.

1) The mapping of pQTLs onto disease and complex trait GWAS partially addresses a point I made in my previous review. Table 1 provides details of how the newly described pQTLs link to disease/complex trait GWAS associations. In their response, the authors report that they did not find any enrichment of either true pQTLs or likely epitope effect pQTL signals with respect to GWAS hits.

Despite the revisions, I did not feel much clearer as a reader how these new data should re-shape our thinking of the functional genetics underlying complex diseases and traits. The authors estimate that approximately one third of testable previously reported pQTLs may arise from epitope effects. However,

it was unclear how this translates into revising understanding of the mechanisms underpinning disease GWAS associations.

Response: How do these new data re-shape our thinking of the functional genetics underlying complex diseases and traits?

The central difference between affinity and MS proteomics is that the former provides a single value that is assumed to reflect the level of the considered protein in blood, while the latter provides a range of additional information that can be analyzed at the level of the raw mass spectrum, the identified peptides, or the thereof quantified proteins and protein groups.

While MS is considered as the method of choice for protein analyses, affinity proteomics, due to its relative simplicity and higher throughput capabilities, has dominated large scale pQTL studies.

We not only focus on the new findings of our paper (i.e. the new pQTLs reported in Table 1 and S2) but also cover relevant aspects that only arise when conducting MS proteomics at scale, such as the role of isoforms and epitope effects, which are questions that can only be addressed using MS approaches.

Thus, the data we generate here should re-shape the way we investigate pQTLs and how we generate hypotheses from them regarding the mechanisms underpinning overlapping disease GWAS associations.

For example, how many protein-disease associations identified by previous pQTL work are now in doubt?

Response: We identified 17 disease-associated pQTLs that are unlikely to be driven by genetically modified protein levels (see below, they are reported in new Table 2b). This implies that the disease association is likely driven by changes to the proteins' function rather than its abundance, which has important implications regarding the ways the pathways these proteins act on can be targeted.

Are there important Mendelian randomisation analyses that gave spurious positive results based on using pQTLs resulting from epitope effects?

Response: A total of 161 of 240 papers citing the UKBPPP (Sun et al., Nature, 2023) and 392 out of 485 citing the deCODE (Ferkingstad et al., Nature Genetics, 2021) GWAS papers include some aspects of Mendelian randomization. Most of these papers include multiple proteins in their analysis and are likely to include some of the pQTLs that may be affected.

Perhaps there were instances where previous pQTL-based MR gave unexpected results discordant from other observational associations (e.g. associations of a protein with incident disease risk) that can now be explained by epitope effects?

Response: We are not aware of any new such instances, but we already report one such case in our paper:

“Our analysis also confirms a previously reported epitope effect for pQTL of GDF-15 (ref), reported by deCODE (rank 71 in Table S9) and UKBPPP (rank 133 in Table S10), which has an MSPA score of zero in our study.”

I appreciate there may be some technical difficulties in quantifying the extent to which SNP-protein-disease links need to be revised because the GWAS Catalog has lumped diseases together with other

traits. Nevertheless, this could be done, and I would suggest that a disease-focused analysis would be most valuable given the authors pitch around drug discovery (i.e. ignore anthropomorphic and more esoteric traits).

Response: Via Open Targets we have already queried all GWAS catalog associations that overlap with all pQTLs from deCODE and UKBPPP (see Tables S9 and S10).

To address the reviewer's point, we now manually curated these associations and retained the strongest disease-relevant association at each locus, ignoring anthropomorphic traits, traits like "age at puberty" and "educational attainment", and all quantitative traits other than established disease markers (see Tables S13 and S17 and our responses below).

A very tractable starting point would be the list of pQTLs in Fig S13 (described by the authors as "high ranking" pQTLs based on protein measurements using Somalogic/Olink but with low MSPA scores). How many of these are also disease GWAS hits and how does the new data revise our previous understanding of the molecular mechanisms underlying disease risk at these loci?

Response: We followed the reviewer's suggestion. We now report 91 pQTLs (36/55 from deCODE/UKBPPP, respectively) with >99% power that were not replicated in Tarkin and had an MSPA score <0.2. Of these, 17 had a disease-relevant association in the GWAS catalog ($LD > 0.7$) (Table S13). We now further report 76 pQTLs (33/43 from deCODE/UKBPPP, respectively) with >99% power that replicated and had an MSPA score >0.8. Of these, 24 overlapped with disease-relevant GWAS associations (Table S17). We report the most salient details in a new Table 2.

2) "a trans-pQTL at the CFH locus that implicates the TNFRSF1A modulator BRE in age-related macular degeneration and IgA nephropathy"

CFH (complement factor H) is a highly pleiotropic locus. The complement system is strongly implicated in age-related macular degeneration and IgA nephropathy and complement targeting therapeutics have been successfully developed/are in late-stage development for both these conditions (e.g. iptacopan just approved by the FDA in IgA nephropathy). Why do the authors think TNFRSF1A is the key mediator of these disease associations rather than CFH itself or the many other trans-associated proteins at this pleiotropic locus? This seems very unlikely to me, particularly given the recent success of complement blocking drugs.

Response: This is a misunderstanding. The HGNC gene name of BRE is "brain and reproductive organ-expressed (TNFRSF1A modulator)". We referred to it as "TNFRSF1A modulator BRE", which apparently was confusing.

Our finding is that CFH is a trans-pQTL for BRE, not TNFRSF1A. This pQTL suggests that genetic variation in CFH causes variation in BRE and agrees with the reviewer's expectation that CFH is the disease mediator. Our findings further suggest (but do not prove) that this mediation may involve BRE downstream of CFH.

We rewrote this sentence as follows:

"..., a trans-pQTL at the CFH locus that implicates brain and reproductive organ-expressed (TNFRSF1A modulator) (BRE) in age-related macular degeneration and IgA nephropathy, ...".

3) I felt the writing was too casual/colloquial, and at times imprecise for scientific writing. While each instance might seem like a minor point, this issue was sufficiently frequent that there was a cumulative effect. Nature Genetics is the premier genetics journal and sets the precedent for the field.

Response: We followed the reviewer's specific suggestions (see below) and tried in general to render the writing less casual/colloquia (see track-changed version of the revised manuscript).

A non-exhaustive list of examples (PS it would have made reviewing much easier if line numbers were provided!):

- "Interestingly, our analysis further revealed that the strongest association with SOMAscan at this locus is with mannosidase MAN2B2."

-> SOMAscan is a platform. There cannot be an association "with SOMAscan". (The authors mean association with proteins assayed on SOMAscan)

Response: Reworded as: "Interestingly, our analysis further revealed that the strongest association of proteins assayed by SOMAscan at this locus is with mannosidase MAN2B2."

"Our Open Targets lookup ..." (imprecise/colloquial)

Response: Reworded as: "Lookup using the Open Targets platform additionally revealed ..."

"although eleven of the novel MS pQTLs had been assayed by the deCODE and/or UKBPPP studies "

-> a pQTL is a genetic association with a protein. It cannot be "assayed" - the protein is assayed, not the pQTL. See below re "MS pQTLs".

Response: Reworded as: "although eleven of the novel MS pQTLs for which the proteins had been assayed by the deCODE and/or UKBPPP studies"

"suggesting that the respective affinity assays may be targeting different isoforms or having other issues."

-> 'or having other issues' is very vague - what is meant by this?

Response: Reworded as: "... suggesting that the respective affinity assays may be targeting different isoforms, not reaching their detection limits or be binding off-targets"

"...and many others that can now be taken into consideration in the development of drug targets for these diseases. "

-> How many others? It would be better to enumerate.

Response: Reworded as: "... and fourteen others that can now be taken into consideration in the development of drug targets for these diseases."

"The pQTL has an MSPA score of 0.5, placing it into the "grey zone". "

-> I understand what is meant, but this is too colloquial.

Response: Reworded as: "The pQTL has an MSPA score of 0.5, giving it support by some, but not all analyzed peptide associations."

" Roughly, the first 100 pQTLs of each study appear to have sufficient power to replicate in Tarkin. "
-> feels imprecise

Response: We now provide a power analysis and report precise numbers (see our response to reviewer #2").

Minor

Table 1. a) Presumably the first column should say "Protein" and not "Gene". b) The authors should indicate cis vs trans. c) Need to define columns more clearly in the legend.

Response: a) yes, corrected to Protein; b) see column "Locus"; c) we added a definition of the columns to the legend of Table 1.

I do not think the language of "affinity pQTLs" or "MS pQTLs" is appropriate.

Fundamentally, a pQTL is a biological entity. Underpinning the association between locus and protein level is a causal genetic variant. That association has a biochemical mechanism (e.g. an allele in a SNP in a promoter that alters transcription factor binding which in turn impacts mRNA and thus protein levels, or a protein-altering variant that affects secretion from the cell). Thus, something either is or isn't a pQTL. The authors can refer to pQTLs reported based on using affinity-based proteomic measurements and they can flag a subset of those as likely artefacts. However, the definition of a pQTL cannot be changed to make it measurement-system dependent: a variant either is or isn't a pQTL.

ditto for "Olink pQTLs" and "SOMAscan pQTLs" - Olink is a commercial company.

Response: We agree - wherever we use a term like "affinity pQTL" we should say "a pQTL identified on an affinity platform" to be precise. However, this would make the writing very cumbersome, as there are many occurrences. We believe that what we want to say is clear and also switched to the longer wording in some places. We are of course open to changing this wording throughout to the more extensive form, should the journal prefer us to do so.

"We found a strong ITLN1 signal in the QMDiab cohort that is not present in Tarkin, which may be discernible only in that cohort due to differences in lifestyle (Figure S5)."

I could not find any data to support the claim that this was due to lifestyle. Was there any?

Response: We provide here a plausible explanation for why we found a stronger signal in the smaller replication cohort on a well-established Crohn's disease locus, suggesting possible gene-environment interactions that may be worth further investigation. To make this clearer, we now write:

"... which may be discernible only in that cohort due to differences in lifestyle, environmental factors or population-specific genetic background"

NB in Figure S5 legend this claim is repeated: discussion is not appropriate for a Fig legend.

Response: We removed the discussion from the legend.

" and 67 (65.7%) had been reported at least once by these and/or other pQTL studies covered by Open Targets "

-> change "covered" to "curated by" or "included in the OpenTargets database"

Response: Changed to "curated by"

"We believe that the MSPA score is a more intuitive measure..."

-> Avoid "We believe..." (this is science, not religion!). Perhaps "We argue..." or "We propose..."

Response: Changed to "we argue"

"GWAS further suggest a role for SPINK5 in lung function phenotypes (COPD, FEV) and pancreatitis."

-> More precise to say GWAS have identified associations between SPINK5 and the traits listed.

Response: Changed to "GWAS further identified associations of SPINK5 with lung function phenotypes (COPD, FEV) and pancreatitis"

"that is, cases where a genetic variant interferes with the affinity binding, but at the same time affects protein expression via some biological feedback mechanism."

-> What was the evidence for the statement regarding this putative feedback mechanism"?

Response: Changed to "that is, cases where a genetic variant interferes with the affinity binding, but at the same time affects protein abundance via some biological pathway."

"While MS proteomics is not biased versus any particular set of pre-selected proteins"

-> Yes it is not biased to human curation of a set of proteins, but it is biased to higher abundance proteins.

Response: By referring to proteins that "are detectable by the applied MS proteomics method" we implied a bias towards higher abundance. To be clearer, we now added this phrase to proteins that "are detectable by the applied MS proteomics method, such as highly abundant proteins."

The authors refer to protein expression. Actually, since they are measuring plasma proteins, expression is not necessarily accurate. Protein expression is intracellular and proteins then reach the circulation in different ways (e.g. secretion, trafficking to cell membrane and cleavage, and others). So protein levels or protein abundance would be more correct.

Response: We changed "protein expression" to "protein abundance" throughout

Manuscript NG-LE65652:

A genome-wide association study of mass spectrometry proteomics using a nanoparticle enrichment platform

Response to the Reviewers' Comments:

Reviewer #2 (Remarks to the Author):

I deeply appreciate the additional effort done by the authors to address my concerns, which helped to clarify technical details. The major finding of the study, the non-replication of previously published 'pQTLs', is robust, important and timely, as the authors also outline in their rebuttal. As a technical report, this paper is great and performed at highest standards.

Response: We thank the reviewer for their efforts and appreciate the positive assessment of our responses to the remaining technical issues.

However, I still take issue in the argument and provided evidence by the authors related to the biological, let alone druggable, relevance of the additional/novel findings presented (see detailed comments below). This is still superficial, which is in part due to the limited sample size, and therefore rather limited 'revenue' of pQTLs with impactful consequences.

Response: Please see our responses to the detailed comments below.

I agree with the authors that pQTLs should ideally tag variants that modulate the abundance of protein in tissues or blood but would propose a more nuanced view that might be taken into account. There are clear cut examples, by which common coding variants associated with disease risk introduce alternate amino acid sequences of relevant proteins that are more or less well targeted with affinity reagents.

While this clearly breaks the underlying assumption of affinity binding reagent take up being proportional to protein abundance, it provides an important indication that the alternate protein product is indeed circulating in blood, and may further, possibly similarly importantly, indicate that protein function rather than protein levels are important for the disease process.

This important aspect is currently lacking in the study.

Response: We followed the reviewer's suggestion and added this aspect to the discussion as follows:

“While affinity binding is assumed to be proportional to protein abundance, derivation from this assumption also provides an important indication that the alternate protein product is indeed circulating in blood, indicating that protein function rather than protein levels are important for the disease process in such cases.”

Detailed comments:

1. I am sorry for being nit-picky, but MS does not measure protein abundances either. They measure the abundance of peptides that may (hopefully) be unique to proteins, but have similar limitations in terms of alternative peptides not in databases or similar that make the in silico ‘reproduction’ of proteins tricky. I agree that true pQTLs should affect abundance of proteins in plasma, but structural changes may well change behaviour of peptides in the MS or similar. In reality, neither affinity nor MS does really measure protein abundance but infer it indirectly with different sets and assumptions. Let alone the fact that protein function rather than abundance might be the truly interesting insight to get at. Further, one might even question the narrow definition of protein isoform, with PAVs likely introducing proteins with different functional properties that are, quite rightly, no longer detected by affinity reagents. This would explain the successful colocalization of very many proteins with disease endpoints that have likely a PAV as underlying causal variant, e.g., GDF-15 and hyperemesis gravidarum (that the authors, rightly?, flag as epitope effect).

Response: We agree with the argument and understand that the reviewer wishes to see a more nuanced definition of the concept of protein abundance as measured by MS methods. To accommodate this request, we included the following text reflecting their comments into the discussion:

“However, it should be noted that MS methods do not measure protein abundance directly either, but infer it indirectly from peptide abundances, which are proportional to the amount of digested and ionized protein fragments and their mapping to the protein isoforms that are assumed to be found in the sample.”

2. The justification for colocalization to attempt replication ‘To account for differences in genetic structure between the cohorts’ does not make sense. As outlined in my previous response, colocalization is highly sensitive to misspecification in the LD-backbone, most importantly, it assumes that the LD-backbone for the two traits is the very same. If the truly underlying causal variant is tagged by only partially overlapping haplotypes in the two cohorts due to ancestral differences, coloc will vote in favour of H3 (which may account for the ITLN1 finding presented).

Trans-ancestral fine-mapping would be needed but would also require whole-genome sequencing data.

Response: We agree that trans-ancestral fine-mapping using whole-genome sequencing data would be needed to identify the truly causal variant. However, our goal here was merely to test pQTL replication in another population, not causal variant identification.

As the reviewer correctly states, in cases where the LD-backbone is not the same in the two studies, coloc will vote in favour of H3. This is OK for our purposes, as we take a conservative approach and do not classify the pQTL as replicated in these cases. To make this clear we write “14 of the non-replicated pQTLs exhibited a significant but different genetic signal in both cohorts”.

Please note that we also provided at the reviewer’s previous request an alternative approach to assessing replication, which does not rely on the LD backbone, showing that both approaches yield comparable results.

3. How is the CFH locus novel? The AMD missense variant and proxies in LD are one of the most pleiotropic loci for the SomaLogic platform.

Response: We agree that the CFH locus is a pleiotropic locus both for the SomaLogic and the Olink platforms. However, we did not claim this locus as a novel discovery. We reported a new trans-pQTL for the BRE protein at this locus, adding to what is already known about the CFH locus from affinity approaches. We included this pQTL in the discussion primarily as an example of platform complementarity.

4. It is essential to demonstrate relevance of pQTLs beyond plasma protein levels, but the conclusion ‘and fourteen others that can now be considered in the development of drug targets for these diseases.’ is simply too early. It is by no means clear that the cis-pQTLs act only on the protein target or other genes nearby as demonstrated for eQTLs. Let alone that most GWAS traits linked do not represent disease that would be considered relevant for drug development efforts.

Response: In the last revision, reviewer #3 asked us to be more specific and enumerate the pQTLs that may be taken into consideration as future drug targets, which we did. The reaction of reviewer #2 to this revision shows that the counting of specific pQTLs may suggest a false level of certainty. We therefore reverted to our original wording referring to “many other [pQTLs] can now be taken into consideration in the development of drug targets for these diseases.” We believe that this is a fair statement that reflects the general interest of pQTLs for pharmaceutical research, but it does not claim that any specific pQTL now is already a firm drug target.

5. I am not happy with the response on the FUCA1/2 example. The authors still suggest relevance to AD, which is not supported by large-scale AD GWAS (the authors didn't even attempt to provide a look-up). FUCA1/2 encode for fucosidases with a potentially broad spectrum of protein targets, and the association with Total PHF-tau might well be due to horizontal rather than vertical pleiotropy. Let alone that the trait analyses mentions SNP x SNP interaction that is not otherwise specified. I like to note here that post-translational modifications and genes acting on those are known pleiotropic pQTLs and may, similar to epitope effects, affect protein quantification by affinity reagents, but also MS.

Response: We followed the reviewer's request and removed all references to AD.

Regarding the reported overlap with the association with Total PHF Tau in a SNP x SNP interaction model, we did specify the relevant reference to the reporting GWAS (ref28: Wang, H. et al. Genome-wide interaction analysis of pathological hallmarks in Alzheimer's disease. Neurobiol Aging 93, 61-68 (2020)).

Also, our general approach of using the Open Targets platform for variant annotation implies a general look-up for available GWAS, including AD. The fact that we did not report one means that no signal on AD as an endpoint has been reported so far.

In any case, the main reason why we chose to highlight this pQTL is that it involves pQTLs from the Olink, Somalogic and Seer platforms and on a same functional theme (protein glycosylation).

We now write:

"We also identified pQTLs that complement findings from affinity-based studies, such as trans-pQTL on the Olink platform for fucosidase FUCA1 (rs11155297) that we replicated. In addition, we found a cis-pQTL for FUCA2 at the same locus, which was not assayed by the Olink platform, but would be expected to account for the trans-association (Table S2). Interestingly, the strongest association by SOMAscan at this locus was with mannosidase MAN2B2. FUCA1, FUCA2, and MAN2B2 are all enzymes involved in the lysosomal degradation of glycoproteins and glycolipids. A lookup using the Open Targets platform additionally revealed a GWAS signal for "Total PHF-tau (SNP x SNP interaction) ($p = 2 \times 10^{-8}$)" [ref28]. These genetic signals obtained from three different proteomics platforms illustrate how pQTLs can be leveraged to generate hypotheses for the drug target discovery' process."

6. The SPINK5 examples lacks the link to the phenotype. While findings are reported, OpenTargets or other data bases do not link the pQTL to associated skin diseases, or any other disease.

Response: The specific variant at the SPINK5 locus has (as for now) not been associated with a disease endpoint in a GWAS. However, as we write in our paper, "mutations in SPINK5 have been

linked to skin disorders characterized by ichthyosis, such as Netherton syndrome, as well as to hair abnormalities". We decided not to speculate about this point in the paper, but it is conceivable that future sufficiently powered GWAS on presently more "exotic" traits, such as "dry skin" might find the SPINK5 locus. We added the following reference for clarity:

Sun Q, Burgren NM, Cheraghlou S, et al. The Genomic and Phenotypic Landscape of Ichthyosis: An Analysis of 1000 Kindreds. *JAMA Dermatol.* 2022;158(1):16–25.

The main reason why we discuss the SPINK5 case is not its clinical implication, but its role as a showcase example of what an isoform-pQTL looks like from an MS perspective. Its validation stems from the overlapping splice-QTL observed by GTex for which we provided regional association plots during the previous revision.

7. I struggle understanding the methodological framework of the missingness GWAS, which is an important addition for the overall technical validity of the paper as it tries to address potential epitope artefacts. It is very reassuring to see that half of the proteins had evidence from multiple nanoparticles, but that also meant that the other half is similar doubtful as epitope effects by affinity methods, I guess.

Response: We added the missingness GWAS to assess whether genetic effects may also affect nanoparticle binding. The fact that we found evidence to the contrary from multiple nanoparticles for half of the proteins does not imply that the other half is in doubt but merely means that no conclusions can be drawn for the other half. The nanoparticles have been designed to specifically enrich different proteins in the five extraction fractions; therefore, many proteins are expected not to be found in more than one fraction.

Are those pQTLs without support from multiple nanoparticles more likely to tag PAVs?

Response: This is an excellent question, which we examined and found that the answer is no. We previously reported that one in three proteins is harboring at least one PAV at a minor allele frequency > 10% (Suhre et al., *Nat Commun* **15**, 989 (2024)). We have now revisited that data and found that this proportion does not depend on whether the proteins were detected on a single or on multiple nanoparticles.

Also, why was not a logistic regression model used instead of a Fisher test that could cope with genetic confounding/population stratification, which likely accounted for the somewhat odd enrichment of loci across ethnicities?

Response: We agree that alternative approaches like logistic regression models could have been used to improve on genetic confounding/population stratification. However, given the objective of this analysis to primarily identify genetically associated missingness we believe that a Fisher's test is sufficient.